# Optimization over Sparse Support-Preserving Sets: Two-Step Projection with Global Optimality Guarantees

William de Vazelhes [1]   Xiao-Tong Yuan [2]   Bin Gu [3]

## Abstract

In sparse optimization, enforcing hard constraints using the $\ell_0$ pseudo-norm offers advantages like controlled sparsity compared to convex relaxations. However, many real-world applications demand not only sparsity constraints but also some extra constraints. While prior algorithms have been developed to address this complex scenario with mixed combinatorial and convex constraints, they typically require the closed form projection onto the mixed constraints which might not exist, and/or only provide local guarantees of convergence which is different from the global guarantees commonly sought in sparse optimization. To fill this gap, in this paper, we study the problem of sparse optimization with extra *support-preserving* constraints commonly encountered in the literature. We present a new variant of iterative hard-thresholding algorithm equipped with a two-step consecutive projection operator customized for these mixed constraints, serving as a simple alternative to the Euclidean projection onto the mixed constraint. By introducing a novel trade-off between sparsity relaxation and sub-optimality, we provide global guarantees in objective value for the output of our algorithm, in the deterministic, stochastic, and zeroth-order settings, under the conventional restricted strong-convexity/smoothness assumptions. As a fundamental contribution in proof techniques, we develop a novel extension of the classic three-point lemma to the considered two-step non-convex projection operator, which allows us to analyze the convergence in objective value in an elegant way that has not been possible with existing tech-

niques. In the zeroth-order case, such technique also improves upon the state-of-the-art result from (de Vazelhes et al., 2022), even in the case without additional constraints, by allowing us to remove a non-vanishing system error present in their work.

## 1. Introduction

In sparse optimization, directly enforcing sparsity with the $\ell_0$ pseudo-norm has several advantages over its convex relaxation counterpart. In compressive sensing for instance (Foucart & Rauhut, 2013), one may seek to recover an unknown vector, which sparsity level is known to be at most $k$. Similarly, in portfolio optimization, due to transaction costs, one may seek to ensure hard constraints on the maximum number of assets invested in (Brodie et al., 2009; DeMiguel et al., 2009). However, in several use cases, one may also seek to enforce additional constraints, such as, for instance, a budget constraint in the case of portfolio optimization, which can be enforced through an extra $\ell_1$ constraint, as in Takeda et al. (2013). As another example, in sparse non-negative matrix factorization, when estimating the hidden components, one seeks to enforce at the same time a norm constraint and a sparsity constraint (Hoyer, 2002). The problem of $\ell_0$ empirical risk minimization (ERM) with additional constraints can be formulated as follows, where $R$ is an empirical risk function, $\Gamma \subseteq \mathbb{R}^d$ denotes a convex constraint set, and $\|\cdot\|_0$ denotes the $\ell_0$ pseudo-norm (number of non-zero components of a vector):

$$\min_{\boldsymbol{w} \in \mathbb{R}^p} R(\boldsymbol{w}), \;\; \text{s.t.} \; \|\boldsymbol{w}\|_0 \leq k \text{ and } \boldsymbol{w} \in \Gamma. \qquad (1)$$

We assume throughout this paper that Problem (1) is well-posed, with $R$ being bounded from below and the set of minimizers being non-empty. In the literature, several algorithms have been developed to address such a problem with mixed constraints, but they typically require the existence of a closed form for the projection onto the mixed constraint, and/or their convergence guarantees are only local, which makes it difficult to estimate the sub-optimality of the output of the algorithm. More precisely, on one hand, some works provide convergence analyses for variants of a (non-convex) projected gradient descent, explicitly for mixed sparse constraints (Metel, 2023; Pan et al., 2017; Lu, 2015; Beck &

[1]GenBio AI, work done while at MBZUAI, Abu Dhabi, UAE [2]School of Intelligence Science and Technology, Nanjing University, Suzhou, China [3]School of Artificial Intelligence, Jilin University, China. Correspondence to: Xiao-Tong Yuan <xtyuan@nju.edu.cn >, Bin Gu <jsgubin@gmail.com>.

*Proceedings of the $42^{nd}$ International Conference on Machine Learning*, Vancouver, Canada. PMLR 267, 2025. Copyright 2025 by the author(s).

Hallak, 2016), or for general proximal terms (which encompasses our mixed constraints) (Frankel et al., 2014; Xu et al., 2019b; Attouch et al., 2013; De Marchi & Themelis, 2022; Yang & Yu, 2020; Gu et al., 2018; Yang & Li, 2023; Bolte et al., 2014; Boţ et al., 2016; Xu et al., 2019a; Li & Lin, 2015; Bauschke et al., 2017; 2019), but such analyses are only local. On the other hand, several existing works on Iterative Hard Thresholding (IHT) provide global guarantees on sub-optimality gap (Jain et al., 2014; Nguyen et al., 2017; Li et al., 2016; Shen & Li, 2017; de Vazelhes et al., 2022), but they do not apply to the mixed constraint case we consider. In between the two approaches, one can also find (Foygel Barber & Ha, 2018) and (Liu & Foygel Barber, 2020) which, in the deterministic case, give global guarantees for general non-convex constraints or thresholding operators, but which do not provide explicit convergence guarantees for the particular mixed constraint setting that we consider: their rates depend on some constants (the relative concavity or the local concavity constant) for which, up to our knowledge, an explicit form is still unknown for the mixed constraints we consider. We present a more detailed review of related works in Appendix B, and an overview of them in Table 1. To fill this gap, we focus on solving problem 1 in the case where $\Gamma$ belongs to a general family of *support-preserving* sets, which encompasses many usual sets encountered in the literature. As will be described in more detail in Section 2, such sets are convex sets for which the projection of a $k$-sparse vector onto them gets its support preserved, such as for instance $\ell_p$ norm balls (for $p \geq 1$), or a broader family of *sign–free* convex sets described for instance in (Lu, 2015) and (Beck & Hallak, 2016).

Adapted to the properties of such constraints, we propose a new variant of IHT, with a two-step projection operator, which, as a first step, identifies the set $S$ of coordinates of the top $k$ components of a given vector and sets the other components to 0 (hard-thresholding), and as a second step projects the resulting vector onto $\Gamma$. This two-step projection can offer a simpler alternative to Euclidean projection onto the mixed constraint in the cases where there is a closed form for the latter projection, and handle the cases where there is not. We then provide global sub-optimality guarantees without system error for the objective value, for such an algorithm as well as its stochastic and zeroth-order variants, under the restricted strong-convexity (RSC) and restricted smoothness (RSS) assumptions, in Theorems 3.7, 4.3, and 4.8. Key to our analysis is a novel extension of the three-point lemma to such non-convex setting with mixed constraints, which also allows, as a byproduct, to simplify existing proofs of convergence in objective value for IHT and its variants. In the zeroth-order case, such technique also allows to obtain, up to our knowledge, the first convergence in risk result without system error for a zeroth-order hard-thresholding algorithm. Additionally, our results high-

light a compromise between sparsity and sub-optimality gap specific to the additional constraints setting: through a free parameter $\rho$, one can obtain smaller upper bounds in terms of risk but at the cost of relaxing further the sparsity level of the iterates, or, alternatively, enforce sparser iterates but at the cost of a larger upper bound on the risk.

**Contributions.** We summarize the main contributions of our paper as follows:

1. We present a variant of IHT to solve hard sparsity problems with additional support-preserving constraints, using a novel two-step projection operator.
2. We describe a novel extension of the three-point lemma to such constraint which allows to simplify existing proofs for IHT and to provide global convergence guarantees in objective value without system error for the algorithm above, in the RSC/RSS setting, highlighting a novel trade-off between sparsity of iterates and sub-optimality gap in such mixed constraints setting.
3. We extend the above algorithm to the stochastic and zeroth-order optimization settings, obtaining similar global optimality guarantees in objective value (without system error) for such mixed constraints setting. In the zeroth-order case, this also provides, up to our knowledge, the first convergence result in objective value without system error for a zeroth-order hard-thresholding algorithm (with or without extra constraints).

## 2. Preliminaries

Throughout this paper, we adopt the following notations. For any $\boldsymbol{w} \in \mathbb{R}^d$, $\Pi_\Gamma(\boldsymbol{w})$ denotes a Euclidean projection of $\boldsymbol{w}$ onto $\Gamma$, that is $\Pi_\Gamma(\boldsymbol{w}) \in \arg\min_{\boldsymbol{z} \in \Gamma} \|\boldsymbol{w} - \boldsymbol{z}\|_2$, and $w_i$ denotes the $i$-th component of $\boldsymbol{w}$. $\mathcal{B}_0(k)$ denotes the $\ell_0$ pseudo-ball of radius $k$, i.e. $\mathcal{B}_0(k) = \{\boldsymbol{w} \in \mathbb{R}^d : \|\boldsymbol{w}\|_0 \leq k\}$, with $\|\cdot\|_0$ the $\ell_0$ pseudo-norm (i.e. the number of non-zero components of a vector). $\mathcal{H}_k$ denotes the Euclidean projection onto $\mathcal{B}_0(k)$, also known as the hard-thresholding operator (which keeps the $k$ largest (in magnitude) components of a vector, and sets the others to 0 (if there are ties, we can break them e.g. lexicographically)). $\|\cdot\|_p$ denotes the $\ell_p$ norm for $p \in [1, +\infty)$, and $\|\cdot\|$ the $\ell_2$ norm (unless otherwise specified). $[n]$ denotes the set $\{1, ..., n\}$ for $n \in \mathbb{N}^*$. For any $S \subseteq [d]$, $|S|$ denotes its number of elements. For any $\boldsymbol{w} \in \mathbb{R}^d$, $\mathrm{supp}(\boldsymbol{w})$ denotes its support, i.e. the set of coordinates of its non-zero components. We also introduce below the usual assumptions on $R$ for IHT proofs, i.e. RSC (Jain et al., 2014; Negahban et al., 2009; Loh & Wainwright, 2013; Yuan et al., 2017; Li et al., 2016; Shen & Li, 2017; Nguyen et al., 2017), and RSS (Jain et al., 2014; Li et al., 2016; Yuan et al., 2017).

**Assumption 2.1** ($(\nu_s, s)$-RSC). $R$ is $\nu_s$ restricted strongly convex with sparsity parameter $s$, i.e. it is differentiable, and there exists a generic constant $\nu_s$ such that for all $(\boldsymbol{x}, \boldsymbol{y}) \in$

*Table 1.* Comparison of results for Iterative Hard Thresholding with/without additional constraints. [1] $\mathcal{S}$: symmetric convex sets being sign-free or non-negative (Lu, 2015), $\mathcal{A}$: sets verifying Definition 2.3. [2] If a paper reports both $\|w - \bar{w}\|$ and $R(w) - R(\bar{w})$, we report only the latter. $\hat{T}$: time index of the $w$ returned by the method (e.g. $\hat{T} = \arg\min_{t \in [T]} R(w_t)$ ). $\bar{w}$: $\bar{k}$-sparse vector in $\Gamma$. $\Delta$: System error (non-vanishing term which depends on the gradient at optimality (e.g. $\mathbb{E}_i \|\nabla R_i(\bar{w})\|$, (see corresponding references))). [4]: $\kappa_s = \frac{L_s}{\nu_s}$ and $\kappa_{s'} = \frac{L_{s'}}{\nu_s}$ (cf. corresponding refs. for defs. of $s$ and $s'$). [3] SM: Lipschitz-smooth, D: Deterministic. S: Stochastic, Z: Zeroth-Order, L: Lipschitz continuous. [5]: see also Thm. 3.4, [6]: see also Thm. 4.2. ♣: Notably, we could eliminate $\Delta$ from (de Vazelhes et al., 2022).

| Reference | $\Gamma$[1] | Convergence[2] | $k$ | Setting[3] |
|---|---|---|---|---|
| (Jain et al., 2014)[5] | $\mathbb{R}^d$ | $R(w_{\hat{T}}) \leq R(\bar{w}) + \varepsilon$ | $\Omega(\kappa_s^2 \bar{k})$ | D, RSS, RSC |
| (Nguyen et al., 2017) | $\mathbb{R}^d$ | $\mathbb{E}\|w_{\hat{T}} - \bar{w}\| \leq \varepsilon + \mathcal{O}(\Delta)$ | $\Omega(\kappa_s^2 \bar{k})$ | S, RSS, RSC |
| (Li et al., 2016) | $\mathbb{R}^d$ | $\mathbb{E}R(w_{\hat{T}}) \leq R(\bar{w}) + \varepsilon + \mathcal{O}(\Delta)$ | $\Omega(\kappa_s^2 \bar{k})$ | S, RSS, RSC |
| (Zhou et al., 2018)[6] | $\mathbb{R}^d$ | $\mathbb{E}R(w_{\hat{T}}) \leq R(\bar{w}) + \varepsilon$ | $\Omega(\kappa_s^2 \bar{k})$ | S, RSS, RSC |
| (de Vazelhes et al., 2022) | $\mathbb{R}^d$ | $\mathbb{E}\|w_{\hat{T}} - \bar{w}\| \leq \varepsilon + \mathcal{O}(\mu) + \mathcal{O}(\Delta)$ | $\Omega(\kappa_{s'}^4 \bar{k})$ | S, Z, RSS', RSC |
| (Lu, 2015), (Beck & Hallak, 2016) | $\Gamma \in \mathcal{S}$ | local convergence | - | D, SM |
| (Metel, 2023) | $\ell_\infty$ ball around 0 | local convergence | - | S, Z, L |
| **IHT-2SP** (Thm. 3.7) | $\Gamma \in \mathcal{A}$ | $R(w_{\hat{T}}) \leq (1 + 2\rho)R(\bar{w}) + \varepsilon$ | $\Omega\left(\frac{\kappa_s^2 \bar{k}}{\rho^2}\right)$ | D, RSS, RSC |
| **HSG-HT-2SP** (Thm. 4.3) | $\Gamma \in \mathcal{A}$ | $\mathbb{E}R(w_{\hat{T}}) \leq (1 + 2\rho)R(\bar{w}) + \varepsilon$ | $\Omega\left(\frac{\kappa_s^2 \bar{k}}{\rho^2}\right)$ | S, RSS, RSC |
| **HZO-HT** (Thm. 4.7) | $\mathbb{R}^d$ | $\mathbb{E}[R(w_{\hat{T}}) - R(\bar{w})] \leq \varepsilon + \mathcal{O}(\mu)$♣ | $\Omega(\kappa_{s'}^2 \bar{k})$ | Z, RSS', RSC |
| **HZO-HT-2SP** (Thm. 4.8) | $\Gamma \in \mathcal{A}$ | $\mathbb{E}R(w_{\hat{T}}) \leq (1 + 2\rho)R(\bar{w}) + \varepsilon + \mathcal{O}(\mu)$ | $\Omega\left(\frac{\kappa_{s'}^2 \bar{k}}{\rho^2}\right)$ | Z, RSS', RSC |

$\mathbb{R}^d$ with $\|x - y\|_0 \leq s$: $R(y) \geq R(x) + \langle \nabla R(x), y - x \rangle + \frac{\nu_s}{2}\|x - y\|^2$

**Assumption 2.2** $((L_s, s)$-RSS$)$. $R$ is $L_s$ restricted smooth with sparsity level $s$, i.e. it is differentiable, and there exists a generic constant $L_s$ such that for all $(x, y) \in \mathbb{R}^d$ with $\|x - y\|_0 \leq s$:
$R(y) \leq R(x) + \langle \nabla R(x), y - x \rangle + \frac{L_s}{2}\|x - y\|^2$

We then define the notion of support-preserving set that we will use throughout the paper. It essentially requires that projecting any $k$-sparse vector $w$ onto $\Gamma$ preserves its support. That is, the convex constraint $\Gamma$ should be compatible with the sparsity level constraint $\|w\|_0 \leq k$.

**Definition 2.3** ($k$-support-preserving set). $\Gamma \subseteq \mathbb{R}^d$ is $k$-*support-preserving* , i.e. it is convex and for any $w \in \mathbb{R}^d$ such that $\|w\|_0 \leq k$, $\text{supp}(\Pi_\Gamma(w)) \subseteq \text{supp}(w)$.

*Remark* 2.4. Below we present some examples of usual sets that also verify Definition 2.3 (see Appendix D for a proof of such statements):

• Elementwise decomposable constraints, such as box constraints of the form $\{w \in \mathbb{R}^d : \forall i \in [d], l_i \leq w_i \leq u_i\}$,

with $l_i \leq 0$ and $u_i \geq 0$.
• Group-wise separable constraints where the constraint on each group is $k$-support-preserving (such as our constraints in Appendix H for the index tracking problem).
• Sign-free convex sets (Lu, 2015; Beck & Hallak, 2016) (def. in App. D), e.g. $\ell_q$ norm-balls.

## 3. Deterministic Case

### 3.1. Algorithm

**Two-Step Projection.** In all the algorithms of this paper, we will make use of a *two-step projection* operator (2SP), which is different in general from the usual Euclidean projection (EP), in order to obtain, from an arbitrary vector $w \in \mathbb{R}^d$, a vector in $w \in \mathcal{B}_0(k) \cap \Gamma$. We consider such a 2SP instead of EP since it enables the derivation of a variant of three-point lemma (Lemma 3.6) which can handle our specific non-convex mixed constraints, and is key to obtaining the convergence analyses we present in Sections 3 and 4. In addition, the 2SP can be more intuitive and efficient to implement than EP (see App. G for more dis-

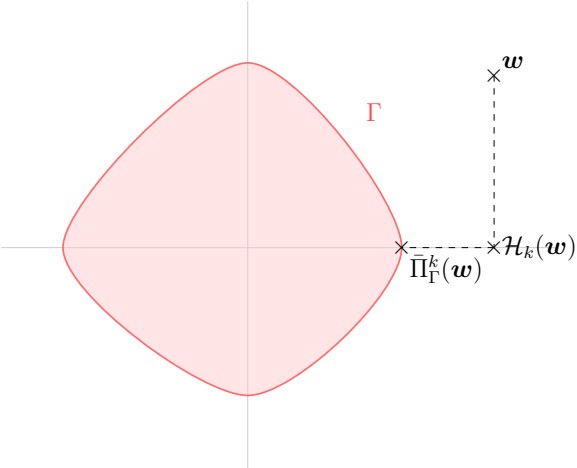

*Figure 1.* Support-preserving set and two-step projection ($d = 2, k = 1$).

cussions about 2SP vs EP). The 2SP procedure, which we denote by $\bar{\Pi}_\Gamma^k$, is as follows: we first project $\boldsymbol{w}$ onto $\mathcal{B}_0(k)$ through the hard-thresholding operator $\mathcal{H}_k$, to obtain a $k$-sparse vector $\boldsymbol{v}_k = \mathcal{H}_k(\boldsymbol{w})$. Then, we project $\boldsymbol{v}_k$ onto $\Gamma$, to obtain a final vector $\boldsymbol{w}_S = \Pi_\Gamma(\boldsymbol{v}_k)$, where $S = \mathrm{supp}(\boldsymbol{v}_k)$. Note that consequently, the obtained $\boldsymbol{w}_S$ is not necessarily the EP of $\boldsymbol{w}$ onto $\mathcal{B}_0(k) \cap \Gamma$, that is, we do not necessarily have $\boldsymbol{w}_S = \Pi_{\mathcal{B}_0(k) \cap \Gamma}(\boldsymbol{w})$. However, when $\Gamma$ is a k-support-preserving set, we have $\boldsymbol{w}_S \in \mathcal{B}_0(k) \cap \Gamma$ (since, by definition of a k-support-preserving set (Definition 2.3), $\mathrm{supp}(\boldsymbol{w}_S) \subseteq \mathrm{supp}(\boldsymbol{v}_k)$ and hence $\|\boldsymbol{w}_S\|_0 \leq \|\boldsymbol{v}_k\|_0 \leq k$), therefore each iteration remains feasible in the constraint. We illustrate such a two-step projection on Figure 1. We now present our full algorithm in the case where $R$ is a deterministic function without further knowledge of its structure. It is similar to the usual (non-convex) projected gradient descent algorithm, that is, a gradient update step followed by a projection step, except that instead of projecting onto $\Gamma \cap \mathcal{B}_0(k)$ using the Euclidean projection, we obtain a vector $\boldsymbol{w}_k \in \Gamma \cap \mathcal{B}_0(k)$ through the two-step projection method described above. We describe the algorithm in Algorithm 1 below.

---

**Algorithm 1:** Deterministic IHT with extra constraints (IHT-2SP)

**Input:** $\boldsymbol{w}_0$: initial value, $\eta$: learning rate, $T$: number of iterations

**for** $t = 1$ *to* $T$ **do**
  $\quad \boldsymbol{w}_t \leftarrow \bar{\Pi}_\Gamma^k(\boldsymbol{w}_{t-1} - \eta \nabla R(\boldsymbol{w}_{t-1}));$
**end**

**Output:** $\boldsymbol{w}_T$

---

*Remark* 3.1. In the case where $\Gamma$ is a *symmetric sign-free* convex set (we refer to (Lu, 2015) for the definition of such sets, which include for instance any $\ell_p$ norm constraint set

for $p \in [1, +\infty)$ ), then the two-step projection is actually the closed form of an Euclidean projection onto the mixed constraint $\Gamma \cap \mathcal{B}_0(k)$ (see Theorem 2.1 from (Lu, 2015)). Therefore, in such cases, Algorithm 1 is identical to a vanilla (non-convex) projected gradient descent algorithm (for which up to now there was still no global optimality guarantees in such a mixed constraints setting in the literature).

### 3.2. Convergence Analysis

Before proceeding with the convergence analysis, we first present below a variant of the usual three-point lemma from constrained convex optimization, which plays a key role in our proofs. The common three-point lemma for a projection onto a convex set $\mathcal{E}$ relates the distance between a point $\boldsymbol{w} \in \mathbb{R}^d$, its projection $\Pi_\mathcal{E}(\boldsymbol{w})$, and any vector $\bar{\boldsymbol{w}}$ from the set $\mathcal{E}$, through the relation $\|\boldsymbol{w} - \bar{\boldsymbol{w}}\|^2 \geq \|\Pi_\mathcal{E}(\boldsymbol{w}) - \boldsymbol{w}\|^2 + \|\Pi_\mathcal{E}(\boldsymbol{w}) - \bar{\boldsymbol{w}}\|^2$. Such a three-point lemma is used for instance in a general Bregman divergence form to prove convergence of mirror descent for smooth functions in (Bubeck et al., 2015). Indeed, although proving the convergence of projected gradient descent in the non-smooth case only needs the non-expansivity of projection onto a convex set, the proof for the smooth case usually needs such a three-point lemma, which can be seen as a stronger version of non-expansivity. However, due to the non-convexity of the $\ell_0$ pseudo-ball, the convex three-point lemma above does not hold. Fortunately, building upon Lemma 4.1 from (Liu & Foygel Barber, 2020), we can obtain a three-point lemma for projection onto the $\ell_0$ pseudo-ball.

**Lemma 3.2** ($\ell_0$ three-point lemma, proof in App. E.1.2). *Consider* $\boldsymbol{w}, \bar{\boldsymbol{w}} \in \mathbb{R}^p$ *with* $\|\bar{\boldsymbol{w}}\|_0 \leq \bar{k}$. *For any* $\bar{k} \leq k$, *with* $\beta := \frac{\bar{k}}{k}$, *it holds that:* $\|\mathcal{H}_k(\boldsymbol{w}) - \boldsymbol{w}\|^2 \leq \|\boldsymbol{w} - \bar{\boldsymbol{w}}\|^2 - \left(1 - \sqrt{\beta}\right) \|\mathcal{H}_k(\boldsymbol{w}) - \bar{\boldsymbol{w}}\|^2$.

Note that if $k \gg \bar{k}$, $\beta \to 0$ and we approach the usual three-point lemma from convex optimization. This is coherent with the literature on IHT, in which relaxing sparsity (i.e. considering some $k \gg \bar{k}$) is known to make the problem easier to solve (see also Remark 3.5 below). In addition, the inequality in Lemma 3.2 is tight with respect to the coefficient $\sqrt{\beta}$, as illustrated by the following lemma.

**Lemma 3.3** (Tightness, proof in App. E.1.3). *Consider an arbitrary pair of integers* $(k, \bar{k})$ *with* $k > \bar{k}$ *and an arbitrary scalar* $\rho \in (0, 1)$. *Then there exist* $\boldsymbol{w}$ *and* $\bar{\boldsymbol{w}}$ *with* $\|\boldsymbol{w}\|_0 = k$ *and* $\|\bar{\boldsymbol{w}}\|_0 = \bar{k}$ *such that the following holds:* $\|\boldsymbol{w} - \mathcal{H}_k(\boldsymbol{w})\|^2 > \|\boldsymbol{w} - \bar{\boldsymbol{w}}\|^2 - \|\mathcal{H}_k(\boldsymbol{w}) - \bar{\boldsymbol{w}}\|^2 + \rho\sqrt{\frac{\bar{k}}{k}}\|\mathcal{H}_k(\boldsymbol{w}) - \bar{\boldsymbol{w}}\|^2$.

Lemma 3.2 allows us to prove the following rate for convergence in risk of IHT without system error, which appeared first in (Jain et al., 2014). Our proof, however, is simpler than the original proof from Jain et al. (2014), as we will

discuss below.

**Theorem 3.4.** *(Equivalent to Thm. 1 from (Jain et al., 2014), see also Thm. 3.1 from (Liu & Foygel Barber, 2020). Proof in App. E.2.1) Assume that $\Gamma = \mathbb{R}^d$. Suppose that Assumption 2.1 and Assumption 2.2 hold, for $s = 2k$. Let $\eta = \frac{1}{L_s}$. Let $\bar{w}$ be an arbitrary $\bar{k}$-sparse vector. Suppose that $k \geq 4\kappa_s^2 \bar{k}$ with $\kappa_s := \frac{L_s}{\nu_s}$. Then for any $\varepsilon > 0$, the iterate of IHT satisfies $R(w_t) \leq R(\bar{w}) + \varepsilon$ if $t \geq \left\lceil \frac{2L_s}{\nu_s} \log \left( \frac{(L_s - \nu_s)\|w_0 - \bar{w}\|^2}{2\varepsilon} \right) \right\rceil + 1$.*

*Proof Sketch.* Using the $L_s$-RSS of $R$ and some algebraic manipulations, and denoting $g_t = \nabla R(w_t)$ and $v_t := \mathcal{H}_k(w_{t-1} - \frac{1}{L_s} g_{t-1})$ $(= w_t$ when $\Gamma = \mathbb{R}^d)$, we have:

$$R(v_t) \leq R(w_{t-1}) + \frac{L_s}{2}\left\|v_t - w_{t-1} + \frac{1}{L_s}g_{t-1}\right\|^2 \quad (2)$$

$$- \frac{1}{2L_s}\|g_{t-1}\|^2$$

$$\overset{(a)}{\leq} R(w_{t-1}) + \frac{L_s}{2}\left\|\bar{w} - w_{t-1} + \frac{1}{L_s}g_{t-1}\right\|^2 \quad (3)$$

$$- \frac{L_s}{2}(1 - \sqrt{\beta})\|v_t - \bar{w}\|^2 - \frac{1}{2L_s}\|g_{t-1}\|^2$$

$$\overset{(b)}{\leq} R(\bar{w}) + \frac{L_s - \nu_s}{2}\|w_{t-1} - \bar{w}\|^2 \quad (4)$$

$$- \frac{L_s}{2}(1 - \sqrt{\beta})\|v_t - \bar{w}\|^2, \quad (5)$$

where (a) follows from Lemma 3.2, and in (b) we used the RSC of $R$ with some rearrangements. The proof for Theorem 3.4 can be concluded with telescopic sum arguments. □

*Remark* 3.5 (Necessity of $k = \Omega(\bar{k}\kappa^2)$). Note that the relaxation of $k$ to $\Omega(\bar{k}\kappa^2)$ in Theorem 3.4 is unimprovable for IHT, as we detail in Appendix E.3 with a counter-example, similar to but slightly simpler than the counter-example from Appendix E.1 in (Axiotis & Sviridenko, 2022)). Therefore, we highlight that all of the following results in this paper will also be expressed in terms of such a relaxed $k$: this is a fundamental limitation of IHT, and not a limitation of our proof techniques. More details on such relaxation (which is widespread amongst IHT-type algorithms as can be seen in Table 1) and how it is a natural way to obtain global guarantees for sparsity enforcing algorithms, can be found in Liu & Foygel Barber (2020); Axiotis & Sviridenko (2021; 2022).

**Comparison with Previous Proofs.** Perhaps the original and most widespread proof framework for convergence in risk of IHT without system error is the one from (Jain et al., 2014) Theorem 1. Their proof framework is also used for instance in some stochastic extensions of IHT (see Theorem 2 in (Zhou et al., 2018), or Theorems 1 and 2 in

(Peste et al., 2021), even if Peste et al. (2021) assume $R$ to have a $\bar{k}$-sparse minimizer which is a strong requirement). The proof from (Jain et al., 2014) uses specific properties of the hard-thresholding operator to carefully bound the magnitude of the components of $\nabla R(w_t)$ on various sets of coordinates (the support of $w_t$, $w_{t+1}$, and $\bar{w}$, and some intersections and unions of such sets). Using such techniques, however, makes it difficult to derive proofs of IHT in other settings (stochastic, zeroth-order, extra constraints). However, recently, Liu & Foygel Barber (2020) provided a proof of convergence for IHT which avoids such complex considerations about the support sets of the gradient, using their Lemma 4.1 on the *relative concavity* of the hard-thresholding operator. Our work goes in a similar line of work, but we build upon their Lemma 4.1 to prove a three-point lemma for hard-thresholding (our Lemma 3.2) which allows us to obtain simple proof frameworks also for the stochastic case (retrieving the previous from (Zhou et al., 2018)) and the zeroth-order case (obtaining a new result). But perhaps more importantly, we are able to extend our Lemma 3.2 to the case with extra constraints $\Gamma$ verifying Definition 2.3 (Lemma 3.6 below). Such a lemma will allows us to obtain convergence results in the new extra constraints setting that we consider in this paper (providing three new results, in the deterministic, stochastic, and zeroth-order case). It relates together the four points involved in the two step projection ($w \in \mathbb{R}^d$, $\mathcal{H}_k(w)$, $\bar{\Pi}_\Gamma^k(w)$, and $\bar{w} \in \Gamma \cap \mathcal{B}_0(k)$ ).

**Lemma 3.6** (Constrained $\ell_0$-Three-Point, proof in App. E.1.4). *Suppose that Definition 2.3 holds for a set $\Gamma$. Consider $w, \bar{w} \in \mathbb{R}^p$ with $\|\bar{w}\|_0 \leq \bar{k}$ and $\bar{w} \in \Gamma$. Then the following holds for any $k > \bar{k}$:*

$$\|\bar{\Pi}_\Gamma^k(w) - w\|^2 \leq \|w - \bar{w}\|^2 - \|\bar{\Pi}_\Gamma^k(w) - \bar{w}\|^2$$

$$+ \sqrt{\beta}\|\mathcal{H}_k(w) - \bar{w}\|^2, \text{ with } \beta := \frac{\bar{k}}{k}.$$

Equipped with such lemma, we can now present the convergence analysis of Algorithm 1 below, using the assumptions from Section 2, and we will describe how the results give rise to a trade-off between the sparsity of the iterates and the tightness of the sub-optimality bound, specific to our mixed constraints setting.

**Theorem 3.7** (Proof in App. E.2.2). *Suppose that Assumption 2.1 and 2.2 hold with $s = 2k$, that $R$ is non-negative (without loss of generality), and let $\Gamma$ be a set verifying Definition 2.3 . Let $\eta = \frac{1}{L_s}$, and $\bar{w}$ be an arbitrary $\bar{k}$-sparse vector. Let $\rho \in (0, \frac{1}{2}]$ be an arbitrary scalar. Suppose that $k \geq \frac{4(1-\rho)^2 L_s^2}{\rho^2 \nu_s^2}\bar{k}$. Then for any $\varepsilon > 0$, for $T \geq \left\lceil \frac{L_s}{\nu_s} \log \left( \frac{(L_s - \nu_s)\|w_0 - \bar{w}\|^2}{2\varepsilon(1-\rho)} \right) \right\rceil + 1 = \mathcal{O}(\kappa_s \log(\frac{1}{\varepsilon}))$, the iterates of IHT-2SP satisfy:*

$$\min_{t \in [T]} R(w_t) \leq (1 + 2\rho)R(\bar{w}) + \varepsilon.$$

*Further, if $\bar{w}$ is a global minimizer of $R$ over $\mathcal{B}_0(k) := \{w : \|w\|_0 \leq k\}$, then, with $\rho = 0.5$ in the expressions of $k$ and $T$ above:* $\min_{t \in [T]} R(w_t) \leq R(\bar{w}) + \varepsilon$.

*Proof Sketch.* To obtain the proof for general $\Gamma$, we reiterate a similar proof as for Theorem 3.4, but this time, instead of Lemma 3.2, we use our more general Lemma 3.6, adapted to general $\Gamma$ and to our two-step projection technique, to obtain (see the Proof Sketch of Thm. 3.4 for the definition of $v_t$):

$$R(w_t) \leq R(\bar{w}) + \frac{L_s - \nu_s}{2}\|w_{t-1} - \bar{w}\|^2$$
$$- \frac{L_s}{2}\|w_t - \bar{w}\|^2 + \frac{L_s}{2}\sqrt{\beta}\|v_t - \bar{w}\|^2. \quad (6)$$

Finally, taking a convex combination of equations 2 ($\times \rho$) and 6 ($\times(1-\rho)$) for $\rho \in (0, 0.5]$, using the bound $\|w_t - \bar{w}\|^2 \leq \|v_t - \bar{w}\|^2$ (non-expansiveness of convex projection onto $\Gamma$), and carefully tuning $k$ depending on $\rho$ (resulting in our final trade-off between sparsity and optimality), we can fall back to a telescopic sum and conclude the proof. $\square$

*Remark* 3.8. Similarly to the work of (Jain et al., 2014), in the setting of linear regression with sub-Gaussian design, we can use results from (Agarwal et al., 2010) to find the number of samples needed to verify the assumptions of Theorem 3.4. We provide such analysis in Appendix C.

*Remark* 3.9. Theorem 3.7 therefore provides a global convergence guarantee in objective value. However, contrary to usual guarantees for IHT algorithms under RSS/RSC conditions (which are bounds of the form $R(w_t) \leq R(\bar{w}) + \varepsilon$ for some $t$), our bound is of the form $R(w_t) \leq (1 + 2\rho)R(\bar{w}) + \varepsilon$. There is a trade-off about the choice of $\rho \in (0, 0.5]$. On one hand, $\rho \to 0$ is preferred in view of the $RHS$ of above bound. On the other hand, the sparsity-level relaxation condition $k \geq \frac{4(1-\rho)^2 L_s^2}{\rho^2 \nu_s^2}\bar{k}$ prefers $\rho \to 0.5$. We illustrate such a trade-off on some synthetic experiments in Appendix H.1.

# 4. Extensions: Stochastic and Zeroth-Order Cases

In this section, we provide extensions of Algorithm 1 to the stochastic and zeroth-order sparse optimization problems, and provide the corresponding convergence guarantees in objective value without system error.

## 4.1. Stochastic Optimization

In this section, we consider the previous risk minimization problem, in a finite-sum setting, i.e. with $R(w) = \frac{1}{n}\sum_{i=1}^n R_i(w)$, as in (Zhou et al., 2018; Nguyen et al., 2017): indeed, stochastic algorithms can tackle more easily large-scale datasets where estimating the full $\nabla R(w)$ is expensive.

### 4.1.1. ALGORITHM

We describe the stochastic variant of our previous Algorithm 1 in Algorithm 2 below, which is an extension of the algorithm from (Zhou et al., 2018), to the considered mixed constraints problem setting, using our two-step projection. More precisely, we approximate the gradient of $R$ by a minibatch stochastic gradient with a batch-size increasing exponentially along training, and following the gradient step, we apply our two-step projection operator.

---

**Algorithm 2:** Hybrid Stochastic IHT with Extra Constraints (HSG-HT-2SP)

**Input:** $w_0$: initial point, $\eta$: learning rate, $T$: number of iterations, $\{s_t\}$: mini-batch sizes.

**for** $t = 1$ *to* $T$ **do**

    Uniformly sample $s_t$ indices $\mathcal{S}_t$ from $[n]$ without replacement

    Compute the approximate gradient

    $g_{t-1} = \frac{1}{s_{t-1}}\sum_{i_t \in \mathcal{S}_t} \nabla R_{i_t}(w_{t-1})$

    $w_t = \bar{\Pi}_\Gamma^k(w_{t-1} - \eta g_{t-1})$;

**end**

**Output:** $\hat{w}_T \in \arg\min_{w \in \{w_1, \dots, w_T\}} R(w)$.

---

### 4.1.2. CONVERGENCE ANALYSIS

Before proceeding with the convergence analysis, we make an additional assumption on the population variance of the stochastic gradients, similar to the one in (Mishchenko et al., 2020).

**Assumption 4.1** (Bounded stochastic gradient variance). For any $w$, the population variance of the gradient estimator is bounded by $B$: $\frac{1}{n}\sum_{i=1}^n \|\nabla R_i(w) - \nabla R(w)\|^2 \leq B$.

We now present our convergence analysis, first with $\Gamma = \mathbb{R}^d$, retrieving Theorem 2 from (Zhou et al., 2018).

**Theorem 4.2** (Equivalent to Theorem 2 from (Zhou et al., 2018), Proof in App. F.2.2). *Assume that $\Gamma = \mathbb{R}^d$. Suppose that Assumption 2.1 and Assumption 2.2 hold with $s = 3k$, and that Assumption 4.1 also holds. Let $\bar{w}$ be an arbitrary $\bar{k}$-sparse vector. Let $C$ be an arbitrary positive constant. Assume that we run HSG-HT-2SP (Algorithm 2) for $T$ timesteps, with $\eta = \frac{1}{L_s + C}$, and denote $\alpha := \frac{C}{L_s} + 1$ and $\kappa_s := \frac{L_s}{\nu_s}$. Suppose that $k \geq 4\alpha^2 \kappa_s^2 \bar{k}$. Finally, assume that we take the following batch-size: $s_t := \lceil \frac{\tau}{\omega^t} \rceil$ with $\omega := 1 - \frac{1}{4\alpha\kappa_s}$ and $\tau := \frac{\eta B}{C}$. Then, we have the following convergence rate:*

$$\mathbb{E}R(\hat{w}_T) - R(\bar{w}) \leq 2\alpha^2 L_s \kappa_s \omega^T \left(\|\bar{w} - w_0\|^2 + \frac{4}{3}\right).$$

Such a Theorem is equivalent to Theorem 2 from (Zhou et al., 2018), however, the proof from (Zhou et al., 2018) is

based on the same framework as (Jain et al., 2014), which makes it more complex. Our proof, on the other hand, is very similar to our proof of Theorem 3.4 above (i.e. closer to convex constrained optimization proofs as discussed above), and simply incorporates the variance of the stochastic gradient estimator (exponentially decreasing thanks to the exponentially increasing batch-size) in a properly weighted telescopic sum (with a technique inspired from (Liu & Foygel Barber, 2020)). We believe this makes the proof more readily usable for future extensions of IHT. And in particular, using a similar technique as for Theorem 3.7, we can extend our result to the case with an extra constraint $\Gamma$ verifying Definition 2.3: we present such extension in Theorem 4.3 below.

**Theorem 4.3** (Proof in App. F.2.3). *Suppose that Assumptions 2.1 and 2.2 hold with $s = 2k$, that 4.1 holds, that $R$ is non-negative (without loss of generality), and let $\Gamma$ be a set verifying Definition 2.3. Let $\bar{w}$ be an arbitrary $\bar{k}$-sparse vector. Let $C$ be an arbitrary positive constant. Assume that we run HSG-HT-2SP (Algorithm 2) for $T$ timesteps, with $\eta = \frac{1}{L_s + C}$, and denote $\alpha := \frac{C}{L_s} + 1$ and $\kappa_s := \frac{L_s}{\nu_s}$. Suppose that $k \geq 4\alpha^2 \frac{1}{\rho^2} \kappa_s^2 \bar{k}$ for some $\rho \in (0, 1)$. Finally, assume that we take the following batch-size: $s_t := \left\lceil \frac{\tau}{\omega^t} \right\rceil$ with $\omega := 1 - \frac{1}{4\alpha \frac{1}{\rho} \kappa_s}$ and $\tau := \frac{\eta B}{C}$. Then, we have the following convergence rate:*

$$\mathbb{E} \min_{t \in [T]} R\left(w_t\right) - (1 + 2\rho) R(\bar{w})$$
$$\leq 2 \frac{\alpha^2}{\rho(1 - \rho)} L_s \kappa_s \omega^T \left( \|\bar{w} - w_0\|^2 + \frac{4}{3} \right).$$

*Further, if $\bar{w}$ is a global minimizer of $R$ over $\mathcal{B}_0(k) := \{w : \|w\|_0 \leq k\}$, then, with $\rho = 0.5$:*

$$\mathbb{E} \min_{t \in [T]} R\left(w_t\right) - R(\bar{w}) \leq 8\alpha^2 L_s \kappa_s \omega^T \left( \|\bar{w} - w_0\|^2 + \frac{4}{3} \right).$$

**Corollary 4.4** (Proof in App. F.3.). *Therefore, the number of calls to a gradient $\nabla R_i$ (#IFO), and the number of hard thresholding operations (#HT) such that the left-hand sides in Theorem 4.3 above are smaller than some $\varepsilon > 0$, are respectively: #HT $= \mathcal{O}(\kappa_s \log(\frac{1}{\varepsilon}))$ and #IFO $= \mathcal{O}\left(\frac{\kappa_s}{\nu_s \varepsilon}\right)$.*

## 4.2. Zeroth-Order Optimization (ZOO)

We now consider the zeroth-order (ZO) case (Nesterov & Spokoiny, 2017), in which one does not have access to the gradient $\nabla R(w)$, but only to function values $R(w)$, which arises for instance when the dataset is private as in distributed learning (Gratton et al., 2021; Zhang et al., 2021) or the model is private as in black-box adversarial attacks (Liu et al., 2018), or when computing $\nabla R(w)$ is too expensive such as in certain graphical modeling tasks (Wainwright et al., 2008). The idea is then to approximate

$\nabla R(w)$ using finite differences. We refer the reader to (Berahas et al., 2021) and (Liu et al., 2020) for an overview of ZO methods.

### 4.2.1. ALGORITHM

In this section, we describe the ZO version of our algorithm. At its core, it uses the ZO estimator from (de Vazelhes et al., 2022). We present the full algorithm in Algorithm 3, where $\mathcal{D}_{s_2}$ is a uniform probability distribution on the following set $\mathcal{B}$ of unit spheres supported on supports of size $s_2 \leq d$: $\mathcal{B} := \{w \in \mathbb{R}^d : \|w\|_0 \leq s_2, \|w\|_2 \leq 1\}$. We can sample from this set by first sampling a random support of size $s_2$, and then sampling from the unit sphere on that support. If we choose $s_2 := d$, this estimator simply becomes the vanilla ZO estimator with unit-sphere smoothing (Liu et al., 2020). Choosing $s_2 < d$ allows to avoid the full-smoothness assumption and can reduce memory consumption by allowing to sample random vectors of size $s_2$ instead of $d$ (see (de Vazelhes et al., 2022) for more details on such a ZO estimator). The difference with (de Vazelhes et al., 2022) (in addition to the mixed constraint setting and the use of the 2SP) is that in our case we sample an exponentially increasing number of random directions, which allows us, for the first time up to our knowledge, to obtain convergence in risk for a ZO hard-thresholding algorithm without any system error (except the unavoidable system error due to the smoothing $\mu$ (cf. Remark 5 in (Ajalloeian & Stich, 2020))).

---

**Algorithm 3:** Hybrid ZO IHT with Extra Constraints (HZO-HT-2SP)

**Input:** $w_0$: initial point, $\eta$: learning rate, $T$: number of iterations, $s_2$: size of the random supports, $\{q_t\}$: number of random directions.

**for** $t = 1$ *to* $T$ **do**
  Uniformly sample $q_{t-1}$ i.i.d. random directions $\{u_i\}_{i=1}^{q_{t-1}} \sim \mathcal{D}_{s_2}$
  Compute the approximate gradient $g_t = \frac{1}{q_{t-1}} \sum_{i=1}^{q_{t-1}} \frac{d}{\mu} \left(R(w_{t-1} + \mu u_i) - R(w_{t-1})\right) u_i$
  $w_t = \bar{\Pi}_\Gamma^k(w_{t-1} - \eta g_{t-1})$
**end**

**Output:** $\hat{w}_T \in \arg\min_{w \in \{w_1, \dots, w_T\}} R(w)$.

---

### 4.2.2. CONVERGENCE ANALYSIS

**Assumption 4.5** (($L_s, s$)-RSS'). (Shen & Li, 2017; Nguyen et al., 2017) $R$ is $L_s$-restricted strongly smooth with sparsity level $s$, i.e. it is differentiable, and there exist a generic constant $L_s$ such that for all $(x, y) \in \mathbb{R}^d$ with $\|x - y\|_0 \leq s$: $\|\nabla R(x) - \nabla R(y)\| \leq L_s \|x - y\|$.

*Remark* 4.6. Note that if a convex function $R$ is ($L_s, s$)-RSS', then it is also ($L_s, s$)-RSS (this can be proven in the same way as for usual smoothness in convex optimization

(see Lemma 1.2.3 from (Nesterov, 2003)). However, the converse is not true here, contrary to what holds for usual smooth and convex functions (cf. Theorem 2.1.5 from (Nesterov, 2003)), as we show through some counter-example in Appendix F.1. Assumption 4.5 is indeed slightly more restrictive than Assumption 2.2, but it is necessary when working with ZO gradient estimators (see more details in (de Vazelhes et al., 2022)).

We now present our main convergence theorem for the ZO setting, first when $\Gamma = \mathbb{R}^d$.

**Theorem 4.7** (Proof in App. F.4.2). *Assume that $\Gamma = \mathbb{R}^d$. Let $\bar{w}$ be an arbitrary $\bar{k}$-sparse vector. Let $s = 3k$, and $s_2 \in \{1, ..., d\}$. Assume that $R$ is $(L_{s'}, s')$-RSS' with $s' = \max(s_2, s)$, and $(\nu_s, s)$-restricted strongly convex. Denote $\kappa_s := \frac{L_{s'}}{\nu_s}$. Let $C$ be an arbitrary positive constant, and denote $\varepsilon_F := \frac{2d}{(s_2+2)}\left(\frac{(s-1)(s_2-1)}{d-1} + 3\right)$, $\varepsilon_{abs} := 2dL_{s'}^2 s s_2 \left(\frac{(s-1)(s_2-1)}{d-1} + 1\right)$, and $\varepsilon_\mu := L_{s'}^2 sd$. Assume that we run HZO-HT-2SP (Algorithm 3) for $T$ timesteps, with $\eta = \frac{1}{L_{s'}+C} = \frac{1}{\alpha L_{s'}}$, with $\alpha := \frac{C}{L_{s'}} + 1$. Suppose that $k \geq 16\alpha^2 \kappa_s^2 \bar{k}$. Finally, assume that we take the following number $q_t$ of random directions at each iteration: $q_t := \left\lceil \frac{\tau}{\omega^t} \right\rceil$ with $\omega := 1 - \frac{1}{8\alpha\kappa_s}$ and $\tau := 16\kappa_s \frac{\varepsilon_F}{(\alpha-1)}$. Then, we have the following convergence rate, with $Z = \varepsilon_\mu \left(\frac{2}{\nu_s} + \frac{1}{C}\right) + \frac{\varepsilon_{abs}}{C}$:*

$$\mathbb{E}R(\hat{w}_T) - R(\bar{w})$$
$$\leq 4\alpha^2 L_{s'} \kappa_s \omega^T \left(\|\bar{w} - w_0\|^2 + \frac{1}{3}\frac{\eta\|\nabla R(\bar{w})\|^2}{\kappa_s L_{s'}}\right) + Z\mu^2,$$

Such a novel result illustrates the power of proof techniques based on our three-point lemma. Up to our knowledge, it is the first global convergence guarantee without system error for a ZO hard-thresholding algorithm (see Table 1), and as such, is a significant improvement over the result from (de Vazelhes et al., 2022). Our proof differs from the one in (de Vazelhes et al., 2022): that latter uses a bound on the expansivity of the hard-thresholding operator, and only provides a result in terms of $\|w - \bar{w}\|$, with a non-vanishing system error which depends on $\nabla R(w)$ (cf. Table 1). We now present our Theorem in the case of a general support-preserving convex set $\Gamma$.

**Theorem 4.8** (Proof in App. F.4.3). *Suppose that Assumptions 2.1, 2.3, and 4.5 hold with $s = 3k$, that $R$ is non-negative (without loss of generality), and let $\Gamma$ be a set verifying Definition 2.3. Let $\bar{w}$ be an arbitrary $\bar{k}$-sparse vector. Let $s_2 \in \{1, ..., d\}$. Denote $\kappa_s := \frac{L_{s'}}{\nu_s}$. Let $C$ be an arbitrary positive constant, and denote $\varepsilon_F := \frac{2d}{(s_2+2)}\left(\frac{(s-1)(s_2-1)}{d-1} + 3\right)$, $\varepsilon_{abs} := 2dL_{s'}^2 s s_2 \left(\frac{(s-1)(s_2-1)}{d-1} + 1\right)$, and $\varepsilon_\mu := L_{s'}^2 sd$. Assume*

*that we run HZO-HT-2SP (Algorithm 3) for $T$ timesteps, with $\eta = \frac{1}{L_{s'}+C} = \frac{1}{\alpha L_{s'}}$, with $\alpha := \frac{C}{L_{s'}} + 1$. Suppose that $k \geq 16\frac{\alpha^2}{\rho^2}\kappa_s^2 \bar{k}$ for some $\rho \in (0, 1)$. Finally, assume that we take $q_t$ random directions at each iteration, with $q_t := \left\lceil \frac{\tau}{\omega^t} \right\rceil$ with $\omega := 1 - \frac{1}{8\frac{1}{\rho}\alpha\kappa_s}$ and $\tau := 16\kappa_s \frac{\varepsilon_F}{(\alpha-1)}$. Then, we have the following convergence rate:*

$$\mathbb{E}\min_{t \in [T]} R(w_t) - (1 + 2\rho)R(\bar{w})$$
$$\leq 4\frac{\alpha^2}{\rho(1-\rho)}L_{s'}\kappa_s\omega^T\left(\|\bar{w} - w_0\|^2 + \frac{1}{3}\frac{\eta\|\nabla R(\bar{w})\|^2}{\kappa_s L_{s'}}\right)$$
$$+ Z\mu^2,$$

*with $Z = \frac{1}{1-\rho}\left(\varepsilon_\mu\left(\frac{2}{\nu_s} + \frac{1}{C}\right) + \frac{\varepsilon_{abs}}{C}\right)$. Further, if $\bar{w}$ is a global minimizer of $R$ over $\mathcal{B}_0(k) := \{w : \|w\|_0 \leq k\}$, then, with $\rho = 0.5$:*

$$\mathbb{E}\min_{t \in [T]} R(w_t) - R(\bar{w})$$
$$\leq 16\alpha^2 L_{s'}\kappa_s\omega^T\left(\|\bar{w} - w_0\|^2 + \frac{1}{3}\frac{\eta\|\nabla R(\bar{w})\|^2}{\kappa_s L_{s'}}\right)$$
$$+ Z\mu^2.$$

**Corollary 4.9** (Proof in App. F.5.). *Additionally, the number of calls to $R$ (#IZO), and the number of hard thresholding operations (#HT) such that the left-hand sides in Theorem 4.8 above are smaller than $\varepsilon + Z\mu^2$, for some $\varepsilon > 0$ are respectively: #HT $= \mathcal{O}(\kappa_s \log(\frac{1}{\varepsilon}))$ and #IZO $= \mathcal{O}\left(\varepsilon_F \frac{\kappa_s^3 L_s}{\varepsilon}\right)$. Note that if $s_2 = d$ (in which case Assumption 4.5 becomes the usual (unrestricted) smoothness assumption), we have $\varepsilon_F = \mathcal{O}(s) = \mathcal{O}(k)$, and therefore we obtain a query complexity that is dimension independent.*

Such a query complexity result also holds when $\Gamma = \mathbb{R}^d$ (cf. Corollary F.8 in Appendix). (de Vazelhes et al., 2022) also achieved a dimension independent rate, but their convergence result exhibited a potentially large non-vanishing system error (cf. Table 1), which we do not have in Theorems 4.7 and 4.8. In strongly convex and smooth ZOO, a dimension independent query complexity is impossible to achieve (Jamieson et al., 2012), unless with additional assumptions (Golovin et al., 2019; Sokolov et al., 2018; Wang et al., 2018; Cai et al., 2022; 2021; Balasubramanian & Ghadimi, 2018; Cai et al., 2022; Liu & Yang, 2021; Jamieson et al., 2012; Nozawa et al., 2024; Yue et al., 2023). Our work confirms that, instead of making extra assumptions, a possible way to obtain a dimension independent query complexity is to instead consider optimization with $\ell_0$ constraints.

## 5. Conclusion

In this paper, we provided global optimality guarantees for variants of iterative hard thresholding that can handle extra

convex support-preserving constraints for sparse learning, via a two-step projection algorithm. We provided our analysis in deterministic, stochastic, and zeroth-order settings. To that end, we used a variant of the three-point lemma, adapted to such mixed constraints, which allows to simplify existing proofs for vanilla constraints (and to provide a new kind of result in the ZO setting), as well as obtaining new proofs in such combined constraints setting. Finally, it would also be interesting to extend this work to a broader family of sparsity structures and constraints, for instance, to matrices or graphs. We leave this for future work.

## Acknowledgements

This work was supported by the Special Fund for Key Program of Science and Technology of Jiangsu Province under Grant No. BG2024042, the Natural Science Foundation of China (NSFC) under Grant U21B2049, and Ant Group through CCF-Ant Research Fund under Grant No.20240512. We are grateful to ICML reviewers for their valuable feedback and suggestions.

## Impact Statement

This paper presents work whose goal is to advance the field of sparse optimization for machine learning. There are many potential societal consequences of our work, none of which we feel must be specifically highlighted here.

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

# Appendix

## A. Notations

Below we aggregate the various notations used throughout the paper, for ease of reference.

- $\Pi_\Gamma(\boldsymbol{w})$: Euclidean projection of $\boldsymbol{w}$ onto a set $\Gamma$, i.e. $\Pi_\Gamma(\boldsymbol{w}) \in \arg\min_{\boldsymbol{z} \in \Gamma} \|\boldsymbol{w} - \boldsymbol{z}\|_2$.

- $w_i$: $i$-th component of $\boldsymbol{w}$.

- $\|\cdot\|$: $\ell_0$ pseudo-norm (number of non-zero components of a vector).

- $\mathcal{B}_0(k)$: $\ell_0$ pseudo-ball of radius $k$, i.e. $\mathcal{B}_0(k) = \{\boldsymbol{w} \in \mathbb{R}^d : \|\boldsymbol{w}\|_0 \leq k\}$.

- $\mathcal{H}_k$: Euclidean projection onto $\mathcal{B}_0(k)$, also known as the hard-thresholding operator (which keeps the $k$ largest (in magnitude) components of a vector, and sets the others to 0 (if there are ties, we can break them e.g. lexicographically)).

- $\bar{\Pi}_\Gamma^k$: Two-step projection of sparsity $k$ onto the set $\Gamma$, i.e. $\bar{\Pi}_\Gamma^k(\cdot) = \Pi_\Gamma(\mathcal{H}_k(\cdot))$.

- $\|\cdot\|_p$: $\ell_p$ norm for $p \in [1, +\infty)$.

- $\|\cdot\|$: $\ell_2$ norm.

- $[n]$: set $\{1, ..., n\}$ for $n \in \mathbb{N}^*$.

- $|S|$: number of elements of a set $S \subseteq [d]$.

- $\operatorname{supp}(\boldsymbol{w})$: support of a vector $\boldsymbol{w} \in \mathbb{R}^d$, i.e. the set of coordinates of its non-zero components.

- 2SP: two-step projection

- EP: Euclidean projection

## B. Related Works

Below we present a more detailed review of the related works.

### B.1. Local Guarantees for Combined Constraints

Among the works considering optimization over the intersection of the $\ell_0$ pseudo-ball of radius $k$ and a set $\Gamma$, (Metel, 2023) analyze the convergence of a first-order and zeroth-order stochastic algorithm with a weighted $\ell_0$ group norm constraint (which generalizes the $\ell_0$ norm), combined with an $\ell_\infty$ ball constraint. (Pan et al., 2017) provide a deterministic algorithm which can tackle extra positivity constraints. (Lu, 2015) and (Beck & Hallak, 2016) analyze the convergence of variants of hard-thresholding in the deterministic case, with extra constraints that are symmetric and sign-free or positive. Other line of works such as (Frankel et al., 2014; Xu et al., 2019b; Attouch et al., 2013; De Marchi & Themelis, 2022; Yang & Yu, 2020; Gu et al., 2018; Yang & Li, 2023; Bolte et al., 2014; Boţ et al., 2016; Xu et al., 2019a; Li & Lin, 2015; Bauschke et al., 2017; 2019) have a general approach, and analyze the convergence of general proximal algorithms, for composite problems of the form $\min_{\boldsymbol{w}} R(\boldsymbol{w}) + h(\boldsymbol{w})$ where $h$ is a more general non-convex regularizer which can include the $\ell_0$ constraint combined with an additional constraint, as long as the closed form for the projection onto the mixed constraint is known (or an approximation of it in the case of (Gu et al., 2018)). However, all of these works only provide guarantees of convergence towards a critical point, or at best, a local optimum. We provide an overview of those works in Table 1. More details about algorithms with local convergence specialized to $\ell_0$ optimization can also be found in Table 1 from (Damadi & Shen, 2022).

### B.2. Global Guarantees for IHT and RSC Functions

On the other hand, in the case of restricted strongly convex (RSC) and restricted smooth (RSS) functions, existing approximate global guarantees for the IHT algorithm do not apply to problems with such combined constraints. Indeed, several works have considered global optimality guarantees for IHT in various settings: the full gradient (deterministic) setting (IHT (Jain et al., 2014)), the stochastic setting (Nguyen et al., 2017; Li et al., 2016; Shen & Li, 2017), and the zeroth-order setting (de Vazelhes et al., 2022). However, they do not address the case where the extra constraint $\Gamma$ is added to the original sparsity constraint. The works of (Foygel Barber & Ha, 2018; Liu & Foygel Barber, 2020) tackle respectively

general non-convex thresholding operators, and general non-convex constraints, in the full gradient (deterministic) setting but however they do not provide explicit convergence rates for the particular type of sets that we consider in this paper: their rates depend on some constants (the relative concavity or the local concavity constant) for which, up to our knowledge, an explicit form is still unknown for the sets we consider.

## C. Sample complexity in sub-Gaussian design

Similarly to (Jain et al., 2014), we can use (Agarwal et al., 2010), Theorem 22, to find the sample complexity needed to verify the Assumptions of Theorem 3.4, in the setting of sub-Gaussian design. Let us consider the sparse linear regression setting, which solves $\min_{\boldsymbol{w} \in \mathcal{B}_0(k)}$, with $R(\boldsymbol{w}) = \frac{1}{n}\|\boldsymbol{A}\boldsymbol{w} - \boldsymbol{y}\|^2$, and where $n$ denotes the number of samples in the dataset (i.e. number of rows of $\boldsymbol{A}$), and each row of $\boldsymbol{A}$ is a vector of dimension $d$ drawn from a sub-Gaussian distribution with covariance $\boldsymbol{\Sigma}$ with $\boldsymbol{\Sigma}_{ii} \leq 1$ for all $i$ in $\{1, ..., d\}$, and where we have for all $i$ in $\{1, ..., n\}$, $y_i = \langle \boldsymbol{A}_{i,\cdot}, \boldsymbol{w} \rangle + \xi_i$, where $\boldsymbol{A}_{i,\cdot}$ denotes the $i$-th row of $\boldsymbol{A}$, and where $\xi_i$ is some label noise, sampled from a normal distribution of standard deviation $\sigma$ (that is $\xi_i \sim \mathcal{N}(0, \sigma^2)$). As described in (Jain et al., 2014) using Theorem 22 from (Agarwal et al., 2010), $R$ is $(\nu_s, s)$-RSC and $(L_s, s)$-RSS with probability at least $1 - e^{c_0 n}$, with $\nu_s = \frac{1}{2}\sigma_{\min}(\boldsymbol{\Sigma}) - c_1 \frac{s \log d}{n}$ and $L_s = 2\sigma_{\max}(\boldsymbol{\Sigma}) + c_1 \frac{s \log d}{n}$, where $\sigma_{\min}(\boldsymbol{\Sigma})$ and $\sigma_{\max}(\boldsymbol{\Sigma})$ are the smallest and largest eigenvalues of $\boldsymbol{\Sigma}$ respectively, and $c_0$ and $c_1$ are universal constants. Let us set $s = 2k$ as required by Theorem 3.4, and take $n > 4c_1 s \log(d)/\sigma_{\min}(\boldsymbol{\Sigma})$ samples. We then obtain $\nu_s \geq \frac{1}{4}\sigma_{\min}(\boldsymbol{\Sigma})$ and $L_s \leq 2.25\sigma_{\max}(\boldsymbol{\Sigma})$, which means that $L_s/(9\nu_s) \leq \kappa(\boldsymbol{\Sigma}) := \sigma_{\max}(\boldsymbol{\Sigma})/\sigma_{\min}(\boldsymbol{\Sigma})$. Thus it is enough to choose $k = 4(9\kappa(\boldsymbol{\Sigma}))^2\bar{k} = 324\kappa(\boldsymbol{\Sigma})^2\bar{k}$ to verify the assumptions of Theorem 3.4 with high probability.

## D. Proof of Remark 2.4

Before proceeding with the proof of Remark 2.4, we recall the definition of sign-free convex sets from (Lu, 2015) and (Beck & Hallak, 2016) below. Essentially, sign-free convex sets are convex sets that are closed by swapping the sign of any coordinate.

**Definition D.1** ((Lu, 2015),(Beck & Hallak, 2016)). A convex set $\Gamma$ is *sign-free* if for all $\boldsymbol{y} \in \{-1, 1\}^d$ and for all $\boldsymbol{x} \in \Gamma$, $\boldsymbol{x} \odot \boldsymbol{y} \in \Gamma$, where $\odot$ denotes the element-wise vector multiplication (Hadamard product for vectors).

We now proceed with the proof of Remark 2.4.

*Proof of Remark 2.4.* It is easy to show that any elementwise decomposable constraint such as box constraint is support-preserving (as projection can be done component-wise, independently). Similarly, for group-wise separable constraints where the constraint on each group is $k$-support-preserving (such as the constraint for the index tracking problem in our Section H), for a $k$-sparse vector $\boldsymbol{x} \in \mathbb{R}^d$, one can project each group of coordinates independently, and each of such projection will have its support preserved (since each such group of coordinates also contains less than $k$ non-zero elements, i.e. they are $k$-sparse). Therefore, we analyze in more detail the case of sign-free convex sets. Let $\Gamma$ be a sign-free convex set, and let $\boldsymbol{x} \in \mathbb{R}^d$ be a $k$-sparse vector. Define $\boldsymbol{z} = \Pi_\Gamma(\boldsymbol{x})$ and assume that $\mathrm{supp}(\boldsymbol{z}) \not\subseteq \mathrm{supp}(\boldsymbol{x})$. This implies that there exist some non-empty set of coordinates $S \subseteq [d]$, such that for all $i \in S$: $z_i \neq 0$ and $x_i = 0$. Define $\boldsymbol{z}'$ such that

$z'_k = \begin{cases} -z_k & \text{if } k \in S \\ z_k & \text{otherwise} \end{cases}$ . Since $\Gamma$ is sign-free, $\boldsymbol{z}' \in \Gamma$. Now, define $\boldsymbol{z}''$ such that $z''_k = \begin{cases} 0 & \text{if } k \in S \\ z_k & \text{if otherwise} \end{cases}$ . Since $\Gamma$ is

convex and since $\boldsymbol{z}'' = \frac{1}{2}\boldsymbol{z}' + \frac{1}{2}\boldsymbol{z}$, we have $\boldsymbol{z}'' \in \Gamma$. Now, we have:

$$\|\boldsymbol{x} - \boldsymbol{z}''\|_2^2 = \sum_{k=1}^d (x_k - z''_k)^2 = \sum_{k \in [d]\setminus S}(x_k - z_k)^2$$

$$< \sum_{k \in [d]\setminus S}(x_k - z_k)^2 + \sum_{k \in S}(x_k - z_k)^2 = \sum_{k=1}^d(x_k - z_k)^2 = \|\boldsymbol{x} - \boldsymbol{z}\|_2^2$$

Therefore, we encounter a contradiction since we have defined $\boldsymbol{z} = \Pi_\Gamma(\boldsymbol{x})$, and therefore, our assumption $\mathrm{supp}(\boldsymbol{z}) \not\subseteq \mathrm{supp}(\boldsymbol{x})$ is wrong, which means that $\mathrm{supp}(\boldsymbol{z}) \subseteq \mathrm{supp}(\boldsymbol{x})$. $\qquad\square$

# E. Proofs of Section 3 (Deterministic Optimization)

## E.1. Proof of Lemmas 3.2 and 3.6

### E.1.1. USEFUL LEMMAS

We first recall some useful definitions and lemmas from the literature.

**Definition E.1** (Relative concavity (Liu & Foygel Barber, 2020)). The *relative concavity* coefficient $\gamma_{k,\beta}$ of a $k$-sparse projection operator $\mathcal{H}_k$, of relative sparsity $\beta := \frac{\bar{k}}{k}$ with $\bar{k} \leq k$ is defined as:

$$\gamma_{k,\beta}\left(\mathcal{H}_k\right) = \sup\left\{\frac{\langle \boldsymbol{y} - \mathcal{H}_k(\boldsymbol{z}), \boldsymbol{z} - \mathcal{H}_k(\boldsymbol{z})\rangle}{\|\boldsymbol{y} - \mathcal{H}_k(\boldsymbol{z})\|_2^2} \;\; \boldsymbol{y}, \boldsymbol{z} \in \mathbb{R}^d, \|\boldsymbol{y}\|_0 \leq \beta k, \boldsymbol{y} \neq \mathcal{H}_k(\boldsymbol{z})\right\}.$$

**Lemma E.2** (Lemma 4.1 (Liu & Foygel Barber, 2020)). *When $\mathcal{H}_k$ is the hard-thresholding operator at sparsity level $k$, we have:*

$$\gamma_{k,\beta}\left(\mathcal{H}_k\right) = \frac{\sqrt{\beta}}{2} = \frac{1}{2}\sqrt{\frac{\bar{k}}{k}}.$$

*Proof of Lemma E.2.* Proof in (Liu & Foygel Barber, 2020). □

### E.1.2. PROOF OF LEMMA 3.2

*Proof of Lemma 3.2.* We have:

$$\|\boldsymbol{w} - \bar{\boldsymbol{w}}\|^2 = \|\boldsymbol{w} - \mathcal{H}_k(\boldsymbol{w})\|^2 + \|\mathcal{H}_k(\boldsymbol{w}) - \bar{\boldsymbol{w}}\|^2 + 2\langle \boldsymbol{w} - \mathcal{H}_k(\boldsymbol{w}), \mathcal{H}_k(\boldsymbol{w}) - \bar{\boldsymbol{w}}\rangle$$

$$\overset{(a)}{\geq} \|\boldsymbol{w} - \mathcal{H}_k(\boldsymbol{w})\|^2 + \|\mathcal{H}_k(\boldsymbol{w}) - \bar{\boldsymbol{w}}\|^2 - 2\gamma_{k,\rho}\|\mathcal{H}_k(\boldsymbol{w}) - \bar{\boldsymbol{w}}\|^2$$

$$= \|\boldsymbol{w} - \mathcal{H}_k(\boldsymbol{w})\|^2 + (1 - 2\gamma_{k,\rho})\|\mathcal{H}_k(\boldsymbol{w}) - \bar{\boldsymbol{w}}\|^2$$

$$\overset{(b)}{=} \|\boldsymbol{w} - \mathcal{H}_k(\boldsymbol{w})\|^2 + \left(1 - \sqrt{\frac{\bar{k}}{k}}\right)\|\mathcal{H}_k(\boldsymbol{w}) - \bar{\boldsymbol{w}}\|^2,$$

where (a) follows from Definition E.1 and (b) follows from Lemma E.2. Therefore, rearranging, we obtain:

$$\|\mathcal{H}_k(\boldsymbol{w}) - \boldsymbol{w}\|^2 \leq \|\boldsymbol{w} - \bar{\boldsymbol{w}}\|^2 - \left(1 - \sqrt{\frac{\bar{k}}{k}}\right)\|\mathcal{H}_k(\boldsymbol{w}) - \bar{\boldsymbol{w}}\|^2.$$

The proof is completed. □

### E.1.3. PROOF OF LEMMA 3.3

*Proof of Lemma 3.3.* Let $a = \sqrt{\frac{k}{\bar{k}}}$ and $b = \frac{\rho+1}{2} \in (\rho, 1)$. Consider

$$\boldsymbol{w} = [\underbrace{1, ..., 1}_{k}, \underbrace{b, ...b}_{\bar{k}}] \in \mathbb{R}^{k+\bar{k}}, \quad \bar{\boldsymbol{w}} = [\underbrace{0, ..., 0}_{k}, \underbrace{a, ...a}_{\bar{k}}] \in \mathbb{R}^{k+\bar{k}}.$$

Then we have $\mathcal{H}_k(\boldsymbol{w}) = [\underbrace{1, ..., 1}_{k}, \underbrace{0, ...0}_{\bar{k}}]$ and

$$\|\boldsymbol{w} - \mathcal{H}_k(\boldsymbol{w})\|^2 = b^2\bar{k}, \;\; \|\boldsymbol{w} - \bar{\boldsymbol{w}}\|^2 = k + (a-b)^2\bar{k}, \;\; \|\mathcal{H}_k(\boldsymbol{w}) - \bar{\boldsymbol{w}}\|^2 = k + a^2\bar{k}.$$

It can be verified that

$$\frac{\|\boldsymbol{w} - \mathcal{H}_k(\boldsymbol{w})\|^2 - \|\boldsymbol{w} - \bar{\boldsymbol{w}}\|^2 + \|\mathcal{H}_k(\boldsymbol{w}) - \bar{\boldsymbol{w}}\|^2}{\|\mathcal{H}_k(\boldsymbol{w}) - \bar{\boldsymbol{w}}\|^2} = \frac{2ab\bar{k}}{k + a^2\bar{k}} = b\sqrt{\frac{\bar{k}}{k}} > \rho\sqrt{\frac{\bar{k}}{k}}.$$

This proves the desired inequality. □

### E.1.4. PROOF OF LEMMA 3.6

*Proof of Lemma 3.6.* Let us abbreviate $v_k := \mathcal{H}_k(w)$. It can be verified that

$$
\begin{aligned}
\|\bar{\Pi}_\Gamma^k(w) - w\|^2 &= \left\|\bar{\Pi}_\Gamma^k(w) - v_k + v_k - w\right\|^2 \\
&\overset{(a)}{=} \left\|\bar{\Pi}_\Gamma^k(w) - v_k\right\|^2 + \|v_k - w\|^2 \\
&\overset{(b)}{\leq} \|v_k - \bar{w}\|^2 - \|\bar{\Pi}_\Gamma^k(w) - \bar{w}\|^2 + \|w - \bar{w}\|^2 - \left(1 - \sqrt{\beta}\right)\|v_k - \bar{w}\|^2 \\
&= \|w - \bar{w}\|^2 - \|\bar{\Pi}_\Gamma^k(w) - \bar{w}\|^2 + \sqrt{\beta}\|v_k - \bar{w}\|^2,
\end{aligned}
$$

where (a) is due to Definition 2.3 and the definition of the two-step projection, which imply that $\bar{\Pi}_\Gamma^k(w) - v_k$ and $v_k - w$ have disjoint supporting sets, and (b) uses the three-point-lemma for projection onto a convex set $\Gamma$, as well as Lemma 3.2. The proof is completed. $\qquad\square$

## E.2. Proof of Theorems 3.4 and 3.7

### E.2.1. PROOF OF THEOREM 3.4

In this section, we present the proof of Theorem 3.4 for the convergence of Algorithm 1 without the additional constraint, which as mentioned above, is needed for the proof of Theorem 3.7, but also, as a byproduct, illustrates how the three-point lemma simplifies previous proofs of Iterative Hard-Thresholding.

*Proof of Theorem 3.4.* The $L_s$- restricted smoothness of $R$ implies that

$$
\begin{aligned}
&R(w_t) \\
&\leq R(w_{t-1}) + \langle \nabla R(w_{t-1}), w_t - w_{t-1} \rangle + \frac{L_s}{2}\|w_t - w_{t-1}\|^2 \\
&= R(w_{t-1}) + \frac{L_s}{2}\left\|w_t - w_{t-1} + \frac{1}{L_s}\nabla R(w_{t-1})\right\|^2 - \frac{1}{2L_s}\|\nabla R(w_{t-1})\|^2 \\
&\overset{(a)}{\leq} R(w_{t-1}) + \frac{L_s}{2}\left\|\bar{w} - w_{t-1} + \frac{1}{L_s}\nabla R(w_{t-1})\right\|^2 - \frac{L_s}{2}(1 - \sqrt{\beta})\|w_t - \bar{w}\|^2 \\
&\quad - \frac{1}{2L_s}\|\nabla R(w_{t-1})\|^2 \\
&= R(w_{t-1}) + \langle \nabla R(w_{t-1}), \bar{w} - w_{t-1} \rangle + \frac{L_s}{2}\|w_{t-1} - \bar{w}\|^2 - \frac{L_s}{2}(1 - \sqrt{\beta})\|w_t - \bar{w}\|^2 \\
&\overset{(b)}{\leq} R(\bar{w}) + \frac{L_s - \nu_s}{2}\|w_{t-1} - \bar{w}\|^2 - \frac{L_s}{2}(1 - \sqrt{\beta})\|w_t - \bar{w}\|^2 \\
&\leq R(\bar{w}) + \frac{L_s - \nu_s}{2}\|w_{t-1} - \bar{w}\|^2 - \frac{2L_s - \nu_s}{4}\|w_t - \bar{w}\|^2, \qquad\qquad (7)
\end{aligned}
$$

where (a) uses Lemma 3.2, (b) is due to the $\nu_s$-restricted strong-convexity of $R$, while the last step is implied by the condition on the sparsity level $k$ from the theorem ($k \geq \frac{4L_s^2}{\nu_s^2}\bar{k}$), and the definition of $\beta$ ($\beta = \sqrt{\frac{\bar{k}}{k}}$).

The update rule composed of the gradient step and the projection from Algorithm 1 can be rewritten into the following (given that the learning rate is $\eta = \frac{1}{L_s}$, and by definition of a projection):

$$
\begin{aligned}
w_t &= \arg\min_{w \text{ s.t.} \|w\|_0 \leq k} \left\|w - \left(w_{t-1} - \frac{1}{L_s}\nabla R(w_{t-1})\right)\right\|^2 \\
&= \arg\min_{w \text{ s.t.} \|w\|_0 \leq k} \frac{2}{L_s}\langle \nabla R(w_{t-1}), w - w_{t-1} \rangle + \|w - w_{t-1}\|^2 + \frac{1}{L_s^2}\|\nabla R(w_{t-1})\|^2 \\
&= \arg\min_{w \text{ s.t.} \|w\|_0 \leq k} R(w_{t-1}) + \langle \nabla R(w_{t-1}), w - w_{t-1} \rangle + \frac{L_s}{2}\|w - w_{t-1}\|^2.
\end{aligned}
$$

Therefore, by definition of an $\arg\min$, we have:

$$R(\boldsymbol{w}_{t-1}) + \langle \nabla R(\boldsymbol{w}_{t-1}), \boldsymbol{w}_t - \boldsymbol{w}_{t-1} \rangle + \frac{L_s}{2} \|\boldsymbol{w}_t - \boldsymbol{w}_{t-1}\|^2$$

$$\leq R(\boldsymbol{w}_{t-1}) + \langle \nabla R(\boldsymbol{w}_{t-1}), \boldsymbol{w}_{t-1} - \boldsymbol{w}_{t-1} \rangle + \frac{L_s}{2} \|\boldsymbol{w}_{t-1} - \boldsymbol{w}_{t-1}\|^2$$

$$= R(\boldsymbol{w}_{t-1}). \tag{8}$$

And from the $L_s$ smoothness of $R$, we also have:

$$R(\boldsymbol{w}_t) \leq R(\boldsymbol{w}_{t-1}) + \langle \nabla R(\boldsymbol{w}_{t-1}), \boldsymbol{w}_t - \boldsymbol{w}_{t-1} \rangle + \frac{L_s}{2} \|\boldsymbol{w}_t - \boldsymbol{w}_{t-1}\|^2. \tag{9}$$

Therefore, combining equations 8 and 9, we obtain:

$$R(\boldsymbol{w}_t) \leq R(\boldsymbol{w}_{t-1}).$$

That is, the sequence $\{R(\boldsymbol{w}_t)\}_{t \geq 0}$ of risk is non-increasing.

Let us now consider

$$T := \left\lceil \frac{2L_s}{\nu_s} \log\left( \frac{(L_s - \nu_s)\|\boldsymbol{w}_0 - \bar{\boldsymbol{w}}\|^2}{2\varepsilon} \right) \right\rceil.$$

We claim that $R(\boldsymbol{w}_t) \leq R(\bar{\boldsymbol{w}}) + \varepsilon$ for $t \geq T + 1$. To show this, suppose that $\exists t \in [T]$ such that $R(\boldsymbol{w}_t) \leq R(\bar{\boldsymbol{w}}) + \varepsilon$. Then the claim is naturally true by monotonicity. Otherwise assume that $R(\boldsymbol{w}_t) > R(\bar{\boldsymbol{w}}) + \varepsilon$ for all $t \in [T]$. Then in view of the inequality equation 7 we know that

$$\|\boldsymbol{w}_T - \bar{\boldsymbol{w}}\|^2 \leq \frac{2L_s - 2\nu_s}{2L_s - \nu_s} \|\boldsymbol{w}_{T-1} - \bar{\boldsymbol{w}}\|^2$$

$$\leq \left(1 - \frac{\nu_s}{2L_s}\right) \|\boldsymbol{w}_{T-1} - \bar{\boldsymbol{w}}\|^2$$

$$\leq \left(1 - \frac{\nu_s}{2L_s}\right)^T \|\boldsymbol{w}_0 - \bar{\boldsymbol{w}}\|^2$$

$$= \exp\left( T \log\left(1 - \frac{\nu_s}{2L_s}\right) \right) \|\boldsymbol{w}_0 - \bar{\boldsymbol{w}}\|^2$$

$$\leq \exp\left( \frac{2L_s}{\nu_s} \log\left( \frac{(L_s - \nu_s)\|\boldsymbol{w}_0 - \bar{\boldsymbol{w}}\|^2}{2\varepsilon} + 1 \right) \log\left(1 - \frac{\nu_s}{2L_s}\right) \right) \|\boldsymbol{w}_0 - \bar{\boldsymbol{w}}\|^2$$

$$= \left(1 - \frac{\nu_s}{2L_s}\right) \exp\left( \frac{2L_s}{\nu_s} \log\left( \frac{(L_s - \nu_s)\|\boldsymbol{w}_0 - \bar{\boldsymbol{w}}\|^2}{2\varepsilon} \right) \log\left(1 - \frac{\nu_s}{2L_s}\right) \right) \|\boldsymbol{w}_0 - \bar{\boldsymbol{w}}\|^2$$

$$\overset{(a)}{\leq} \left(1 - \frac{\nu_s}{2L_s}\right) \exp\left( \frac{2L_s}{\nu_s} \log\left( \frac{2\varepsilon}{(L_s - \nu_s)\|\boldsymbol{w}_0 - \bar{\boldsymbol{w}}\|^2} \right) \frac{\nu_s}{2L_s} \right) \|\boldsymbol{w}_0 - \bar{\boldsymbol{w}}\|^2$$

$$= \left(1 - \frac{\nu_s}{2L_s}\right) \frac{2\varepsilon}{L_s - \nu_s} \overset{(b)}{\leq} \frac{2\varepsilon}{L_s - \nu_s},$$

where (a) follows from the fact that for all $x$ in $(-\infty, 1)$: $\log(1 - x) \leq -x$, and (b) uses the fact that $\left(1 - \frac{\nu_s}{2L_s}\right) \leq 1$.

Then according to equation 7 we must have

$$R(\boldsymbol{w}_{T+1}) \leq R(\bar{\boldsymbol{w}}) + \frac{L_s - \nu_s}{2} \|\boldsymbol{w}_T - \bar{\boldsymbol{w}}\|^2 \leq R(\bar{\boldsymbol{w}}) + \varepsilon,$$

which implies the desired claim. The proof is completed. $\qquad\square$

*Remark* E.3. Theorem 3.4 recovers the result of Jain et al. (2014, Theorem 1). Our proof is shorter yet more intuitive than in that paper.

E.2.2. PROOF OF THEOREM 3.7

Using the above results, we can now proceed to the full proof of convergence of Theorem 3.7 below.

*Proof of Theorem 3.7.* Denote $\boldsymbol{v}_t = \mathcal{H}_k(\boldsymbol{w}_{t-1} - \frac{1}{L_s}\nabla R(\boldsymbol{w}_{t-1}))$ for any $t \in \mathbb{N}$. Similar to the arguments for equation 7, based on the $L_s$-restricted smoothness of $R$ we can show that:

$$R(\boldsymbol{w}_t)$$
$$\leq R(\boldsymbol{w}_{t-1}) + \langle \nabla R(\boldsymbol{w}_{t-1}), \boldsymbol{w}_t - \boldsymbol{w}_{t-1}\rangle + \frac{L_s}{2}\|\boldsymbol{w}_t - \boldsymbol{w}_{t-1}\|^2$$
$$= R(\boldsymbol{w}_{t-1}) + \frac{L_s}{2}\left\|\boldsymbol{w}_t - \boldsymbol{w}_{t-1} + \frac{1}{L_s}\nabla R(\boldsymbol{w}_{t-1})\right\|^2 - \frac{1}{2L_s}\|\nabla R(\boldsymbol{w}_{t-1})\|^2$$
$$\overset{(a)}{\leq} R(\boldsymbol{w}_{t-1}) + \frac{L_s}{2}\left\|\bar{\boldsymbol{w}} - \boldsymbol{w}_{t-1} + \frac{1}{L_s}\nabla R(\boldsymbol{w}_{t-1})\right\|^2 - \frac{L_s}{2}\|\boldsymbol{w}_t - \bar{\boldsymbol{w}}\|^2$$
$$\quad + \frac{L_s}{2}\sqrt{\beta}\|\boldsymbol{v}_t - \bar{\boldsymbol{w}}\|^2 - \frac{1}{2L_s}\|\nabla R(\boldsymbol{w}_{t-1})\|^2$$
$$= R(\boldsymbol{w}_{t-1}) + \langle \nabla R(\boldsymbol{w}_{t-1}), \bar{\boldsymbol{w}} - \boldsymbol{w}_{t-1}\rangle + \frac{L_s}{2}\|\boldsymbol{w}_{t-1} - \bar{\boldsymbol{w}}\|^2 - \frac{L_s}{2}\|\boldsymbol{w}_t - \bar{\boldsymbol{w}}\|^2$$
$$\quad + \frac{L_s}{2}\sqrt{\beta}\|\boldsymbol{v}_t - \bar{\boldsymbol{w}}\|^2$$
$$\overset{(b)}{\leq} R(\bar{\boldsymbol{w}}) + \frac{L_s - \nu_s}{2}\|\boldsymbol{w}_{t-1} - \bar{\boldsymbol{w}}\|^2 - \frac{L_s}{2}\|\boldsymbol{w}_t - \bar{\boldsymbol{w}}\|^2 + \frac{L_s}{2}\sqrt{\beta}\|\boldsymbol{v}_t - \bar{\boldsymbol{w}}\|^2$$
$$\leq R(\bar{\boldsymbol{w}}) + \frac{L_s - \nu_s}{2}\|\boldsymbol{w}_{t-1} - \bar{\boldsymbol{w}}\|^2 - \frac{L_s}{2}\|\boldsymbol{w}_t - \bar{\boldsymbol{w}}\|^2 + \frac{\rho\nu_s}{4(1-\rho)}\|\boldsymbol{v}_t - \bar{\boldsymbol{w}}\|^2, \tag{10}$$

where (a) uses Lemma 3.2, (b) is due to the $\nu_s$-restricted strong-convexity of $R$, and the last step is due to the condition on sparsity level $k$ from the theorem ($k \geq \frac{4L_s^2(1-\rho)^2}{\nu_s^2\rho^2}\bar{k}$), and the definition of $\beta = \sqrt{\frac{\bar{k}}{k}}$.

In view of equation 7, which is valid under the given conditions, we know that

$$R(\boldsymbol{v}_t) \leq R(\bar{\boldsymbol{w}}) + \frac{L_s - \nu_s}{2}\|\boldsymbol{w}_{t-1} - \bar{\boldsymbol{w}}\|^2 - \frac{2L_s - \nu_s}{4}\|\boldsymbol{v}_t - \bar{\boldsymbol{w}}\|^2. \tag{11}$$

After proper scaling and summing both sides of equation 10 and equation 11 yields that

$$(1-\rho)R(\boldsymbol{w}_t) + \rho R(\boldsymbol{v}_t)$$
$$\leq R(\bar{\boldsymbol{w}}) + \frac{L_s - \nu_s}{2}\|\boldsymbol{w}_{t-1} - \bar{\boldsymbol{w}}\|^2 - \frac{(1-\rho)L_s}{2}\|\boldsymbol{w}_t - \bar{\boldsymbol{w}}\|^2 - \frac{\rho(L_s - \nu_s)}{2}\|\boldsymbol{v}_t - \bar{\boldsymbol{w}}\|^2$$
$$= R(\bar{\boldsymbol{w}}) + \frac{L_s - \nu_s}{2}\|\boldsymbol{w}_{t-1} - \bar{\boldsymbol{w}}\|^2 - \frac{L_s - \rho\nu_s}{2}\|\boldsymbol{w}_t - \bar{\boldsymbol{w}}\|^2, \tag{12}$$

where in the second inequality we have used $\bar{\boldsymbol{w}} \in \Gamma$ and the non-expansiveness of projection over convex sets.

Let us now consider

$$T := \left\lceil \frac{2L_s}{\nu_s}\log\left(\frac{(L_s - \nu_s)\|\boldsymbol{w}_0 - \bar{\boldsymbol{w}}\|^2}{2\varepsilon}\right)\right\rceil. \tag{13}$$

We claim that:

$$\min_{t \in [T+1]}\{(1-\rho)R(\boldsymbol{w}_t) + \rho R(\boldsymbol{v}_t)\} \leq R(\bar{\boldsymbol{w}}) + \varepsilon. \tag{14}$$

To show this, suppose that $\exists t \in [T]$ such that $(1-\rho)R(\boldsymbol{w}_t) + \rho R(\boldsymbol{v}_t) \leq R(\bar{\boldsymbol{w}}) + \varepsilon$. Then the claim is naturally true. Otherwise assume that $(1-\rho)R(\boldsymbol{w}_t) + \rho R(\boldsymbol{v}_t) > R(\bar{\boldsymbol{w}}) + \varepsilon$ for all $t \in [T]$. Then in view of the inequality equation 12 we know that

$$\|\boldsymbol{w}_T - \bar{\boldsymbol{w}}\|^2 \le \frac{L_s - \nu_s}{L_s - \rho\nu_s}\|\boldsymbol{w}_{T-1} - \bar{\boldsymbol{w}}\|^2 \le \left(1 - \frac{(1-\rho)\nu_s}{L_s}\right)\|\boldsymbol{w}_{T-1} - \bar{\boldsymbol{w}}\|^2$$
$$\le \left(1 - \frac{(1-\rho)\nu_s}{L_s}\right)^T \|\boldsymbol{w}_0 - \bar{\boldsymbol{w}}\|^2 \le \frac{2\varepsilon}{L_s - \nu_s}.$$

Then according to equation 12 we must have

$$(1-\rho)R(\boldsymbol{w}_{T+1}) + \rho R(\boldsymbol{v}_{T+1}) \le R(\bar{\boldsymbol{w}}) + \frac{L_s - \nu_s}{2}\|\boldsymbol{w}_T - \bar{\boldsymbol{w}}\|^2 \le R(\bar{\boldsymbol{w}}) + \varepsilon, \tag{15}$$

which proves the claim from equation 14. Now, recall that we have assumed in the Assumptions of Theorem 3.7, without loss of generality, that $R$ is non-negative (if not, we can redefine $R$ by adding a constant, without modifying the gradient of $R$, keeping the algorithm untouched), which implies that $R(\boldsymbol{v}_t) \ge 0$. Plugging this in equation 14, for $T \ge \left\lceil \frac{2L_s}{\nu_s}\log\left(\frac{(L_s-\nu_s)\|\boldsymbol{w}_0-\bar{\boldsymbol{w}}\|^2}{2\varepsilon'(1-\rho)}\right)\right\rceil + 1$ implies that:

$$\min_{t\in[T]} R(\boldsymbol{w}_t) \le \frac{1}{1-\rho}R(\bar{\boldsymbol{w}}) + \frac{\varepsilon}{1-\rho} \le (1+2\rho)R(\bar{\boldsymbol{w}}) + \frac{\varepsilon}{1-\rho}. \tag{16}$$

Plugging the change of variable $\varepsilon' = \frac{\varepsilon}{1-\rho}$ into equation 16 above, and in 13, we obtain that when $T \ge \left\lceil \frac{2L_s}{\nu_s}\log\left(\frac{(L_s-\nu_s)\|\boldsymbol{w}_0-\bar{\boldsymbol{w}}\|^2}{2\varepsilon'(1-\rho)}\right)\right\rceil + 1$:

$$\min_{t\in[T]} R(\boldsymbol{w}_t) \le (1+2\rho)R(\bar{\boldsymbol{w}}) + \varepsilon'.$$

Further, consider an ideal case where $\bar{\boldsymbol{w}}$ is a global minimizer of $R$ over $\mathcal{B}_0(k) := \{\boldsymbol{w} : \|\boldsymbol{w}\|_0 \le k\}$. Then $R(\boldsymbol{v}_t) \ge R(\bar{\boldsymbol{w}})$ is always true for all $t \ge 1$. It follows that the bound in equation 14 yields, for $T \ge \left\lceil \frac{2L_s}{\nu_s}\log\left(\frac{(L_s-\nu_s)\|\boldsymbol{w}_0-\bar{\boldsymbol{w}}\|^2}{2\varepsilon}\right)\right\rceil + 1$:

$$\min_{t\in[T]} \{(1-\rho)R(\boldsymbol{w}_t) + \rho R(\bar{\boldsymbol{w}})\} \le \min_{t\in[T]} \{(1-\rho)R(\boldsymbol{w}_t) + \rho R(\boldsymbol{v}_t)\} \le R(\bar{\boldsymbol{w}}) + \varepsilon,$$

which implies: $\min_{t\in[T]} R(\boldsymbol{w}_t) \le R(\bar{\boldsymbol{w}}) + \frac{\varepsilon}{1-\rho}$. In this case, we can simply set $\rho = 0.5$, and define $\varepsilon' = \frac{\varepsilon}{1-\rho} = 2\varepsilon$ similarly as above. This implies the desired claims. The proof is completed.

$\square$

### E.3. Lower Bound on the Sparsity Relaxation

Consider $\kappa > 1$, $p = \bar{k} + \kappa^2\bar{k}$ and the following defined diagonal matrix $A$ of size $p \times p$ and vector $\boldsymbol{b}$ of size $p$:

$$\boldsymbol{A} = \begin{bmatrix} \kappa & 0 & \cdots & 0 \\ 0 & 1 & \cdots & 0 \\ \vdots & & \ddots & 0 \\ 0 & 0 & \cdots & 1 \end{bmatrix} \in \mathbb{R}^{p\times p}, \quad \boldsymbol{b} = [\underbrace{1, \kappa, \ldots, \kappa}_{\bar{k}}, \underbrace{1, \ldots, 1}_{\kappa^2\bar{k}}]^\top \in \mathbb{R}^p.$$

Clearly, $\boldsymbol{A}$ is $\kappa$-smooth and 1-strongly convex. Let us consider the following quadratic objective function:

$$f(\boldsymbol{w}) = \frac{1}{2}(\boldsymbol{w} - \boldsymbol{b})^\top \boldsymbol{A}(\boldsymbol{w} - \boldsymbol{b}).$$

Let $k \in [\bar{k}, \kappa^2\bar{k}]$ be the relaxed sparsity level used for IHT, and being an even number (without loss of generality). Consider the following defined $p$-dimensional sparse vectors such that $\|\bar{x}\|_0 = \bar{k}$ and $\|x\|_0 = k$:

$$\bar{\boldsymbol{w}} = [\underbrace{1, \kappa, \ldots, \kappa}_{\bar{k}}, \underbrace{0, \ldots, 0}_{\kappa^2\bar{k}}]^\top \in \mathbb{R}^p, \quad \boldsymbol{w} = [\underbrace{0, \ldots, 0}_{\bar{k}/2}, \underbrace{\kappa, \ldots, \kappa}_{\bar{k}/2}, \underbrace{1, \ldots, 1}_{k-\bar{k}/2}, \underbrace{0, \ldots, 0}_{\kappa^2\bar{k}-k+\bar{k}/2}]^\top \in \mathbb{R}^p.$$

We next prove the following theorem which shows that $k \ge \mathcal{O}(\kappa^2\bar{k})$ is indeed necessary for IHT to converge in some extreme cases for optimizing $f$.

**Theorem E.4.** *If $\bar{k} \geq 4$ and $k \leq \frac{\kappa^2 \bar{k}}{8}$, then it holds that*

$$f(\boldsymbol{w}) \geq f(\bar{\boldsymbol{w}}) + \frac{\kappa^2 \bar{k}}{16},$$

*while $\boldsymbol{w}$ is a fixed point of IHT with sparsity level $k$ and step-size $\eta = \frac{1}{2\kappa}$, i.e.,*

$$\boldsymbol{w} = \mathcal{H}_k \left( \boldsymbol{w} - \eta \nabla f(\boldsymbol{w}) \right).$$

*Proof.* It can be seen that $f(\bar{\boldsymbol{w}}) = \frac{1}{2}\kappa^2 \bar{k}$ and

$$f(\boldsymbol{w}) = \frac{1}{2} \left( \kappa + \left( \frac{\bar{k}}{2} - 1 \right) \kappa^2 + \kappa^2 \bar{k} + \frac{\bar{k}}{2} - k \right).$$

Therefore

$$
\begin{aligned}
f(\boldsymbol{w}) - f(\bar{\boldsymbol{w}}) &= \frac{1}{2} \left( \kappa + \left( \frac{\bar{k}}{2} - 1 \right) \kappa^2 + \frac{\bar{k}}{2} - k \right) \\
&\geq \frac{1}{2} \left( \left( \frac{\bar{k}}{2} - 1 \right) \kappa^2 - k \right) \\
&\overset{\zeta_1}{\geq} \frac{1}{2} \left( \frac{\bar{k}}{4} \kappa^2 - k \right) \geq \frac{\kappa^2 \bar{k}}{16},
\end{aligned}
$$

where $\zeta_1$ uses $\bar{k} \geq 4$, and the last inequality is due to $k \leq \frac{\kappa^2 \bar{k}}{8}$. Note that

$$\nabla f(\boldsymbol{w}) = \boldsymbol{A}(\boldsymbol{w} - \boldsymbol{b}) = [\underbrace{-\kappa, \ldots, -\kappa}_{\bar{k}/2}, \underbrace{0, \ldots, 0}_{k}, \underbrace{-1, \ldots, -1}_{\kappa^2 \bar{k} - k + \bar{k}/2}]^\top.$$

Given $\eta = \frac{1}{2\kappa}$, we can show that

$$\boldsymbol{w} - \eta \nabla f(\boldsymbol{w}) = [\underbrace{0.5, \ldots, 0.5}_{\bar{k}/2}, \underbrace{\kappa, \ldots, \kappa}_{\bar{k}/2}, \underbrace{1, \ldots, 1}_{k - \bar{k}/2}, \underbrace{0.5/\kappa, \ldots, 0.5/\kappa}_{\kappa^2 \bar{k} - k + \bar{k}/2}]^\top,$$

which directly yields (as $\kappa > 1$)

$$\boldsymbol{w} = \mathcal{H}_k(\boldsymbol{w} - \eta \nabla f(\boldsymbol{w})),$$

and thus $\boldsymbol{w}$ is a fixed point of IHT with sparsity level $k$ and step-size $\eta = \frac{1}{2\kappa}$. $\qquad \square$

*Remark* E.5. The example is inspired by the one from (Axiotis & Sviridenko, 2022), though slightly simpler. A main difference is that in our example the supporting sets of $\boldsymbol{w}$ and $\bar{\boldsymbol{w}}$ are allowed to be significantly overlapped, while in theirs the supporting sets of the two vectors are constructed to be disjoint.

# F. Proofs of Section 4 (Stochastic and Zeroth-Order Optimization)

## F.1. Discussion on Restricted Smoothness Assumptions

In this section, we provide additional details on the difference between Assumptions 2.2 and 4.5. First, we recall the standard definition of smoothness:

**Definition F.1.** A differentiable function $f$ is $L$-smooth if for all $\boldsymbol{x}, \boldsymbol{y} \in (\mathbb{R}^d)^2$:

$$\|\nabla f(\boldsymbol{x}) - \nabla f(\boldsymbol{y})\| \leq L \|\boldsymbol{x} - \boldsymbol{y}\|$$

We now provide the counter-example below, illustrating that Assumptions 2.2 and 4.5 are not always equivalent, even if $f$ is convex (and that those two assumptions are also different from the usual smoothness assumption).

**Lemma F.2.** *Let us consider the following convex function $f : \mathbb{R}^2 \to \mathbb{R}$ defined as*

$$\forall (x_1, x_2) \in \mathbb{R}^2 : f(x_1, x_2) = x_1^2 + x_2^2 + x_1 x_2$$

*$f$ has the following regularity properties, with the given constants being each time the smallest possible:*

- *(i) 3-smooth*

- *(ii) 2-restricted smooth (Assumption 2.2) with sparsity level 1*

- *(iii) $\sqrt{5}$-restricted strongly smooth (Assumption 4.5) with sparsity level 1*

*Proof.* F.1.1. PROOF OF (I)

The Hessian of $f$ is:

$$\boldsymbol{H} = \begin{bmatrix} 2 & 1 \\ 1 & 2 \end{bmatrix},$$

and its diagonalization is:

$$\boldsymbol{H} = \boldsymbol{P}\boldsymbol{D}\boldsymbol{P}^{-1},$$

with:

$$\boldsymbol{P} = \begin{bmatrix} 1 & -1 \\ 1 & 1 \end{bmatrix}, \boldsymbol{P}^{-1} = \frac{1}{2}\begin{bmatrix} 1 & 1 \\ -1 & 1 \end{bmatrix}, \text{ and } \boldsymbol{D} = \begin{bmatrix} 3 & 0 \\ 0 & 1 \end{bmatrix}.$$

Therefore, the smallest $L$ such that we have $\boldsymbol{H} \preceq L\boldsymbol{I}_{2\times2}$ is 3, which implies from Lemma 1.2.2 in (Nesterov et al., 2018) that $f$ is smooth with smoothness constant 3.

F.1.2. PROOF OF (II):

Let us take two $\boldsymbol{x}, \boldsymbol{y}$ in $(\mathbb{R}^d)^2$ such that $\|\boldsymbol{x} - \boldsymbol{y}\|_0 \leq 1$, which therefore implies that: $x_1 = y_1$ or $x_2 = y_2$ (or both). Let us suppose that (E): $x_2 = y_2$. Note that this implies that $\|\boldsymbol{x} - \boldsymbol{y}\|_2 = (x_1 - y_1)^2$. We now need to find the smallest $L$ such that:

$$f(\boldsymbol{y}) \leq f(\boldsymbol{x}) + \langle \nabla f(\boldsymbol{x}), \boldsymbol{y} - \boldsymbol{x} \rangle + \frac{L}{2}\|\boldsymbol{x} - \boldsymbol{y}\|_2^2$$

$$\Leftrightarrow$$

$$y_1^2 + y_2^2 + y_1 y_2 \leq x_1^2 + x_2^2 + x_1 x_2 + (2x_1 + x_2)(y_1 - x_1) + (2x_2 + x_1)(y_2 - x_2) + \frac{L}{2}(x_1 - y_1)^2$$

$$\overset{(E)}{\Leftrightarrow}$$

$$y_1^2 + x_2^2 + y_1 x_2 \leq x_1^2 + x_2^2 + x_1 x_2 + (2x_1 + x_2)(y_1 - x_1) + (2x_2 + x_1)(x_2 - x_2) + \frac{L}{2}(x_1 - y_1)^2$$

$$\Leftrightarrow$$

$$y_1^2 + x_1^2 - 2y_1 x_1 \leq \frac{L}{2}(x_1 - y_1)^2$$

$$\Leftrightarrow$$

$$(x_1 - y_1)^2 \leq \frac{L}{2}(x_1 - y_1)^2$$

Therefore, the smallest $L$ possible which can verify the above is $L = 2$. By symmetry, we would have the same chain of equivalence in the alternative case where we would replace $x_2 = y_2$ by $x_1 = y_1$. Therefore, we need some $L$ that will work for both cases, so again, such smallest $L$ is 2.

F.1.3. PROOF OF (III)

Let us take two $\boldsymbol{x}, \boldsymbol{y}$ such that $\|\boldsymbol{x} - \boldsymbol{y}\|_0 \leq 1$, which therefore implies that: $x_1 = y_1$ or $x_2 = y_2$ (or both). Let us suppose that (E): $x_2 = y_2$. Note that this means that $\|\boldsymbol{x} - \boldsymbol{y}\|_2 = (x_1 - y_1)^2$. What we need to find is the smallest $L$ such that:

$$\|\nabla f(\boldsymbol{x}) - \nabla f(\boldsymbol{y})\|^2 \leq L^2\|\boldsymbol{x} - \boldsymbol{y}\|_2^2$$

$$\Leftrightarrow$$

$$(2x_1 + x_2 - (2y_1 + y_2))^2 + (2x_2 + x_1 - (2y_2 + y_1))^2 \leq L^2(x_1 - y_1)^2$$

$$\overset{(E)}{\Leftrightarrow}$$

$$(2x_1 + x_2 - (2y_1 + x_2))^2 + (2x_2 + x_1 - (2x_2 + y_1))^2 \leq L^2(x_1 - y_1)^2$$

$$\Leftrightarrow$$

$$4(x_1 - y_1)^2 + (x_1 - y_1)^2 \leq L^2(x_1 - y_1)^2$$

$$\Leftrightarrow$$

$$5(x_1 - y_1)^2 \leq L^2(x_1 - y_1)^2$$

Therefore, the smallest $L$ possible which can verify the above is $L = \sqrt{5}$. By symmetry, we would have the same chain of equivalence in the alternative case where we would replace $x_2 = y_2$ by $x_1 = y_1$. So therefore we need some $L$ that will work for both cases, so again, that smallest $L$ is $\sqrt{5}$.

$\square$

## F.2. Proof of Theorems 4.2 and 4.3

For the proof of Theorem 4.3, we use a similar technique as in Theorem 3.7 to deal with the extra constraint, i.e. we start first from the case $\Gamma = \mathbb{R}^d$ (Theorem 4.2). Based on our $\ell_0$ three-point lemma (Lemma 3.2), such proof of Theorem 4.2 is simpler than the corresponding proof of (Zhou et al., 2018) (Proof of Theorem 2, Appendix B.3). Also, compared to the deterministic setting, here, we need to carefully incorporate the exponentially decreasing error of the gradient estimator into a properly weighted telescopic sum containing terms in $\|\boldsymbol{w}_t - \bar{\boldsymbol{w}}\|^2$. Below we provide several intermediary results needed for the proof of Theorem 4.3. Then, the proof of Theorem 4.3 will be provided in Section F.2.3.

### F.2.1. USEFUL LEMMA

Before starting the proof, we present the following lemma from (Mishchenko et al., 2020), which relates the batch-size $s_t$ and the error of the gradient estimator:

**Lemma F.3** ((Mishchenko et al., 2020), Lemma 1)**.** *Let $\boldsymbol{w}_t \in \mathbb{R}^d$. Assume that $\boldsymbol{g}_t$ is the sampled gradient in Algorithm 2 and that the population variance of $R_i(\boldsymbol{w}_t)$ is bounded by $B$ as in Assumption 4.1. Then the gradient estimate $\boldsymbol{g}_t$ is an unbiased estimate of $\nabla R(\boldsymbol{w}_t)$, and its variance is as follows:*

$$\mathbb{E}\|\boldsymbol{g}_t - \nabla R(\boldsymbol{w}_t)\|^2 \leq \frac{n - s_t}{n - 1} \frac{1}{s_t} B, \tag{17}$$

Note that the original Lemma from (Mishchenko et al., 2020) is written as an equality, in terms of the exact population variance of a random variable, denoted $\sigma^2$, but we rewrite it as an inequality here for simplicity, in order to have a general bound that applies at each iteration.

*Proof of Lemma F.3.* Proof in (Mishchenko et al., 2020). $\square$

### F.2.2. PROOF OF THEOREM 4.2

Below we now first present a proof for the convergence of Algorithm 2 without the additional constraint (Theorem 4.2), which is needed for the proof of Theorem 4.3, and also, as a byproduct, illustrates how the three-point lemma simplifies such proof.

*Proof of Theorem 4.2.* The $L_s$-smoothness of $R$ implies that

$$R(\boldsymbol{w}_t)$$

$$\leq R(\boldsymbol{w}_{t-1}) + \langle \nabla R(\boldsymbol{w}_{t-1}), \boldsymbol{w}_t - \boldsymbol{w}_{t-1} \rangle + \frac{L_s}{2} \|\boldsymbol{w}_t - \boldsymbol{w}_{t-1}\|^2$$

$$= R(\boldsymbol{w}_{t-1}) + \langle \boldsymbol{g}_{t-1}, \boldsymbol{w}_t - \boldsymbol{w}_{t-1} \rangle + \frac{L_s}{2} \|\boldsymbol{w}_t - \boldsymbol{w}_{t-1}\|^2 + \langle \nabla R(\boldsymbol{w}_{t-1}) - \boldsymbol{g}_{t-1}, \boldsymbol{w}_t - \boldsymbol{w}_{t-1} \rangle$$

$$= R(\boldsymbol{w}_{t-1}) + \frac{1}{2\eta} \left[ \|\boldsymbol{w}_t - (\boldsymbol{w}_{t-1} - \eta \boldsymbol{g}_{t-1})\|^2 - \eta^2 \|\boldsymbol{g}_{t-1}\|^2 - \|\boldsymbol{w}_t - \boldsymbol{w}_{t-1}\|^2 \right] + \frac{L_s}{2} \|\boldsymbol{w}_t - \boldsymbol{w}_{t-1}\|^2$$

$$\quad + \langle \nabla R(\boldsymbol{w}_{t-1}) - \boldsymbol{g}_{t-1}, \boldsymbol{w}_t - \boldsymbol{w}_{t-1} \rangle$$

$$= R(\boldsymbol{w}_{t-1}) + \frac{1}{2\eta} \|\boldsymbol{w}_t - (\boldsymbol{w}_{t-1} - \eta \boldsymbol{g}_{t-1})\|^2 - \frac{\eta}{2} \|\boldsymbol{g}_{t-1}\|^2 + \left[ \frac{L_s - \frac{1}{\eta}}{2} \right] \|\boldsymbol{w}_t - \boldsymbol{w}_{t-1}\|^2$$

$$\quad + \langle \nabla R(\boldsymbol{w}_{t-1}) - \boldsymbol{g}_{t-1}, \boldsymbol{w}_t - \boldsymbol{w}_{t-1} \rangle$$

$$\overset{(a)}{\leq} R(\boldsymbol{w}_{t-1}) + \frac{1}{2\eta} \left[ \|\bar{\boldsymbol{w}} - (\boldsymbol{w}_{t-1} - \eta \boldsymbol{g}_{t-1})\|^2 - (1 - \sqrt{\beta}) \|\boldsymbol{w}_t - \bar{\boldsymbol{w}}\|^2 \right] - \frac{\eta}{2} \|\boldsymbol{g}_{t-1}\|^2$$

$$\quad + \left[ \frac{L_s - \frac{1}{\eta}}{2} \right] \|\boldsymbol{w}_t - \boldsymbol{w}_{t-1}\|^2 + \langle \nabla R(\boldsymbol{w}_{t-1}) - \boldsymbol{g}_{t-1}, \boldsymbol{w}_t - \boldsymbol{w}_{t-1} \rangle$$

$$= R(\boldsymbol{w}_{t-1}) + \frac{1}{2\eta} \left[ \|\bar{\boldsymbol{w}} - \boldsymbol{w}_{t-1}\|^2 + \eta^2 \|\boldsymbol{g}_{t-1}\|^2 - 2\langle \eta \boldsymbol{g}_{t-1}, \boldsymbol{w}_{t-1} - \bar{\boldsymbol{w}} \rangle \right] - \frac{1}{2\eta} (1 - \sqrt{\beta}) \|\boldsymbol{w}_t - \bar{\boldsymbol{w}}\|^2$$

$$\quad - \frac{\eta}{2} \|\boldsymbol{g}_{t-1}\|^2 + \left[ \frac{L_s - \frac{1}{\eta}}{2} \right] \|\boldsymbol{w}_t - \boldsymbol{w}_{t-1}\|^2 + \langle \nabla R(\boldsymbol{w}_{t-1}) - \boldsymbol{g}_{t-1}, \boldsymbol{w}_t - \boldsymbol{w}_{t-1} \rangle$$

$$= R(\boldsymbol{w}_{t-1}) + \frac{1}{2\eta} \left[ \|\bar{\boldsymbol{w}} - \boldsymbol{w}_{t-1}\|^2 - 2\langle \eta \boldsymbol{g}_{t-1}, \boldsymbol{w}_{t-1} - \bar{\boldsymbol{w}} \rangle \right] - \frac{1}{2\eta} (1 - \sqrt{\beta}) \|\boldsymbol{w}_t - \bar{\boldsymbol{w}}\|^2$$

$$\quad + \left[ \frac{L_s - \frac{1}{\eta}}{2} \right] \|\boldsymbol{w}_t - \boldsymbol{w}_{t-1}\|^2 + \langle \nabla R(\boldsymbol{w}_{t-1}) - \boldsymbol{g}_{t-1}, \boldsymbol{w}_t - \boldsymbol{w}_{t-1} \rangle$$

$$\overset{(b)}{=} R(\boldsymbol{w}_{t-1}) + \frac{1}{2\eta} \|\bar{\boldsymbol{w}} - \boldsymbol{w}_{t-1}\|^2 - \langle \boldsymbol{g}_{t-1}, \boldsymbol{w}_{t-1} - \bar{\boldsymbol{w}} \rangle - \frac{1}{2\eta} (1 - \sqrt{\beta}) \|\boldsymbol{w}_t - \bar{\boldsymbol{w}}\|^2$$

$$\quad + \left[ \frac{L_s - \frac{1}{\eta} + C}{2} \right] \|\boldsymbol{w}_t - \boldsymbol{w}_{t-1}\|^2 + \frac{1}{2C} \|\nabla R(\boldsymbol{w}_{t-1}) - \boldsymbol{g}_{t-1}\|^2,$$

where (a) follows from Lemma 3.2 and (b) follows from the inequality $\langle a, b \rangle \leq \frac{C}{2} a^2 + \frac{1}{2C} b^2$, for any $(a, b) \in (\mathbb{R}^d)^2$ with $C > 0$ an arbitrary strictly positive constant.

Let us now assume that $\eta = \frac{1}{L_s + C}$: therefore the term $\left[ \frac{L_s - \frac{1}{\eta} + C}{2} \right] \|\boldsymbol{w}_t - \boldsymbol{w}_{t-1}\|^2$ above is 0. We now take the conditional expectation (conditioned on $\mathrm{w}_{t-1}$, which is the random variable which realizations are $\boldsymbol{w}_{t-1}$), on both sides, and from Lemma F.3 we obtain the inequality below (we slightly abuse notations and denote $\mathbb{E}[\cdot | \mathrm{w}_{t-1} = \boldsymbol{w}_{t-1}]$ by $\mathbb{E}[\cdot | \boldsymbol{w}_{t-1}]$):

$$\mathbb{E}[R(\boldsymbol{w}_t) | \boldsymbol{w}_{t-1}] \leq R(\boldsymbol{w}_{t-1}) + \frac{1}{2\eta} \|\bar{\boldsymbol{w}} - \boldsymbol{w}_{t-1}\|^2 - \langle \nabla R(\boldsymbol{w}_{t-1}), \boldsymbol{w}_{t-1} - \bar{\boldsymbol{w}} \rangle$$

$$- \frac{1}{2\eta} (1 - \sqrt{\beta}) \mathbb{E} \left[ \|\boldsymbol{w}_t - \bar{\boldsymbol{w}}\|^2 | \boldsymbol{w}_{t-1} \right] + \frac{B(n - s_{t-1})}{2C s_{t-1}(n - 1)}$$

$$\overset{(a)}{\leq} R(\boldsymbol{w}_{t-1}) + \frac{1}{2\eta} \|\bar{\boldsymbol{w}} - \boldsymbol{w}_{t-1}\|^2 + \left[ R(\bar{\boldsymbol{w}}) - R(\boldsymbol{w}_{t-1}) - \frac{\nu_s}{2} \|\boldsymbol{w}_{t-1} - \bar{\boldsymbol{w}}\|^2 \right]$$

$$- \frac{1}{2\eta} (1 - \sqrt{\beta}) \mathbb{E} \left[ \|\boldsymbol{w}_t - \bar{\boldsymbol{w}}\|^2 | \boldsymbol{w}_{t-1} \right] + \frac{B}{2C s_{t-1}}$$

$$= R(\bar{\boldsymbol{w}}) + \left[ \frac{\frac{1}{\eta} - \nu_s}{2} \right] \|\bar{\boldsymbol{w}} - \boldsymbol{w}_{t-1}\|^2 - \frac{1}{2\eta} (1 - \sqrt{\beta}) \mathbb{E} \left[ \|\boldsymbol{w}_t - \bar{\boldsymbol{w}}\|^2 | \boldsymbol{w}_{t-1} \right]$$

$$+ \frac{B}{2Cs_{t-1}},$$

where (a) follows from the RSC condition, and the fact that $s_{t-1} \in \mathbb{N}^*$.

We recall that $\eta = \frac{1}{L_s+C}$. Let us define $\alpha := \frac{C}{L_s} + 1$. Then $C = (\alpha - 1)L_s$, and $\eta = \frac{1}{\alpha L_s}$. Also recall that $\kappa_s = \frac{L_s}{\nu_s}$.
We can simplify the inequality above into:

$$\mathbb{E}[R(\boldsymbol{w}_t)|\boldsymbol{w}_{t-1}] - R(\bar{\boldsymbol{w}}) \leq \frac{1}{2\eta} \left[ \left(1 - \frac{1}{\alpha\kappa_s}\right) \|\bar{\boldsymbol{w}} - \boldsymbol{w}_{t-1}\|^2 - (1 - \sqrt{\beta})\mathbb{E}\left[\|\boldsymbol{w}_t - \bar{\boldsymbol{w}}\|^2|\boldsymbol{w}_{t-1}\right] \right. $$
$$\left. + \frac{\eta B}{Cs_{t-1}} \right].$$

We now take the expectation over $\mathrm{w}_{t-1}$ of the above inequality (i.e. we take $\mathbb{E}_{\mathrm{w}_{t-1}}[\cdot]$): using the law of total expectation ($\mathbb{E}[\cdot] = \mathbb{E}_{\mathrm{w}_{t-1}}[\mathbb{E}[\cdot|\boldsymbol{w}_{t-1}]]$) we obtain:

$$\mathbb{E}R(\boldsymbol{w}_t) - R(\bar{\boldsymbol{w}}) \leq \frac{1}{2\eta} \left[ \left(1 - \frac{1}{\alpha\kappa_s}\right) \mathbb{E}\|\bar{\boldsymbol{w}} - \boldsymbol{w}_{t-1}\|^2 - (1 - \sqrt{\beta})\mathbb{E}\|\boldsymbol{w}_t - \bar{\boldsymbol{w}}\|^2 + \frac{\eta B}{Cs_{t-1}} \right] \qquad (18)$$

Similarly as in (Liu & Foygel Barber, 2020), we now take a weighted sum over $t = 1, ..., T$, to obtain:

$$\sum_{t=1}^T 2\eta \left( \frac{1 - \frac{1}{\alpha\kappa_s}}{1 - \sqrt{\beta}} \right)^{T-t} \mathbb{E}[R(\boldsymbol{w}_t) - R(\bar{\boldsymbol{w}})]$$

$$\leq \sum_{t=1}^T \left( \frac{1 - \frac{1}{\alpha\kappa_s}}{1 - \sqrt{\beta}} \right)^{T-t} \left[ \left(1 - \frac{1}{\alpha\kappa_s}\right) \mathbb{E}\|\bar{\boldsymbol{w}} - \boldsymbol{w}_{t-1}\|^2 - (1 - \sqrt{\beta})\mathbb{E}\|\boldsymbol{w}_t - \bar{\boldsymbol{w}}\|^2 + \frac{\eta B}{Cs_{t-1}} \right]$$

$$= \sum_{t=1}^T \left( \frac{1 - \frac{1}{\alpha\kappa_s}}{1 - \sqrt{\beta}} \right)^{T-t} \left[ \left(1 - \frac{1}{\alpha\kappa_s}\right) \mathbb{E}\|\bar{\boldsymbol{w}} - \boldsymbol{w}_{t-1}\|^2 - (1 - \sqrt{\beta})\mathbb{E}\|\boldsymbol{w}_t - \bar{\boldsymbol{w}}\|^2 \right]$$
$$+ \sum_{t=1}^T \left( \frac{1 - \frac{1}{\alpha\kappa_s}}{1 - \sqrt{\beta}} \right)^{T-t} \frac{\eta B}{Cs_{t-1}}$$

$$= (1 - \sqrt{\beta}) \sum_{t=1}^T \left[ \left( \frac{1 - \frac{1}{\alpha\kappa_s}}{1 - \sqrt{\beta}} \right)^{T-t+1} \mathbb{E}\|\bar{\boldsymbol{w}} - \boldsymbol{w}_{t-1}\|^2 - \left( \frac{1 - \frac{1}{\alpha\kappa_s}}{1 - \sqrt{\beta}} \right)^{T-t} \mathbb{E}\|\boldsymbol{w}_t - \bar{\boldsymbol{w}}\|^2 \right]$$
$$+ \sum_{t=1}^T \left( \frac{1 - \frac{1}{\alpha\kappa_s}}{1 - \sqrt{\beta}} \right)^{T-t} \frac{\eta B}{Cs_{t-1}}$$

$$\overset{(a)}{=} (1 - \sqrt{\beta}) \left[ \left( \frac{1 - \frac{1}{\alpha\kappa_s}}{1 - \sqrt{\beta}} \right)^{T} \|\bar{\boldsymbol{w}} - \boldsymbol{w}_0\|^2 - \mathbb{E}\|\boldsymbol{w}_T - \bar{\boldsymbol{w}}\|^2 \right] + \sum_{t=1}^T \left( \frac{1 - \frac{1}{\alpha\kappa_s}}{1 - \sqrt{\beta}} \right)^{T-t} \frac{\eta B}{Cs_{t-1}}$$

$$\leq (1 - \sqrt{\beta}) \left( \frac{1 - \frac{1}{\alpha\kappa_s}}{1 - \sqrt{\beta}} \right)^{T} \|\bar{\boldsymbol{w}} - \boldsymbol{w}_0\|^2 + \sum_{t=1}^T \left( \frac{1 - \frac{1}{\alpha\kappa_s}}{1 - \sqrt{\beta}} \right)^{T-t} \frac{\eta B}{Cs_{t-1}}$$

$$\leq \left( \frac{1 - \frac{1}{\alpha\kappa_s}}{1 - \sqrt{\beta}} \right)^{T} \|\bar{\boldsymbol{w}} - \boldsymbol{w}_0\|^2 + \sum_{t=1}^T \left( \frac{1 - \frac{1}{\alpha\kappa_s}}{1 - \sqrt{\beta}} \right)^{T-t} \frac{\eta B}{Cs_{t-1}}, \qquad (19)$$

where (a) follows from simplifying the telescopic sum.

We now choose $k$ and $s_t$ as follows: we choose $k \geq 4\alpha^2 \kappa_s^2 \bar{k}$, which implies that:

$$\sqrt{\beta} \leq \frac{1}{2\alpha\kappa_s}$$

$$\implies \sqrt{\beta} \leq \frac{1}{2\alpha\kappa_s - 1}$$

$$\implies 1 - \sqrt{\beta} \geq 1 - \frac{1}{2\alpha\kappa_s - 1} = \frac{2\alpha\kappa_s - 2}{2\alpha\kappa_s - 1} = \frac{1 - \frac{1}{\alpha\kappa_s}}{1 - \frac{1}{2\alpha\kappa_s}}$$

$$\implies \left( \frac{1 - \frac{1}{\alpha\kappa_s}}{1 - \sqrt{\beta}} \right) \leq 1 - \frac{1}{2\alpha\kappa_s}. \tag{20}$$

And we choose $s_t := \left\lceil \frac{\tau}{\omega^t} \right\rceil$ with $\omega := 1 - \frac{1}{4\alpha\kappa_s}$ and $\tau := \frac{\eta B}{C}$.

Let us call $\nu := 1 - \frac{1}{2\alpha\kappa_s}$. Note that we have:

$$\nu \leq \omega. \tag{21}$$

And that we have the inequality below:

$$\frac{\nu}{\omega} = \frac{1 - \frac{1}{2\alpha\kappa_s}}{1 - \frac{1}{4\alpha\kappa_s}} = \frac{4\alpha\kappa_s - 2}{4\alpha\kappa_s - 1} = 1 - \frac{1}{4\alpha\kappa_s - 1} \leq 1 - \frac{1}{4\alpha\kappa_s} = \omega. \tag{22}$$

This allows us to simplify equation 19 into:

$$\mathbb{E} \sum_{t=1}^{T} 2\eta \left( \frac{1 - \frac{1}{\alpha\kappa_s}}{1 - \sqrt{\beta}} \right)^{T-t} [R(\boldsymbol{w}_t) - R(\bar{\boldsymbol{w}})] \leq \nu^T \|\bar{\boldsymbol{w}} - \boldsymbol{w}_0\|^2 + \sum_{t=1}^{T} \nu^{T-t} \omega^{t-1}$$

$$= \nu^T \|\bar{\boldsymbol{w}} - \boldsymbol{w}_0\|^2 + \frac{\omega^T}{\omega} \sum_{t=1}^{T} \left( \frac{\nu}{\omega} \right)^{T-t}$$

$$= \nu^T \|\bar{\boldsymbol{w}} - \boldsymbol{w}_0\|^2 + \frac{\omega^T}{\omega} \frac{1 - \left( \frac{\nu}{\omega} \right)^T}{1 - \left( \frac{\nu}{\omega} \right)}$$

$$\leq \nu^T \|\bar{\boldsymbol{w}} - \boldsymbol{w}_0\|^2 + \frac{\omega^T}{\omega} \frac{1}{1 - \left( \frac{\nu}{\omega} \right)}$$

$$\overset{(a)}{\leq} \nu^T \|\bar{\boldsymbol{w}} - \boldsymbol{w}_0\|^2 + \frac{\omega^T}{\omega} \frac{1}{1 - \omega}$$

$$\overset{(b)}{\leq} \nu^T \|\bar{\boldsymbol{w}} - \boldsymbol{w}_0\|^2 + \frac{4}{3} \omega^T \frac{1}{1 - \omega}$$

$$\overset{(c)}{\leq} \omega^T \|\bar{\boldsymbol{w}} - \boldsymbol{w}_0\|^2 + \frac{4}{3} \omega^T \frac{1}{1 - \omega}$$

$$\overset{(d)}{\leq} \frac{\omega^T}{1 - \omega} \|\bar{\boldsymbol{w}} - \boldsymbol{w}_0\|^2 + \frac{4}{3} \omega^T \frac{1}{1 - \omega}$$

$$= \frac{\omega^T}{1 - \omega} \left( \|\bar{\boldsymbol{w}} - \boldsymbol{w}_0\|^2 + \frac{4}{3} \right)$$

$$= 4\alpha\kappa_s \omega^T \left( \|\bar{\boldsymbol{w}} - \boldsymbol{w}_0\|^2 + \frac{4}{3} \right),$$

where in the left hand side we have used the linearity of expectation, and where (a) uses equation 22, (b) uses the fact that $\frac{1}{\omega} = \frac{1}{1 - \frac{1}{4\alpha\kappa_s}} \leq \frac{1}{1 - \frac{1}{4}} = \frac{4}{3}$ (since $\kappa_s \geq 1$ and $\alpha \geq 1$ (indeed, from the theorem's assumption $\alpha = \frac{C}{L_s} + 1$ with $C > 0$)), (c) uses equation 21, and (d) uses the fact that $\omega < 1$ so $1 < \frac{1}{1 - \omega}$.

Let us now normalize the above inequality:

$$\mathbb{E}\frac{\sum_{t=1}^{T} 2\eta \left(\frac{1-\frac{1}{\alpha\kappa_s}}{1-\sqrt{\beta}}\right)^{T-t} R(\boldsymbol{w}_t)}{\sum_{t=1}^{T} 2\eta \left(\frac{1-\frac{1}{\alpha\kappa_s}}{1-\sqrt{\beta}}\right)^{T-t}} \le R(\bar{\boldsymbol{w}}) + \frac{4\alpha\kappa_s\omega^T \left(\|\bar{\boldsymbol{w}} - \boldsymbol{w}_0\|^2 + \frac{4}{3}\right)}{\sum_{t=1}^{T} 2\eta \left(\frac{1-\frac{1}{\alpha\kappa_s}}{1-\sqrt{\beta}}\right)^{T-t}}.$$

The left hand side above is a weighted sum, which is an upper bound on the smallest term of the sum. Regarding the right hand side, we can simplify it using the fact that $0 < \left(\frac{1-\frac{1}{\alpha\kappa_s}}{1-\sqrt{\beta}}\right)$ , and therefore:

$$\sum_{t=1}^{T} \left(\frac{1 - \frac{1}{\alpha\kappa_s}}{1 - \sqrt{\beta}}\right)^{T-t} \ge 1.$$

Therefore, we obtain:

$$\mathbb{E}\min_{t\in\{1,..,T\}} R(\boldsymbol{w}_t) - R(\bar{\boldsymbol{w}}) \le \frac{4\alpha\kappa_s\omega^T \left(\|\bar{\boldsymbol{w}} - \boldsymbol{w}_0\|^2 + \frac{4}{3}\right)}{2\eta} = 2\alpha^2 L_s\kappa_s\omega^T \left(\|\bar{\boldsymbol{w}} - \boldsymbol{w}_0\|^2 + \frac{4}{3}\right)$$

Which can be simplified into the expression below, using the definition of $\hat{\boldsymbol{w}}_T$:

$$\mathbb{E}R(\hat{\boldsymbol{w}}_T) - R(\bar{\boldsymbol{w}}) \le 2\alpha^2 L_s\kappa_s\omega^T \left(\|\bar{\boldsymbol{w}} - \boldsymbol{w}_0\|^2 + \frac{4}{3}\right).$$

The proof is completed. □

**Corollary F.4.** *Under the assumptions of Theorem 4.2, let $\varepsilon$ be a small enough positive number $\varepsilon > 0$. To achieve an error $\mathbb{E}R(\hat{\boldsymbol{w}}_T) - R(\bar{\boldsymbol{w}}) \le \varepsilon$ using Algorithm 2 the number of calls to a gradient $\nabla R_i$ (#IFO), and the number of hard thresholding operations (#HT) are respectively:*

$$\#HT = \mathcal{O}(\kappa_s \log(\frac{1}{\varepsilon})), \quad \#IFO = \mathcal{O}\left(\frac{\kappa_s}{\nu_s\varepsilon}\right).$$

*Proof of Corollary F.4.* Let $\varepsilon \in \mathbb{R}_+^*$. Let us find $T$ to ensure that $\mathbb{E}R(\hat{\boldsymbol{w}}_T) - R(\bar{\boldsymbol{w}}) \le \varepsilon$. This will be enforced if:

$$2\alpha^2 L_s\kappa_s\omega^T \left(\|\bar{\boldsymbol{w}} - \boldsymbol{w}_0\|^2 + \frac{4}{3}\right) \le \varepsilon$$

$$\iff T\log(\omega) \le \log\left(\frac{\varepsilon}{2\alpha^2 L_s\kappa_s \left(\|\bar{\boldsymbol{w}} - \boldsymbol{w}_0\|^2 + \frac{4}{3}\right)}\right)$$

$$\iff T \ge \frac{1}{\log(\frac{1}{\omega})}\log\left(\frac{2\alpha^2 L_s\kappa_s \left(\|\bar{\boldsymbol{w}} - \boldsymbol{w}_0\|^2 + \frac{4}{3}\right)}{\varepsilon}\right).$$

Therefore, let us take:

$$T := \left\lceil \frac{1}{\log(\frac{1}{\omega})}\log\left(\frac{2\alpha^2 L_s\kappa_s \left(\|\bar{\boldsymbol{w}} - \boldsymbol{w}_0\|^2 + \frac{4}{3}\right)}{\varepsilon}\right) \right\rceil. \tag{23}$$

We can now derive the #IFO and #HT. First, we have one hard-thresholding operation at each iteration, therefore #HT= $T$. Using the fact that $\frac{1}{\log(\frac{1}{\omega})} = \frac{1}{-\log(\omega)} = \frac{1}{-\log(1-\frac{1}{4\alpha\kappa_s})} \le \frac{1}{\frac{1}{4\alpha\kappa_s}} = 4\alpha\kappa_s$ (since by property of the logarithm, for all $x \in (-\infty, -1) : \log(1-x) \le -x$ ), we obtain that #HT $= \mathcal{O}(\kappa_s \log\left(\frac{1}{\varepsilon}\right))$.

We now turn to computing the #IFO. At each iteration $t$ we have $s_t$ gradient evaluations, therefore:

$$\#\text{IFO} = \sum_{t=0}^{T-1} s_t$$

$$\leq \sum_{t=0}^{T-1} \left( \frac{\tau}{\omega^t} + 1 \right)$$

$$= T + \tau \frac{\left(\frac{1}{\omega}\right)^T - 1}{\frac{1}{\omega} - 1}$$

$$\leq T + \frac{\tau}{\frac{1}{\omega} - 1} \left(\frac{1}{\omega}\right)^T$$

$$= T + \frac{\tau}{\frac{1}{\omega} - 1} \exp\left( T \log\left(\frac{1}{\omega}\right) \right)$$

$$\overset{(a)}{\leq} 1 + \frac{1}{\log(\frac{1}{\omega})} \log\left( \frac{2\alpha^2 L_s \kappa_s \left(\|\bar{w} - w_0\|^2 + \frac{4}{3}\right)}{\varepsilon} \right)$$

$$+ \frac{\tau}{\frac{1}{\omega} - 1} \exp\left( \log\left(\frac{1}{\omega}\right) \left[ \frac{1}{\log(\frac{1}{\omega})} \log\left( \frac{2\alpha^2 L_s \kappa_s \left(\|\bar{w} - w_0\|^2 + \frac{4}{3}\right)}{\varepsilon} \right) + 1 \right] \right)$$

$$= 1 + \frac{1}{\log(\frac{1}{\omega})} \log\left( \frac{2\alpha^2 L_s \kappa_s \left(\|\bar{w} - w_0\|^2 + \frac{4}{3}\right)}{\varepsilon} \right) + \frac{\frac{\tau}{\omega}}{\frac{1}{\omega} - 1} \frac{2\alpha^2 L_s \kappa_s \left(\|\bar{w} - w_0\|^2 + \frac{4}{3}\right)}{\varepsilon}$$

$$= 1 + \frac{1}{\log(\frac{1}{\omega})} \log\left( \frac{2\alpha^2 L_s \kappa_s \left(\|\bar{w} - w_0\|^2 + \frac{4}{3}\right)}{\varepsilon} \right) + \frac{\tau}{1 - \omega} \frac{2\alpha^2 L_s \kappa_s \left(\|\bar{w} - w_0\|^2 + \frac{4}{3}\right)}{\varepsilon}$$

$$= 1 + \frac{1}{\log(\frac{1}{\omega})} \log\left( \frac{2\alpha^2 L_s \kappa_s \left(\|\bar{w} - w_0\|^2 + \frac{4}{3}\right)}{\varepsilon} \right) + \tau \frac{8\alpha^3 L_s \kappa_s^2 \left(\|\bar{w} - w_0\|^2 + \frac{4}{3}\right)}{\varepsilon}$$

$$\overset{(b)}{=} 1 + \frac{1}{\log(\frac{1}{\omega})} \log\left( \frac{2\alpha^2 L_s \kappa_s \left(\|\bar{w} - w_0\|^2 + \frac{4}{3}\right)}{\varepsilon} \right)$$

$$+ \frac{B}{\alpha L_s} \frac{1}{L_s(\alpha - 1)} \frac{8\alpha^3 L_s}{\varepsilon} \frac{L_s}{\nu_s} \kappa_s \left( \|\bar{w} - w_0\|^2 + \frac{4}{3} \right)$$

$$= 1 + \frac{1}{\log(\frac{1}{\omega})} \log\left( \frac{2\alpha^2 L_s \kappa_s \left(\|\bar{w} - w_0\|^2 + \frac{4}{3}\right)}{\varepsilon} \right) + \frac{8B\alpha^2 \kappa_s \left(\|\bar{w} - w_0\|^2 + \frac{4}{3}\right)}{(\alpha - 1)\nu_s} \frac{1}{\varepsilon},$$

where (a) follows from equation 23, and for (b) we recall that $\tau = \frac{\eta B}{C}$, $\eta = \frac{1}{\alpha L_s}$ and $C = L_s(\alpha - 1)$.

Therefore, overall, the IFO complexity is in $\mathcal{O}(\frac{\kappa_s}{\nu_s \varepsilon})$.

$\square$

### F.2.3. PROOF OF THEOREM 4.3

We now proceed with the full proof of Theorem 4.3.

*Proof of Theorem 4.3.* Similary as in the proof of Theorem 4.2 in Section F.2.2, let us take: $\eta := \frac{1}{L_s + C}$, and $\alpha := \frac{C}{L_s} + 1$. Then $C = (\alpha - 1)L_s$, and $\eta = \frac{1}{\alpha L_s}$. Recall that $\kappa_s := \frac{L_s}{\nu_s}$. Denote $v_t = \mathcal{H}_k(w_{t-1} - \eta \nabla R(w_{t-1}))$ for any $t \in \mathbb{N}$.

Similarly as in Section F.2.2, the $L_s$-smoothness of $R$ implies that

$$
\begin{aligned}
R(\boldsymbol{w}_t) \leq & R(\boldsymbol{w}_{t-1}) + \langle \nabla R(\boldsymbol{w}_{t-1}), \boldsymbol{w}_t - \boldsymbol{w}_{t-1} \rangle + \frac{L_s}{2} \|\boldsymbol{w}_t - \boldsymbol{w}_{t-1}\|^2 \\
= & R(\boldsymbol{w}_{t-1}) + \langle \boldsymbol{g}_{t-1}, \boldsymbol{w}_t - \boldsymbol{w}_{t-1} \rangle + \frac{L_s}{2} \|\boldsymbol{w}_t - \boldsymbol{w}_{t-1}\|^2 + \langle \nabla R(\boldsymbol{w}_{t-1}) - \boldsymbol{g}_{t-1}, \boldsymbol{w}_t - \boldsymbol{w}_{t-1} \rangle \\
= & R(\boldsymbol{w}_{t-1}) + \frac{1}{2\eta} \left[ \|\boldsymbol{w}_t - (\boldsymbol{w}_{t-1} - \eta \boldsymbol{g}_{t-1})\|^2 - \eta^2 \|\boldsymbol{g}_{t-1}\|^2 - \|\boldsymbol{w}_t - \boldsymbol{w}_{t-1}\|^2 \right] + \frac{L_s}{2} \|\boldsymbol{w}_t - \boldsymbol{w}_{t-1}\|^2 \\
& + \langle \nabla R(\boldsymbol{w}_{t-1}) - \boldsymbol{g}_{t-1}, \boldsymbol{w}_t - \boldsymbol{w}_{t-1} \rangle \\
= & R(\boldsymbol{w}_{t-1}) + \frac{1}{2\eta} \|\boldsymbol{w}_t - (\boldsymbol{w}_{t-1} - \eta \boldsymbol{g}_{t-1})\|^2 - \frac{\eta}{2} \|\boldsymbol{g}_{t-1}\|^2 + \left[ \frac{L_s - \frac{1}{\eta}}{2} \right] \|\boldsymbol{w}_t - \boldsymbol{w}_{t-1}\|^2 \\
& + \langle \nabla R(\boldsymbol{w}_{t-1}) - \boldsymbol{g}_{t-1}, \boldsymbol{w}_t - \boldsymbol{w}_{t-1} \rangle \\
\overset{(a)}{\leq} & R(\boldsymbol{w}_{t-1}) + \frac{1}{2\eta} \left[ \|\bar{\boldsymbol{w}} - (\boldsymbol{w}_{t-1} - \eta \boldsymbol{g}_{t-1})\|^2 - \|\boldsymbol{w}_t - \bar{\boldsymbol{w}}\|^2 + \sqrt{\beta} \|\boldsymbol{v}_t - \bar{\boldsymbol{w}}\|^2 \right] - \frac{\eta}{2} \|\boldsymbol{g}_{t-1}\|^2 \\
& + \left[ \frac{L_s - \frac{1}{\eta}}{2} \right] \|\boldsymbol{w}_t - \boldsymbol{w}_{t-1}\|^2 + \langle \nabla R(\boldsymbol{w}_{t-1}) - \boldsymbol{g}_{t-1}, \boldsymbol{w}_t - \boldsymbol{w}_{t-1} \rangle \\
= & R(\boldsymbol{w}_{t-1}) + \frac{1}{2\eta} \left[ \|\bar{\boldsymbol{w}} - \boldsymbol{w}_{t-1}\|^2 + \eta^2 \|\boldsymbol{g}_{t-1}\|^2 - 2\langle \eta \boldsymbol{g}_{t-1}, \boldsymbol{w}_{t-1} - \bar{\boldsymbol{w}} \rangle \right] - \frac{1}{2\eta} \|\boldsymbol{w}_t - \bar{\boldsymbol{w}}\|^2 \\
& + \frac{\sqrt{\beta}}{2\eta} \|\boldsymbol{v}_t - \bar{\boldsymbol{w}}\|^2 - \frac{\eta}{2} \|\boldsymbol{g}_{t-1}\|^2 + \left[ \frac{L_s - \frac{1}{\eta}}{2} \right] \|\boldsymbol{w}_t - \boldsymbol{w}_{t-1}\|^2 + \langle \nabla R(\boldsymbol{w}_{t-1}) - \boldsymbol{g}_{t-1}, \boldsymbol{w}_t - \boldsymbol{w}_{t-1} \rangle \\
= & R(\boldsymbol{w}_{t-1}) + \frac{1}{2\eta} \left[ \|\bar{\boldsymbol{w}} - \boldsymbol{w}_{t-1}\|^2 - 2\langle \eta \boldsymbol{g}_{t-1}, \boldsymbol{w}_{t-1} - \bar{\boldsymbol{w}} \rangle \right] - \frac{1}{2\eta} \|\boldsymbol{w}_t - \bar{\boldsymbol{w}}\|^2 + \frac{\sqrt{\beta}}{2\eta} \|\boldsymbol{v}_t - \bar{\boldsymbol{w}}\|^2 \quad (24) \\
& + \left[ \frac{L_s - \frac{1}{\eta}}{2} \right] \|\boldsymbol{w}_t - \boldsymbol{w}_{t-1}\|^2 + \langle \nabla R(\boldsymbol{w}_{t-1}) - \boldsymbol{g}_{t-1}, \boldsymbol{w}_t - \boldsymbol{w}_{t-1} \rangle \\
\overset{(b)}{=} & R(\boldsymbol{w}_{t-1}) + \frac{1}{2\eta} \|\bar{\boldsymbol{w}} - \boldsymbol{w}_{t-1}\|^2 - \langle \boldsymbol{g}_{t-1}, \boldsymbol{w}_{t-1} - \bar{\boldsymbol{w}} \rangle - \frac{1}{2\eta} \|\boldsymbol{w}_t - \bar{\boldsymbol{w}}\|^2 + \frac{\sqrt{\beta}}{2\eta} \|\boldsymbol{v}_t - \bar{\boldsymbol{w}}\|^2 \\
& + \left[ \frac{L_s - \frac{1}{\eta} + C}{2} \right] \|\boldsymbol{w}_t - \boldsymbol{w}_{t-1}\|^2 + \frac{1}{2C} \|\nabla R(\boldsymbol{w}_{t-1}) - \boldsymbol{g}_{t-1}\|^2, \quad (25)
\end{aligned}
$$

where (a) follows from Lemma 3.6 and (b) follows from the inequality $\langle a, b \rangle \leq \frac{C}{2} a^2 + \frac{1}{2C} b^2$, for any $(a, b) \in (\mathbb{R}^d)^2$ with $C > 0$ an arbitrary strictly positive constant. Let us now take $\eta := \frac{1}{L_s + C}$: therefore the term $\left[ \frac{L_s - \frac{1}{\eta} + C}{2} \right] \|\boldsymbol{w}_t - \boldsymbol{w}_{t-1}\|^2$ above is 0. We now take the conditional expectation (conditioned on $\mathsf{w}_{t-1}$, which is the random variable which realizations are $\boldsymbol{w}_{t-1}$), on both sides, and from Lemma F.3 we obtain the inequality below (we slightly abuse notations and denote $\mathbb{E}[\cdot | \mathsf{w}_{t-1} = \boldsymbol{w}_{t-1}]$ by $\mathbb{E}[\cdot | \boldsymbol{w}_{t-1}]$):

$$
\begin{aligned}
\mathbb{E}[R(\boldsymbol{w}_t) | \boldsymbol{w}_{t-1}] \leq & R(\boldsymbol{w}_{t-1}) + \frac{1}{2\eta} \|\bar{\boldsymbol{w}} - \boldsymbol{w}_{t-1}\|^2 - \langle \nabla R(\boldsymbol{w}_{t-1}), \boldsymbol{w}_{t-1} - \bar{\boldsymbol{w}} \rangle \\
& - \frac{1}{2\eta} \mathbb{E}\left[ \|\boldsymbol{w}_t - \bar{\boldsymbol{w}}\|^2 | \boldsymbol{w}_{t-1} \right] + \frac{\sqrt{\beta}}{2\eta} \mathbb{E}\left[ \|\boldsymbol{v}_t - \bar{\boldsymbol{w}}\|^2 | \boldsymbol{w}_{t-1} \right] + \frac{B(n - s_{t-1})}{2C s_{t-1}(n - 1)} \\
\overset{(a)}{\leq} & R(\boldsymbol{w}_{t-1}) + \frac{1}{2\eta} \|\bar{\boldsymbol{w}} - \boldsymbol{w}_{t-1}\|^2 + \left[ R(\bar{\boldsymbol{w}}) - R(\boldsymbol{w}_{t-1}) - \frac{\nu_s}{2} \|\boldsymbol{w}_{t-1} - \bar{\boldsymbol{w}}\|^2 \right] \\
& - \frac{1}{2\eta} \mathbb{E}\left[ \|\boldsymbol{w}_t - \bar{\boldsymbol{w}}\|^2 | \boldsymbol{w}_{t-1} \right] + \frac{\sqrt{\beta}}{2\eta} \mathbb{E}\left[ \|\boldsymbol{v}_t - \bar{\boldsymbol{w}}\|^2 | \boldsymbol{w}_{t-1} \right] + \frac{B}{2C s_{t-1}} \\
= & R(\bar{\boldsymbol{w}}) + \left[ \frac{\frac{1}{\eta} - \nu_s}{2} \right] \|\bar{\boldsymbol{w}} - \boldsymbol{w}_{t-1}\|^2 - \frac{1}{2\eta} \mathbb{E}\left[ \|\boldsymbol{w}_t - \bar{\boldsymbol{w}}\|^2 | \boldsymbol{w}_{t-1} \right] + \frac{\sqrt{\beta}}{2\eta} \mathbb{E}\left[ \|\boldsymbol{v}_t - \bar{\boldsymbol{w}}\|^2 | \boldsymbol{w}_{t-1} \right] \\
& + \frac{B}{2C s_{t-1}}, \quad (26)
\end{aligned}
$$

where (a) follows from the RSC condition, and the fact that $s_{t-1} \in \mathbb{N}^*$.

Now recall that we have taken $\eta = \frac{1}{L_s+C}$, and let us define $\alpha := \frac{C}{L_s} + 1$. Then $C = (\alpha - 1)L_s$, and $\eta = \frac{1}{\alpha L_s}$. Also recall that $\kappa_s = \frac{L_s}{\nu_s}$.

We can simplify the inequality above into:

$$\mathbb{E}[R(\boldsymbol{w}_t)|\boldsymbol{w}_{t-1}] - R(\bar{\boldsymbol{w}})$$
$$\leq \frac{1}{2\eta}\left[\left(1 - \frac{1}{\alpha\kappa_s}\right)\|\bar{\boldsymbol{w}} - \boldsymbol{w}_{t-1}\|^2 - \mathbb{E}\left[\|\boldsymbol{w}_t - \bar{\boldsymbol{w}}\|^2|\boldsymbol{w}_{t-1}\right] + \sqrt{\beta}\mathbb{E}\left[\|\boldsymbol{v}_t - \bar{\boldsymbol{w}}\|^2|\boldsymbol{w}_{t-1}\right] + \frac{\eta B}{Cs_{t-1}}\right].$$

We now take the expectation over $\mathrm{w}_{t-1}$ of the above inequality (i.e. we take $\mathbb{E}_{\mathrm{w}_{t-1}}[\cdot]$): using the law of total expectation ($\mathbb{E}[\cdot] = \mathbb{E}_{\mathrm{w}_{t-1}}[\mathbb{E}[\cdot|\boldsymbol{w}_{t-1}]]$) we obtain:

$$\mathbb{E}R(\boldsymbol{w}_t) - R(\bar{\boldsymbol{w}}) \leq \frac{1}{2\eta}\left[\left(1 - \frac{1}{\alpha\kappa_s}\right)\mathbb{E}\|\bar{\boldsymbol{w}} - \boldsymbol{w}_{t-1}\|^2 - \mathbb{E}\|\boldsymbol{w}_t - \bar{\boldsymbol{w}}\|^2 + \sqrt{\beta}\mathbb{E}\|\boldsymbol{v}_t - \bar{\boldsymbol{w}}\|^2 + \frac{\eta B}{Cs_{t-1}}\right].$$

Additionally, in view of equation 18 applied at $\boldsymbol{v}_t$ instead of $\boldsymbol{w}_t$, (since $\boldsymbol{v}_t$ here corresponds to the $\boldsymbol{w}_t$ from Section F.2.2, i.e. $\boldsymbol{v}_t$ is the hard-thresholding of an iterate after a gradient step), we know that:

$$\mathbb{E}R(\boldsymbol{v}_t) - R(\bar{\boldsymbol{w}}) \leq \frac{1}{2\eta}\left[\left(1 - \frac{1}{\alpha\kappa_s}\right)\mathbb{E}\|\bar{\boldsymbol{w}} - \boldsymbol{w}_{t-1}\|^2 - (1 - \sqrt{\beta})\mathbb{E}\|\boldsymbol{w}_t - \bar{\boldsymbol{w}}\|^2 + \frac{\eta B}{Cs_{t-1}}\right].$$

We now take a convex combination similarly as in the case without additional constraint (section E.2), for some $\rho \in (0, 1)$.

$$\mathbb{E}(1 - \rho)R(\boldsymbol{w}_t) + \rho R(\boldsymbol{v}_t)$$
$$\leq R(\bar{\boldsymbol{w}}) + \frac{1}{2\eta}\left[\left(1 - \frac{1}{\alpha\kappa_s}\right)\mathbb{E}\|\bar{\boldsymbol{w}} - \boldsymbol{w}_{t-1}\|^2 - (1 - \rho)\mathbb{E}\|\boldsymbol{w}_t - \bar{\boldsymbol{w}}\|^2\right.$$
$$\left. + \left((1 - \rho)\sqrt{\beta} - (1 - \sqrt{\beta})\rho\right)\mathbb{E}\|\boldsymbol{v}_t - \bar{\boldsymbol{w}}\|^2 + \frac{\eta B}{Cs_{t-1}}\right]$$
$$= R(\bar{\boldsymbol{w}}) + \frac{1}{2\eta}\left[\left(1 - \frac{1}{\alpha\kappa_s}\right)\mathbb{E}\|\bar{\boldsymbol{w}} - \boldsymbol{w}_{t-1}\|^2 - (1 - \rho)\mathbb{E}\|\boldsymbol{w}_t - \bar{\boldsymbol{w}}\|^2\right.$$
$$\left. - \left(\rho - \sqrt{\beta}\right)\mathbb{E}\|\boldsymbol{v}_t - \bar{\boldsymbol{w}}\|^2 + \frac{\eta B}{Cs_{t-1}}\right]$$
$$\overset{(b)}{\leq} R(\bar{\boldsymbol{w}}) + \frac{1}{2\eta}\left[\left(1 - \frac{1}{\alpha\kappa_s}\right)\mathbb{E}\|\bar{\boldsymbol{w}} - \boldsymbol{w}_{t-1}\|^2 - (1 - \rho)\mathbb{E}\|\boldsymbol{w}_t - \bar{\boldsymbol{w}}\|^2\right.$$
$$\left. - \left(\rho - \sqrt{\beta}\right)\mathbb{E}\|\boldsymbol{w}_t - \bar{\boldsymbol{w}}\|^2 + \frac{\eta B}{Cs_{t-1}}\right]$$
$$= R(\bar{\boldsymbol{w}}) + \frac{1}{2\eta}\left[\left(1 - \frac{1}{\alpha\kappa_s}\right)\mathbb{E}\|\bar{\boldsymbol{w}} - \boldsymbol{w}_{t-1}\|^2 - (1 - \sqrt{\beta})\mathbb{E}\|\boldsymbol{w}_t - \bar{\boldsymbol{w}}\|^2 + \frac{\eta B}{Cs_{t-1}}\right],$$

where in (b), we have assumed that $\sqrt{\beta} \leq \rho$ (later we will verify that our choice of $k$ ensures such a condition), and have used the fact that projection onto a convex set is non-expansive (which implies that $\|\boldsymbol{v}_t - \bar{\boldsymbol{w}}\|^2 \geq \|\boldsymbol{w}_t - \bar{\boldsymbol{w}}\|^2$). Similarly as in F.2.2, we now take a weighted sum over $t = 1, ..., T$, to obtain:

$$\sum_{t=1}^{T} 2\eta \left( \frac{1 - \frac{1}{\alpha \kappa_s}}{1 - \sqrt{\beta}} \right)^{T-t} \mathbb{E}[(1-\rho)R(\boldsymbol{w}_t) + \rho R(\boldsymbol{v}_t) - R(\bar{\boldsymbol{w}})]$$

$$\leq \sum_{t=1}^{T} \left( \frac{1 - \frac{1}{\alpha \kappa_s}}{1 - \sqrt{\beta}} \right)^{T-t} \left[ \left( 1 - \frac{1}{\alpha \kappa_s} \right) \mathbb{E}\|\bar{\boldsymbol{w}} - \boldsymbol{w}_{t-1}\|^2 - (1 - \sqrt{\beta})\mathbb{E}\|\boldsymbol{w}_t - \bar{\boldsymbol{w}}\|^2 + \frac{\eta B}{C s_{t-1}} \right]$$

$$= \sum_{t=1}^{T} \left( \frac{1 - \frac{1}{\alpha \kappa_s}}{1 - \sqrt{\beta}} \right)^{T-t} \left[ \left( 1 - \frac{1}{\alpha \kappa_s} \right) \mathbb{E}\|\bar{\boldsymbol{w}} - \boldsymbol{w}_{t-1}\|^2 - (1 - \sqrt{\beta})\mathbb{E}\|\boldsymbol{w}_t - \bar{\boldsymbol{w}}\|^2 \right]$$

$$+ \sum_{t=1}^{T} \left( \frac{1 - \frac{1}{\alpha \kappa_s}}{1 - \sqrt{\beta}} \right)^{T-t} \frac{\eta B}{C s_{t-1}}$$

$$= (1 - \sqrt{\beta}) \sum_{t=1}^{T} \left[ \left( \frac{1 - \frac{1}{\alpha \kappa_s}}{1 - \sqrt{\beta}} \right)^{T-t+1} \mathbb{E}\|\bar{\boldsymbol{w}} - \boldsymbol{w}_{t-1}\|^2 - \left( \frac{1 - \frac{1}{\alpha \kappa_s}}{1 - \sqrt{\beta}} \right)^{T-t} \mathbb{E}\|\boldsymbol{w}_t - \bar{\boldsymbol{w}}\|^2 \right]$$

$$+ \sum_{t=1}^{T} \left( \frac{1 - \frac{1}{\alpha \kappa_s}}{1 - \sqrt{\beta}} \right)^{T-t} \frac{\eta B}{C s_{t-1}}$$

$$\overset{(a)}{=} (1 - \sqrt{\beta}) \left[ \left( \frac{1 - \frac{1}{\alpha \kappa_s}}{1 - \sqrt{\beta}} \right)^{T} \|\bar{\boldsymbol{w}} - \boldsymbol{w}_0\|^2 - \mathbb{E}\|\boldsymbol{w}_T - \bar{\boldsymbol{w}}\|^2 \right] + \sum_{t=1}^{T} \left( \frac{1 - \frac{1}{\alpha \kappa_s}}{1 - \sqrt{\beta}} \right)^{T-t} \frac{\eta B}{C s_{t-1}}$$

$$\leq (1 - \sqrt{\beta}) \left( \frac{1 - \frac{1}{\alpha \kappa_s}}{1 - \sqrt{\beta}} \right)^{T} \|\bar{\boldsymbol{w}} - \boldsymbol{w}_0\|^2 + \sum_{t=1}^{T} \left( \frac{1 - \frac{1}{\alpha \kappa_s}}{1 - \sqrt{\beta}} \right)^{T-t} \frac{\eta B}{C s_{t-1}}$$

$$\leq \left( \frac{1 - \frac{1}{\alpha \kappa_s}}{1 - \sqrt{\beta}} \right)^{T} \|\bar{\boldsymbol{w}} - \boldsymbol{w}_0\|^2 + \sum_{t=1}^{T} \left( \frac{1 - \frac{1}{\alpha \kappa_s}}{1 - \sqrt{\beta}} \right)^{T-t} \frac{\eta B}{C s_{t-1}}, \tag{27}$$

where (a) follows from simplifying the telescopic sum.

We now choose $k$ and $s_t$ as follows: we choose $k \geq 4 \frac{1}{\rho^2} \alpha^2 \kappa_s^2 \bar{k}$, which implies that:

$\rho \geq \sqrt{\beta}$ (thereby verifying the assumption made earlier), and that:

$$\sqrt{\beta} \leq \frac{1}{2\alpha \frac{1}{\rho} \kappa_s}$$

$$\implies \sqrt{\beta} \leq \frac{1}{2\alpha \frac{1}{\rho} \kappa_s - 1}$$

$$\implies 1 - \sqrt{\beta} \geq 1 - \frac{1}{2\alpha \frac{1}{\rho} \kappa_s - 1} = \frac{2\alpha \frac{1}{\rho} \kappa_s - 2}{2\alpha \frac{1}{\rho} \kappa_s - 1} = \frac{1 - \frac{1}{\alpha \frac{1}{\rho} \kappa_s}}{1 - \frac{1}{2\alpha \frac{1}{\rho} \kappa_s}} \overset{(a)}{\geq} \frac{1 - \frac{1}{\alpha \kappa_s}}{1 - \frac{1}{2\alpha \frac{1}{\rho} \kappa_s}}$$

$$\implies \left( \frac{1 - \frac{1}{\alpha \kappa_s}}{1 - \sqrt{\beta}} \right) \leq 1 - \frac{1}{2\alpha \frac{1}{\rho} \kappa_s}, \tag{28}$$

where (a) follows from the fact that $\rho \leq 1$.

And we now choose $s_t := \left\lceil \frac{\tau}{\omega^t} \right\rceil$, with $\omega := 1 - \frac{1}{4\alpha \frac{1}{\rho} \kappa_s}$ and $\tau := \frac{\eta B}{C}$.

Let us call $\nu := 1 - \frac{1}{2\alpha \frac{1}{\rho} \kappa_s}$. Note that we have:

$$\nu \leq \omega. \tag{29}$$

And that we have the inequality below:

$$\frac{\nu}{\omega} = \frac{1 - \frac{1}{2\alpha\frac{1}{\rho}\kappa_s}}{1 - \frac{1}{4\alpha\frac{1}{\rho}\kappa_s}} = \frac{4\alpha\frac{1}{\rho}\kappa_s - 2}{4\alpha\frac{1}{\rho}\kappa_s - 1} = 1 - \frac{1}{4\alpha\frac{1}{\rho}\kappa_s - 1} \leq 1 - \frac{1}{4\alpha\frac{1}{\rho}\kappa_s} = \omega. \tag{30}$$

This allows us to simplify equation 27 into:

$$\mathbb{E} \sum_{t=1}^{T} 2\eta \left( \frac{1 - \frac{1}{\alpha\kappa_s}}{1 - \sqrt{\beta}} \right)^{T-t} [(1-\rho)R(\boldsymbol{w}_t) + \rho R(\boldsymbol{v}_t) - R(\bar{\boldsymbol{w}})]$$

$$\leq \nu^T \|\bar{\boldsymbol{w}} - \boldsymbol{w}_0\|^2 + \sum_{t=1}^{T} \nu^{T-t} \omega^{t-1}$$

$$= \nu^T \|\bar{\boldsymbol{w}} - \boldsymbol{w}_0\|^2 + \frac{\omega^T}{\omega} \sum_{t=1}^{T} \left( \frac{\nu}{\omega} \right)^{T-t}$$

$$= \nu^T \|\bar{\boldsymbol{w}} - \boldsymbol{w}_0\|^2 + \frac{\omega^T}{\omega} \frac{1 - \left( \frac{\nu}{\omega} \right)^T}{1 - \left( \frac{\nu}{\omega} \right)}$$

$$\leq \nu^T \|\bar{\boldsymbol{w}} - \boldsymbol{w}_0\|^2 + \frac{\omega^T}{\omega} \frac{1}{1 - \left( \frac{\nu}{\omega} \right)}$$

$$\overset{(a)}{\leq} \nu^T \|\bar{\boldsymbol{w}} - \boldsymbol{w}_0\|^2 + \frac{\omega^T}{\omega} \frac{1}{1 - \omega}$$

$$\overset{(b)}{\leq} \nu^T \|\bar{\boldsymbol{w}} - \boldsymbol{w}_0\|^2 + \frac{4}{3}\omega^T \frac{1}{1 - \omega}$$

$$\overset{(c)}{\leq} \omega^T \|\bar{\boldsymbol{w}} - \boldsymbol{w}_0\|^2 + \frac{4}{3}\omega^T \frac{1}{1 - \omega}$$

$$\overset{(d)}{\leq} \frac{\omega^T}{1 - \omega} \|\bar{\boldsymbol{w}} - \boldsymbol{w}_0\|^2 + \frac{4}{3}\omega^T \frac{1}{1 - \omega}$$

$$= \frac{\omega^T}{1 - \omega} \left( \|\bar{\boldsymbol{w}} - \boldsymbol{w}_0\|^2 + \frac{4}{3} \right)$$

$$= 4\alpha\frac{1}{\rho}\kappa_s \omega^T \left( \|\bar{\boldsymbol{w}} - \boldsymbol{w}_0\|^2 + \frac{4}{3} \right),$$

where in the left hand side we have used the linearity of expectation, and where (a) uses equation 30, (b) uses the fact that $\frac{1}{\omega} = \frac{1}{1 - \frac{1}{4\alpha\frac{1}{\rho}\kappa_s}} \leq \frac{1}{1 - \frac{1}{4}} = \frac{4}{3}$ (since $\kappa_s \geq 1$ and $\alpha \geq 1$ (indeed, from the theorem's assumption $\alpha = \frac{C}{L_s} + 1$ with $C > 0$), so consequently $\alpha\frac{1}{\rho} \geq 1$), (c) uses equation 29, and (d) uses the fact that $\omega < 1$ so $1 < \frac{1}{1-\omega}$.

Let us now normalize the above inequality:

$$\mathbb{E} \frac{\sum_{t=1}^{T} 2\eta \left( \frac{1 - \frac{1}{\alpha\kappa_s}}{1 - \sqrt{\beta}} \right)^{T-t} (1-\rho)R(\boldsymbol{w}_t) + \rho R(\boldsymbol{v}_t)}{\sum_{t=1}^{T} 2\eta \left( \frac{1 - \frac{1}{\alpha\kappa_s}}{1 - \sqrt{\beta}} \right)^{T-t}} \leq R(\bar{\boldsymbol{w}}) + \frac{4\alpha\frac{1}{\rho}\kappa_s \omega^T \left( \|\bar{\boldsymbol{w}} - \boldsymbol{w}_0\|^2 + \frac{4}{3} \right)}{\sum_{t=1}^{T} 2\eta \left( \frac{1 - \frac{1}{\alpha\kappa_s}}{1 - \sqrt{\beta}} \right)^{T-t}}.$$

The left hand side above is a weighted sum, which is an upper bound on the smallest term of the sum.

Regarding the right hand side, we can simplify it using the fact that $0 < \left( \frac{1 - \frac{1}{\alpha\kappa_s}}{1 - \sqrt{\beta}} \right)$, and therefore:

$$\sum_{t=1}^{T} \left( \frac{1 - \frac{1}{\alpha\kappa_s}}{1 - \sqrt{\beta}} \right)^{T-t} \geq 1.$$

Therefore, we obtain:

$$\mathbb{E} \min_{t \in \{1,..,T\}} (1-\rho)R(\boldsymbol{w}_t) + \rho R(\boldsymbol{v}_t) - R(\bar{\boldsymbol{w}}) \leq \frac{4\alpha \frac{1}{\rho}\kappa_s \omega^T \left(\|\bar{\boldsymbol{w}} - \boldsymbol{w}_0\|^2 + \frac{4}{3}\right)}{2\eta}$$

$$= 2\alpha^2 \frac{1}{\rho} L_s \kappa_s \omega^T \left(\|\bar{\boldsymbol{w}} - \boldsymbol{w}_0\|^2 + \frac{4}{3}\right). \tag{31}$$

We denote by $\varepsilon_T$ the right-hand side above:

$$\varepsilon_T = 2\alpha^2 \frac{1}{\rho} L_s \kappa_s \omega^T \left(\|\bar{\boldsymbol{w}} - \boldsymbol{w}_0\|^2 + \frac{4}{3}\right).$$

We now proceed similarly as in the proof of Theorem 3.7 above. Recall that we have assumed in the Assumptions of Theorem 4.3, without loss of generality, that $R$ is non-negative, which implies that $R(\boldsymbol{v}_t) \geq 0$. Plugging this in equation 31 implies that:

$$\mathbb{E} \min_{t \in [T]} R(\boldsymbol{w}_t) \leq \frac{1}{1-\rho}R(\bar{\boldsymbol{w}}) + \frac{\varepsilon_T}{1-\rho} \leq (1+2\rho)R(\bar{\boldsymbol{w}}) + \frac{\varepsilon_T}{1-\rho}. \tag{32}$$

Plugging the change of variable $\varepsilon'_T = \frac{\varepsilon_T}{1-\rho}$ into equation 32 above, we obtain that:

$$\mathbb{E} \min_{t \in [T]} R(\boldsymbol{w}_t) \leq (1+2\rho)R(\bar{\boldsymbol{w}}) + \varepsilon'_T.$$

Further, consider an ideal case where $\bar{\boldsymbol{w}}$ is a global minimizer of $R$ over $\mathcal{B}_0(k) := \{\boldsymbol{w} : \|\boldsymbol{w}\|_0 \leq k\}$. Then $R(\boldsymbol{v}_t) \geq R(\bar{\boldsymbol{w}})$ is always true for all $t \geq 1$. It follows that the bound in equation 32 yields:

$$\mathbb{E} \min_{t \in [T]} \{(1-\rho)R(\boldsymbol{w}_t) + \rho R(\bar{\boldsymbol{w}})\} \leq \mathbb{E} \min_{t \in [T]} \{(1-\rho)R(\boldsymbol{w}_t) + \rho R(\boldsymbol{v}_t)\} \leq R(\bar{\boldsymbol{w}}) + \varepsilon_T,$$

which implies: $\mathbb{E} \min_{t \in [T]} R(\boldsymbol{w}_t) \leq R(\bar{\boldsymbol{w}}) + \frac{\varepsilon_T}{1-\rho}$. In this case, we can simply set $\rho = 0.5$, and define $\varepsilon'_T = \frac{\varepsilon_T}{1-\rho} = 2\varepsilon_T$ similarly as above.. The proof is completed. $\qquad\square$

### F.3. Proof of Corollary 4.4

*Proof of Corollary 4.4.* We proceed similarly as in the proof of Corollary F.4 in Section F.2.2:

Let $\varepsilon \in \mathbb{R}_+^*$. Let us find $T$ to ensure that $\mathbb{E} \min_{t \in \{1,..,T\}} (1-\rho)R(\boldsymbol{w}_t) + \rho R(\boldsymbol{v}_t) - R(\bar{\boldsymbol{w}}) \leq \varepsilon$ This will be enforced if:

$$2\alpha^2 \frac{1}{\rho} L_s \kappa_s \omega^T \left(\|\bar{\boldsymbol{w}} - \boldsymbol{w}_0\|^2 + \frac{4}{3}\right) \leq \varepsilon$$

$$\iff T\log(\omega) \leq \log\left(\frac{\varepsilon}{2\alpha^2 \frac{1}{\rho} L_s \kappa_s \left(\|\bar{\boldsymbol{w}} - \boldsymbol{w}_0\|^2 + \frac{4}{3}\right)}\right)$$

$$\iff T \geq \frac{1}{\log(\frac{1}{\omega})} \log\left(\frac{2\alpha^2 \frac{1}{\rho} L_s \kappa_s \left(\|\bar{\boldsymbol{w}} - \boldsymbol{w}_0\|^2 + \frac{4}{3}\right)}{\varepsilon}\right).$$

Therefore, let us take:

$$T := \left\lceil \frac{1}{\log(\frac{1}{\omega})} \log\left(\frac{2\alpha^2 \frac{1}{\rho} L_s \kappa_s \left(\|\bar{\boldsymbol{w}} - \boldsymbol{w}_0\|^2 + \frac{4}{3}\right)}{\varepsilon}\right) \right\rceil. \tag{33}$$

We can now derive the #IFO and #HT. First, we have one hard-thresholding operation at each iteration, therefore #HT$= T$. Using the fact that $\frac{1}{\log(\frac{1}{\omega})} = \frac{1}{-\log(\omega)} = \frac{1}{-\log(1-\frac{1}{4\alpha \frac{1}{\rho}\kappa_s})} \leq \frac{1}{\frac{1}{4\alpha \frac{1}{\rho}\kappa_s}} = 4\alpha \frac{1}{\rho}\kappa_s$ (since by property of the logarithm, for all $x \in (-\infty, -1) : \log(1-x) \leq -x$ ), we obtain that #HT $= \mathcal{O}(\kappa_s \log\left(\frac{1}{\varepsilon}\right))$.

We now turn to computing the #IFO. At each iteration $t$ we have $s_t$ gradient evaluations, therefore:

$$
\begin{aligned}
\#\text{IFO} &= \sum_{t=0}^{T-1} s_t \\
&\leq \sum_{t=0}^{T-1} \left( \frac{\tau}{\omega^t} + 1 \right) \\
&= T + \tau \frac{\left(\frac{1}{\omega}\right)^T - 1}{\frac{1}{\omega} - 1} \\
&\leq T + \frac{\tau}{\frac{1}{\omega} - 1} \left(\frac{1}{\omega}\right)^T \\
&= T + \frac{\tau}{\frac{1}{\omega} - 1} \exp\left( T \log\left(\frac{1}{\omega}\right) \right) \\
&\overset{(a)}{\leq} 1 + \frac{1}{\log(\frac{1}{\omega})} \log\left( \frac{2\alpha^2 \frac{1}{\rho} L_s \kappa_s \left( \|\bar{\boldsymbol{w}} - \boldsymbol{w}_0\|^2 + \frac{4}{3} \right)}{\varepsilon} \right) \\
&\quad + \frac{\tau}{\frac{1}{\omega} - 1} \exp\left( \log\left(\frac{1}{\omega}\right) \left[ \frac{1}{\log(\frac{1}{\omega})} \log\left( \frac{2\alpha^2 \frac{1}{\rho} L_s \kappa_s \left( \|\bar{\boldsymbol{w}} - \boldsymbol{w}_0\|^2 + \frac{4}{3} \right)}{\varepsilon} \right) + 1 \right] \right) \\
&= 1 + \frac{1}{\log(\frac{1}{\omega})} \log\left( \frac{2\alpha^2 \frac{1}{\rho} L_s \kappa_s \left( \|\bar{\boldsymbol{w}} - \boldsymbol{w}_0\|^2 + \frac{4}{3} \right)}{\varepsilon} \right) + \frac{\frac{\tau}{\omega}}{\frac{1}{\omega} - 1} \frac{2\alpha^2 \frac{1}{\rho} L_s \kappa_s \left( \|\bar{\boldsymbol{w}} - \boldsymbol{w}_0\|^2 + \frac{4}{3} \right)}{\varepsilon} \\
&= 1 + \frac{1}{\log(\frac{1}{\omega})} \log\left( \frac{2\alpha^2 \frac{1}{\rho} L_s \kappa_s \left( \|\bar{\boldsymbol{w}} - \boldsymbol{w}_0\|^2 + \frac{4}{3} \right)}{\varepsilon} \right) + \frac{\tau}{1 - \omega} \frac{2\alpha^2 \frac{1}{\rho} L_s \kappa_s \left( \|\bar{\boldsymbol{w}} - \boldsymbol{w}_0\|^2 + \frac{4}{3} \right)}{\varepsilon} \\
&= 1 + \frac{1}{\log(\frac{1}{\omega})} \log\left( \frac{2\alpha^2 \frac{1}{\rho} L_s \kappa_s \left( \|\bar{\boldsymbol{w}} - \boldsymbol{w}_0\|^2 + \frac{4}{3} \right)}{\varepsilon} \right) + \tau \frac{8\alpha^3 \frac{1}{\rho^2} L_s \kappa_s^2 \left( \|\bar{\boldsymbol{w}} - \boldsymbol{w}_0\|^2 + \frac{4}{3} \right)}{\varepsilon} \\
&\overset{(b)}{=} 1 + \frac{1}{\log(\frac{1}{\omega})} \log\left( \frac{2\alpha^2 \frac{1}{\rho} L_s \kappa_s \left( \|\bar{\boldsymbol{w}} - \boldsymbol{w}_0\|^2 + \frac{4}{3} \right)}{\varepsilon} \right) \\
&\quad + \frac{B}{\alpha L_s} \frac{1}{L_s(\alpha - 1)} \frac{8\alpha^3 \frac{1}{\rho^2} L_s}{\varepsilon} \frac{L_s}{\nu_s} \kappa_s \left( \|\bar{\boldsymbol{w}} - \boldsymbol{w}_0\|^2 + \frac{4}{3} \right) \\
&= 1 + \frac{1}{\log(\frac{1}{\omega})} \log\left( \frac{2\alpha^2 \frac{1}{\rho} L_s \kappa_s \left( \|\bar{\boldsymbol{w}} - \boldsymbol{w}_0\|^2 + \frac{4}{3} \right)}{\varepsilon} \right) + \frac{8B\alpha^2 \frac{1}{\rho^2} \kappa_s \left( \|\bar{\boldsymbol{w}} - \boldsymbol{w}_0\|^2 + \frac{4}{3} \right)}{(\alpha - 1)\nu_s} \frac{1}{\varepsilon},
\end{aligned}
$$

where (a) follows from equation 33, and for (b) we recall that $\tau = \frac{\eta B}{C}$, $\eta = \frac{1}{\alpha L_s}$ and $C = L_s(\alpha - 1)$. Therefore, overall, the IFO complexity is in $\mathcal{O}(\frac{\kappa_s}{\nu_s \varepsilon})$.

$\square$

## F.4. Proof of Theorems 4.7 and 4.8

Our proof for Theorem 4.8 is similar to the one for Theorem 4.3, though we needed to refine some results from (de Vazelhes et al., 2022) to properly express the variance of the ZO gradient estimator and incorporate it into the telescopic sum. Before proving the main Theorem 4.8, below we provide several intermediary results needed for the proof of Theorem 4.8. Then, the proof of Theorem 4.3 will be provided in Section F.4.3.

### F.4.1. USEFUL LEMMAS

We first recall the following results from (de Vazelhes et al., 2022):

**Proposition F.5** (Proposition 1 (i) (de Vazelhes et al., 2022))**.** *Let us consider any support $F \subseteq [d]$ of size $s$ ($|F| = s$). For the Z0 gradient estimator $\boldsymbol{g}_t$ in Algorithm 3 at $\boldsymbol{w}_t$, with $q_t$ random directions, and random supports of size $s_2$, and*

*assuming that $R$ is $(L_{s_2}, s_2)$-RSS' , we have, with $[\boldsymbol{u}]_F$ denoting the hard thresholding of a vector $\boldsymbol{u}$ on $F$ (that is, we set all coordinates not in $F$ to 0):*

$$\|[\mathbb{E}\boldsymbol{g}_t]_F - [\nabla R(\boldsymbol{w}_t)]_F\|^2 \leq \varepsilon_\mu \mu^2 \tag{34}$$

*with $\varepsilon_\mu := L_{s_2}^2 sd$*

*Proof of Proposition F.5.* Proof in (de Vazelhes et al., 2022). $\qquad\square$

**Lemma F.6** (Lemma C.2 (de Vazelhes et al., 2022))**.** *For any $(L_{s_2}, s_2)$-RSS' function $R$, using the gradient estimator $\boldsymbol{g}_t$ defined in Algorithm 3 with $q_t = 1$, we have, for any support $F \subseteq [d]$, with $|F| = s$, and $F^c := [d] \setminus F$:*

$$\mathbb{E}\|[\boldsymbol{g}_t]_F\|^2 = \varepsilon_F \|[\nabla R(\boldsymbol{w}_t)]_F\|^2 + \varepsilon_{F^c} \|[\nabla R(\boldsymbol{w}_t)]_{F^c}\|^2 + \varepsilon_{abs}\mu^2 \tag{35}$$

*with:*
*(i) $\varepsilon_F := \frac{2d}{(s_2+2)}\left(\frac{(s-1)(s_2-1)}{d-1} + 3\right)$*
*(ii) $\varepsilon_{F^c} := \frac{2d}{(s_2+2)}\left(\frac{s(s_2-1)}{d-1}\right)$*
*(iii) $\varepsilon_{abs} := 2dL_s^2 s s_2 \left(\frac{(s-1)(s_2-1)}{d-1} + 1\right)$.*

*Proof of Lemma F.6.* Proof in (de Vazelhes et al., 2022). $\qquad\square$

We now use the above lemma to bound the variance of the zeroth-order gradient estimator $\boldsymbol{g}_t$.

**Lemma F.7.** *The gradient estimator $\boldsymbol{g}_t$ defined in Algorithm 3 verifies the following properties for any $q_t \in \mathbb{N}^*$:*

$$\mathbb{E}\|[\boldsymbol{g}_t]_F - \mathbb{E}[\boldsymbol{g}_t]_F\|^2 \leq \frac{\varepsilon_F}{q_t}\|\nabla R(\boldsymbol{w})\|^2 + \frac{\varepsilon_{abs}}{q_t}\mu^2 \tag{36}$$

*with $\varepsilon_F$ and $\varepsilon_{abs}$ defined above in Lemma F.6*

*Proof of Lemma F.7.* If $q_t = 1$, we have:

$$
\begin{aligned}
\mathbb{E}\|[\boldsymbol{g}_t]_F - \mathbb{E}[\boldsymbol{g}_t]_F\|^2 &\overset{(a)}{=} \mathbb{E}\|[\boldsymbol{g}_t]_F\|^2 - \|[\mathbb{E}\boldsymbol{g}]_F\|^2 \\
&\leq \mathbb{E}\|[\boldsymbol{g}_t]_F\|^2 \\
&\overset{(35)}{\leq} \varepsilon_F\|[\nabla R(\boldsymbol{w})]_F\|^2 + \varepsilon_{F^c}\|[\nabla R(\boldsymbol{w})]_{F^c}\|^2 + \varepsilon_{\text{abs}}\mu^2 \\
&\overset{(b)}{\leq} \varepsilon_F\|\nabla R(\boldsymbol{w})\|^2 + \varepsilon_{abs}\mu^2,
\end{aligned}
$$

where (a) follows from the bias-variance formula $\mathbb{E}\|X - E[X]\|_2^2 = \mathbb{E}\|X\|_2^2 - \|\mathbb{E}X\|_2^2$ for a multidimensional random variable $X$, and (b) follows from the fact that

$$\varepsilon_F = \frac{2d}{s_2+2}\left(\frac{s(s_2-1)}{d-1} + 3 - \frac{s_2-1}{d}\right) > \frac{2d}{s_2+2}\left(\frac{s(s_2-1)}{d-1}\right) = \varepsilon_{F^c}$$

(since $s_2 \leq d$), and since $\|[\nabla R(\boldsymbol{w})]_F\|^2 + \|[\nabla R(\boldsymbol{w})]_{F^c}\|^2 = \|\nabla R(\boldsymbol{w})\|^2$ (by definition of the Euclidean norm).

Now, if $q_t \geq 1$, we know that the variance of an average of $q_t$ i.i.d. realizations of a random variable of total variance $\sigma^2$ is $\frac{\sigma^2}{q_t}$ (and its expected value remains the same by linearity of expectation): indeed, for any random multidimensional random

variable $X$, for which we consider the $q$ i.i.d. random variables $X_i$ of same distribution, we have:

$$\mathbb{E}\left\|\frac{1}{q_t}\sum_{i=1}^{q_t}X_i - \mathbb{E}\left[\frac{1}{q_t}\sum_{i=1}^{q_t}X_i\right]\right\|_2^2 = \mathbb{E}\left\|\frac{1}{q_t}\sum_{i=1}^{q_t}(X_i - \mathbb{E}X_i)\right\|_2^2$$

$$= \frac{1}{q_t^2}\left(\sum_{i=1}^{q_t}(X_i - \mathbb{E}X_i)\right)^\top\left(\sum_{i=1}^{q_t}(X_i - \mathbb{E}X_i)\right)$$

$$\stackrel{(a)}{=} \frac{1}{q_t^2}\sum_{i=1}^{q_t}\|X_i - \mathbb{E}X_i\|_2^2$$

$$= \frac{1}{q_t^2}\sum_{i=1}^{q_t}\|X - \mathbb{E}X\|_2^2$$

$$= \frac{1}{q_t^2}q_t\|X - \mathbb{E}X\|_2^2$$

$$= \frac{1}{q_t}\|X - \mathbb{E}X\|_2^2,$$

where (a) follows from the fact that $X_i$ are i.i.d hence for $i \neq j$: $\mathrm{Cov}(X_i, X_j) = \mathbb{E}(X_i - \mathbb{E}X_i)^\top(X_j - \mathbb{E}X_j) = 0$. Applying this to the random variable which realizations are $[g_t]_F$, this concludes the proof. $\qquad\square$

### F.4.2. PROOF OF THEOREM 4.7

Below we now first present some results (and their proofs) for the convergence of Algorithm 3 without the additional constraint, which is needed for the proof of Theorem 4.8, and also, as a byproduct, provides, up to our knowledge, the first convergence guarantee in objective value without system error for a zeroth-order hard-thresholding algorithm.

*Proof of Theorem 4.7.* Let us denote for simplicity: $C_1 := \frac{\varepsilon_F}{q_t}$, $C_2 := \frac{\varepsilon_{abs}}{q_t}$, and $C_3 := \varepsilon_\mu\mu^2$. Moreover, let us denote $F := \mathrm{supp}(w_t) \cup \mathrm{supp}(w_{t-1}) \cup \mathrm{supp}(\bar{w})$, where supp denotes the support of a vector, i.e. the set of coordinates of its non-zero components. Note that therefore we have $|F| \leq 2k + \bar{k} \leq 3k$. In addition $[u]_F$ denotes the thresholding of $u$ to the support $F$, that is, the vector $u$ with its components that are not in $F$ set to 0.

The fact that $R$ is $(L_{s'}, s')$-RSS', therefore also $(L_{s'}, s)$-RSS', implies from the remark in 4.5 that it is also $(L_{s'}, s)$-RSS, therefore:

$$R(w_t)$$

$$\leq R(w_{t-1}) + \langle\nabla R(w_{t-1}), w_t - w_{t-1}\rangle + \frac{L_{s'}}{2}\|w_t - w_{t-1}\|^2$$

$$= R(w_{t-1}) + \langle g_{t-1}, w_t - w_{t-1}\rangle + \frac{L_{s'}}{2}\|w_t - w_{t-1}\|^2 + \langle\nabla R(w_{t-1}) - g_{t-1}, w_t - w_{t-1}\rangle$$

$$= R(w_{t-1}) + \frac{1}{2\eta}\left[\|w_t - (w_{t-1} - \eta g_{t-1})\|^2 - \eta^2\|g_{t-1}\|^2 - \|w_t - w_{t-1}\|^2\right] + \frac{L_{s'}}{2}\|w_t - w_{t-1}\|^2$$

$$\quad + \langle\nabla R(w_{t-1}) - g_{t-1}, w_t - w_{t-1}\rangle$$

$$= R(w_{t-1}) + \frac{1}{2\eta}\|w_t - (w_{t-1} - \eta g_{t-1})\|^2 - \frac{\eta}{2}\|g_{t-1}\|^2 + \left[\frac{L_{s'} - \frac{1}{\eta}}{2}\right]\|w_t - w_{t-1}\|^2$$

$$\quad + \langle[\nabla R(w_{t-1}) - g_{t-1}]_F, w_t - w_{t-1}\rangle$$

$$\stackrel{(a)}{\leq} R(w_{t-1}) + \frac{1}{2\eta}\left[\|\bar{w} - (w_{t-1} - \eta g_{t-1})\|^2 - (1 - \sqrt{\beta})\|w_t - \bar{w}\|^2\right] - \frac{\eta}{2}\|g_{t-1}\|^2$$

$$\quad + \left[\frac{L_{s'} - \frac{1}{\eta}}{2}\right]\|w_t - w_{t-1}\|^2 + \langle[\nabla R(w_{t-1}) - g_{t-1}]_F, w_t - w_{t-1}\rangle$$

$$= R(w_{t-1}) + \frac{1}{2\eta}\left[\|\bar{w} - w_{t-1}\|^2 + \eta^2\|g_{t-1}\|^2 - 2\langle\eta g_{t-1}, w_{t-1} - \bar{w}\rangle\right] - \frac{1}{2\eta}(1 - \sqrt{\beta})\|w_t - \bar{w}\|^2$$

$$-\frac{\eta}{2}\|\boldsymbol{g}_{t-1}\|^2 + \left[\frac{L_{s'} - \frac{1}{\eta}}{2}\right]\|\boldsymbol{w}_t - \boldsymbol{w}_{t-1}\|^2 + \langle[\nabla R(\boldsymbol{w}_{t-1}) - \boldsymbol{g}_{t-1}]_F, \boldsymbol{w}_t - \boldsymbol{w}_{t-1}\rangle$$

$$=R(\boldsymbol{w}_{t-1}) + \frac{1}{2\eta}\left[\|\bar{\boldsymbol{w}} - \boldsymbol{w}_{t-1}\|^2 - 2\langle\eta\boldsymbol{g}_{t-1}, \boldsymbol{w}_{t-1} - \bar{\boldsymbol{w}}\rangle\right] - \frac{1}{2\eta}(1 - \sqrt{\beta})\|\boldsymbol{w}_t - \bar{\boldsymbol{w}}\|^2$$

$$+ \left[\frac{L_{s'} - \frac{1}{\eta}}{2}\right]\|\boldsymbol{w}_t - \boldsymbol{w}_{t-1}\|^2 + \langle[\nabla R(\boldsymbol{w}_{t-1}) - \boldsymbol{g}_{t-1}]_F, \boldsymbol{w}_t - \boldsymbol{w}_{t-1}\rangle$$

$$\overset{(b)}{=}R(\boldsymbol{w}_{t-1}) + \frac{1}{2\eta}\|\bar{\boldsymbol{w}} - \boldsymbol{w}_{t-1}\|^2 - \langle\boldsymbol{g}_{t-1}, \boldsymbol{w}_{t-1} - \bar{\boldsymbol{w}}\rangle - \frac{1}{2\eta}(1 - \sqrt{\beta})\|\boldsymbol{w}_t - \bar{\boldsymbol{w}}\|^2$$

$$+ \left[\frac{L_{s'} - \frac{1}{\eta} + C}{2}\right]\|\boldsymbol{w}_t - \boldsymbol{w}_{t-1}\|^2 + \frac{1}{2C}\|[\nabla R(\boldsymbol{w}_{t-1}) - \boldsymbol{g}_{t-1}]_F\|^2$$

$$=R(\boldsymbol{w}_{t-1}) + \frac{1}{2\eta}\|\bar{\boldsymbol{w}} - \boldsymbol{w}_{t-1}\|^2 - \langle\nabla R(\boldsymbol{w}_{t-1}), \boldsymbol{w}_{t-1} - \bar{\boldsymbol{w}}\rangle + \langle[\nabla R(\boldsymbol{w}_{t-1}) - \boldsymbol{g}_{t-1}]_F, \boldsymbol{w}_{t-1} - \bar{\boldsymbol{w}}\rangle$$

$$- \frac{1}{2\eta}(1 - \sqrt{\beta})\|\boldsymbol{w}_t - \bar{\boldsymbol{w}}\|^2 + \left[\frac{L_{s'} - \frac{1}{\eta} + C}{2}\right]\|\boldsymbol{w}_t - \boldsymbol{w}_{t-1}\|^2 + \frac{1}{2C}\|[\nabla R(\boldsymbol{w}_{t-1}) - \boldsymbol{g}_{t-1}]_F\|^2,$$

where (a) follows from Lemma 3.2 and (b) follows from the inequality $\langle a, b\rangle \leq \frac{C}{2}a^2 + \frac{1}{2C}b^2$, for any $(a, b) \in (\mathbb{R}^d)^2$ with $C > 0$ an arbitrary strictly positive constant.

Let us now choose $\eta := \frac{1}{L_{s'} + C}$: therefore the term $\left[\frac{L_{s'} - \frac{1}{\eta} + C}{2}\right]\|\boldsymbol{w}_t - \boldsymbol{w}_{t-1}\|^2$ above is 0. We now take the conditional expectation (conditioned on $\mathrm{w}_{t-1}$, which is the random variable which realizations are $\boldsymbol{w}_{t-1}$), on both sides, and from Lemma F.3 we obtain the inequality below (we slightly abuse notations and denote $\mathbb{E}[\cdot|\mathrm{w}_{t-1} = \boldsymbol{w}_{t-1}]$ by $\mathbb{E}[\cdot|\boldsymbol{w}_{t-1}]$):

$$\mathbb{E}[R(\boldsymbol{w}_t)|\boldsymbol{w}_{t-1}]$$

$$\leq R(\boldsymbol{w}_{t-1}) + \frac{1}{2\eta}\|\bar{\boldsymbol{w}} - \boldsymbol{w}_{t-1}\|^2 - \langle\nabla R(\boldsymbol{w}_{t-1}), \boldsymbol{w}_{t-1} - \bar{\boldsymbol{w}}\rangle$$

$$- \frac{1}{2\eta}(1 - \sqrt{\beta})\mathbb{E}\left[\|\boldsymbol{w}_t - \bar{\boldsymbol{w}}\|^2|\boldsymbol{w}_{t-1}\right] + \langle[\nabla R(\boldsymbol{w}_{t-1}) - \mathbb{E}[\boldsymbol{g}_{t-1}|\boldsymbol{w}_{t-1}]]_F, \boldsymbol{w}_{t-1} - \bar{\boldsymbol{w}}\rangle$$

$$+ \mathbb{E}\left[\frac{1}{2C}\|[\nabla R(\boldsymbol{w}_{t-1}) - \boldsymbol{g}_{t-1}]_F\|^2|\boldsymbol{w}_{t-1}\right]$$

$$\overset{(a)}{\leq} R(\boldsymbol{w}_{t-1}) + \frac{1}{2\eta}\|\bar{\boldsymbol{w}} - \boldsymbol{w}_{t-1}\|^2 - \langle\nabla R(\boldsymbol{w}_{t-1}), \boldsymbol{w}_{t-1} - \bar{\boldsymbol{w}}\rangle$$

$$- \frac{1}{2\eta}(1 - \sqrt{\beta})\mathbb{E}\left[\|\boldsymbol{w}_t - \bar{\boldsymbol{w}}\|^2|\boldsymbol{w}_{t-1}\right] + \frac{G}{2}\|\nabla R(\boldsymbol{w}_{t-1}) - \mathbb{E}[\boldsymbol{g}_{t-1}|\boldsymbol{w}_{t-1}]\|_F^2$$

$$+ \frac{1}{2G}\|\boldsymbol{w}_{t-1} - \bar{\boldsymbol{w}}\|^2 + \frac{1}{2C}\mathbb{E}\left[\|\nabla R(\boldsymbol{w}_{t-1}) - \boldsymbol{g}_{t-1}\|^2|\boldsymbol{w}_{t-1}\right]$$

$$=R(\boldsymbol{w}_{t-1}) + \left[\frac{1}{2\eta} + \frac{1}{2G}\right]\|\bar{\boldsymbol{w}} - \boldsymbol{w}_{t-1}\|^2 - \langle\nabla R(\boldsymbol{w}_{t-1}), \boldsymbol{w}_{t-1} - \bar{\boldsymbol{w}}\rangle$$

$$- \frac{1}{2\eta}(1 - \sqrt{\beta})\mathbb{E}\left[\|\boldsymbol{w}_t - \bar{\boldsymbol{w}}\|^2|\boldsymbol{w}_{t-1}\right] + \frac{G}{2}\|\nabla R(\boldsymbol{w}_{t-1}) - \mathbb{E}[\boldsymbol{g}_{t-1}|\boldsymbol{w}_{t-1}]\|_F^2$$

$$+ \frac{1}{2C}\mathbb{E}\left[\|[\nabla R(\boldsymbol{w}_{t-1}) - \boldsymbol{g}_{t-1}]_F\|^2|\boldsymbol{w}_{t-1}\right]$$

$$\overset{(b)}{\leq} R(\boldsymbol{w}_{t-1}) + \left[\frac{1}{2\eta} + \frac{1}{2G}\right]\|\bar{\boldsymbol{w}} - \boldsymbol{w}_{t-1}\|^2 - \langle\nabla R(\boldsymbol{w}_{t-1}), \boldsymbol{w}_{t-1} - \bar{\boldsymbol{w}}\rangle$$

$$- \frac{1}{2\eta}(1 - \sqrt{\beta})\mathbb{E}\left[\|\boldsymbol{w}_t - \bar{\boldsymbol{w}}\|^2|\boldsymbol{w}_{t-1}\right] + \frac{G}{2}\|\nabla R(\boldsymbol{w}_{t-1}) - \mathbb{E}[\boldsymbol{g}_{t-1}|\boldsymbol{w}_{t-1}]\|_F^2$$

$$+ \frac{1}{2C}\left(2\|[\nabla R(\boldsymbol{w}_{t-1}) - \mathbb{E}[\boldsymbol{g}_{t-1}|\boldsymbol{w}_{t-1}]]_F\|^2 + 2\|[\boldsymbol{g}_{t-1} - \mathbb{E}[\boldsymbol{g}_{t-1}|\boldsymbol{w}_{t-1}]]_F\|^2\right)$$

$$\overset{(34)+(36)}{\leq} R(\boldsymbol{w}_{t-1}) + \left[\frac{1}{2\eta} + \frac{1}{2G}\right] \|\bar{\boldsymbol{w}} - \boldsymbol{w}_{t-1}\|^2 - \langle \nabla R(\boldsymbol{w}_{t-1}), \boldsymbol{w}_{t-1} - \bar{\boldsymbol{w}} \rangle$$

$$- \frac{1}{2\eta}(1 - \sqrt{\beta})\mathbb{E}\left[\|\boldsymbol{w}_t - \bar{\boldsymbol{w}}\|^2 | \boldsymbol{w}_{t-1}\right] + \frac{G}{2}C_3$$

$$+ \frac{1}{2C}\left(2C_3 + 2C_1\|\nabla R(\boldsymbol{w}_{t-1})\|^2 + 2C_2\mu^2\right)$$

$$\overset{(c)}{\leq} R(\boldsymbol{w}_{t-1}) + \left[\frac{1}{2\eta} + \frac{1}{2G}\right] \|\bar{\boldsymbol{w}} - \boldsymbol{w}_{t-1}\|^2 - \langle \nabla R(\boldsymbol{w}_{t-1}), \boldsymbol{w}_{t-1} - \bar{\boldsymbol{w}} \rangle$$

$$- \frac{1}{2\eta}(1 - \sqrt{\beta})\mathbb{E}\left[\|\boldsymbol{w}_t - \bar{\boldsymbol{w}}\|^2 | \boldsymbol{w}_{t-1}\right] + \frac{G}{2}C_3$$

$$+ \frac{1}{2C}\left(2C_1\left(2\|\nabla R(\boldsymbol{w}_{t-1}) - \nabla R(\bar{\boldsymbol{w}})\|^2 + 2\|\nabla R(\bar{\boldsymbol{w}})\|^2\right) + 2C_2\mu^2 + 2C_3\right)$$

$$\overset{(d)}{\leq} R(\boldsymbol{w}_{t-1}) + \left[\frac{1}{2\eta} + \frac{1}{2G}\right] \|\bar{\boldsymbol{w}} - \boldsymbol{w}_{t-1}\|^2 - \langle \nabla R(\boldsymbol{w}_{t-1}), \boldsymbol{w}_{t-1} - \bar{\boldsymbol{w}} \rangle$$

$$- \frac{1}{2\eta}(1 - \sqrt{\beta})\mathbb{E}\left[\|\boldsymbol{w}_t - \bar{\boldsymbol{w}}\|^2 | \boldsymbol{w}_{t-1}\right] + \frac{G}{2}C_3$$

$$+ \frac{1}{2C}\left(2C_1\left(2L_{s'}^2\|\boldsymbol{w}_{t-1} - \bar{\boldsymbol{w}}\|^2 + 2\|\nabla R(\bar{\boldsymbol{w}})\|^2\right) + 2C_2\mu^2 + 2C_3\right)$$

$$= R(\boldsymbol{w}_{t-1}) + \left[\frac{1}{2\eta} + \frac{1}{2G} + \frac{2C_1 L_{s'}^2}{C}\right] \|\bar{\boldsymbol{w}} - \boldsymbol{w}_{t-1}\|^2 - \langle \nabla R(\boldsymbol{w}_{t-1}), \boldsymbol{w}_{t-1} - \bar{\boldsymbol{w}} \rangle$$

$$- \frac{1}{2\eta}(1 - \sqrt{\beta})\mathbb{E}\left[\|\boldsymbol{w}_t - \bar{\boldsymbol{w}}\|^2 | \boldsymbol{w}_{t-1}\right] + \frac{G}{2}C_3 + \frac{1}{C}\left(2C_1\|\nabla R(\bar{\boldsymbol{w}})\|^2 + C_2\mu^2 + C_3\right)$$

$$\overset{(e)}{\leq} R(\boldsymbol{w}_{t-1}) + \left[\frac{1}{2\eta} + \frac{1}{2G} + \frac{2C_1 L_{s'}^2}{C}\right] \|\bar{\boldsymbol{w}} - \boldsymbol{w}_{t-1}\|^2 + \left[R(\bar{\boldsymbol{w}}) - R(\boldsymbol{w}_{t-1}) - \frac{\nu_s}{2}\|\boldsymbol{w}_{t-1} - \bar{\boldsymbol{w}}\|^2\right]$$

$$- \frac{1}{2\eta}(1 - \sqrt{\beta})\mathbb{E}\left[\|\boldsymbol{w}_t - \bar{\boldsymbol{w}}\|^2 | \boldsymbol{w}_{t-1}\right] + \frac{G}{2}C_3 + \frac{1}{C}\left(2C_1\|\nabla R(\bar{\boldsymbol{w}})\|^2 + C_2\mu^2 + C_3\right)$$

$$= R(\bar{\boldsymbol{w}}) + \left[\frac{\frac{1}{\eta} - \nu_s}{2} + \frac{1}{2G} + \frac{2C_1 L_{s'}^2}{C}\right] \|\bar{\boldsymbol{w}} - \boldsymbol{w}_{t-1}\|^2$$

$$- \frac{1}{2\eta}(1 - \sqrt{\beta})\mathbb{E}\left[\|\boldsymbol{w}_t - \bar{\boldsymbol{w}}\|^2 | \boldsymbol{w}_{t-1}\right] + \frac{G}{2}C_3 + \frac{1}{C}\left(2C_1\|\nabla R(\bar{\boldsymbol{w}})\|^2 + C_2\mu^2 + C_3\right)$$

$$\overset{(f)}{\leq} R(\bar{\boldsymbol{w}}) + \left[\frac{\frac{1}{\eta} - \nu_s}{2} + \frac{1}{2G} + \frac{2\varepsilon_F L_{s'}^2}{\tau C}\right] \|\bar{\boldsymbol{w}} - \boldsymbol{w}_{t-1}\|^2$$

$$- \frac{1}{2\eta}(1 - \sqrt{\beta})\mathbb{E}\left[\|\boldsymbol{w}_t - \bar{\boldsymbol{w}}\|^2 | \boldsymbol{w}_{t-1}\right] + \frac{G}{2}C_3 + \frac{1}{C}\left(2C_1\|\nabla R(\bar{\boldsymbol{w}})\|^2 + C_2\mu^2 + C_3\right), \quad (37)$$

where (a) follows from the inequality $\langle a, b \rangle \leq \frac{G}{2}a^2 + \frac{1}{2G}b^2$, for any $(a, b) \in (\mathbb{R}^d)^2$ with $G > 0$ an arbitrary strictly positive constant, (b) and (c) follow from the inequality $\|a + b\|^2 \leq 2\|a\|^2 + 2\|b\|^2$ for any $(a, b) \in (\mathbb{R}^d)^2$, (d) follows from the fact that $R$ is $(L_{s'}, s')$-RSS' (Assumption 4.5 with sparsity level $s'$), therefore it is also $(L_{s'}, s_2)$-RSS', (e) follows from the RSC condition, and for (f), we recall that $C_1 = \frac{\varepsilon_F}{q_t}$, and we define $q_t = \left\lceil \frac{\tau}{\omega^t} \right\rceil$, for some $\omega > 1$ and $\tau > 0$ that will be chosen later in the proof. Recall that we have chosen $\eta = \frac{1}{L_{s'} + C}$. Let us define $\alpha := \frac{C}{L_{s'}} + 1$. Then $C = (\alpha - 1)L_{s'}$, and $\eta = \frac{1}{\alpha L_{s'}}$. Also recall that $\kappa_s = \frac{L_{s'}}{\nu_s}$.

We will now choose the constant $G$ and $C$, in order to simplify the inequality above, such that it matches as much as possible the structure of the previous proofs:

We will seek to rewrite:

$$\left[\frac{\frac{1}{\eta} - \nu_s}{2} + \frac{1}{2G} + \frac{2\frac{\varepsilon_F}{\tau}L_{s'}^2}{C}\right] \left(= \frac{1}{2\eta}\left[1 + \frac{1}{G\alpha L_{s'}} + \frac{4L_{s'}^2\frac{\varepsilon_F}{\tau}}{(\alpha-1)\alpha L_{s'}^2} - \frac{1}{\alpha\kappa_s}\right]\right), \text{ into :}$$

$$\frac{1}{2\eta}\left[1 - \frac{1}{\alpha'\kappa_s}\right] \text{ for some } \alpha' > 0 \text{ (we will seek } \alpha' \propto \alpha, \text{ with a dimensionless proportionality constant for simplicity)}.$$

Therefore, let us choose $G := \frac{4}{\nu_s}$, which implies:

$$\frac{1}{G\alpha L_{s'}} = \frac{1}{4\alpha\kappa_s}. \tag{38}$$

And let us choose $\tau := \frac{16\kappa_s\varepsilon_F}{(\alpha-1)}$, which implies:

$$\frac{4L_{s'}^2 \frac{\varepsilon_F}{\tau}}{(\alpha-1)\alpha L_{s'}^2} = \frac{1}{4\alpha\kappa_s}. \tag{39}$$

Therefore, using equations 38 and 39, we obtain:

$$\left[ \frac{\frac{1}{\eta} - \nu_s}{2} + \frac{1}{2G} + \frac{2\frac{\varepsilon_F}{\tau}L_{s'}^2}{C} \right] = \frac{1}{2\eta}\left[ 1 + \frac{1}{G\alpha L_{s'}} + \frac{4L_{s'}^2 \frac{\varepsilon_F}{\tau}}{(\alpha-1)\alpha L_{s'}^2} - \frac{1}{\alpha\kappa_s} \right]$$

$$= \frac{1}{2\eta}\left[ 1 + \frac{1}{4\alpha\kappa_s} + \frac{1}{4\alpha\kappa_s} - \frac{1}{\alpha\kappa_s} \right]$$

$$= \frac{1}{2\eta}\left[ 1 - \frac{1}{2\alpha\kappa_s} \right] = \frac{1}{2\eta}\left[ 1 - \frac{1}{\alpha'\kappa_s} \right],$$

where for simplicity we have denoted $\alpha' = 2\alpha$. We can therefore simplify (37) into:

$$\mathbb{E}[R(\boldsymbol{w}_t)|\boldsymbol{w}_{t-1}] - R(\bar{\boldsymbol{w}}) \leq \frac{1}{2\eta}\left[ \left(1 - \frac{1}{\alpha'\kappa_s}\right)\|\bar{\boldsymbol{w}} - \boldsymbol{w}_{t-1}\|^2 - (1 - \sqrt{\beta})\mathbb{E}\left[\|\boldsymbol{w}_t - \bar{\boldsymbol{w}}\|^2|\boldsymbol{w}_{t-1}\right] \right.$$

$$\left. + 2\eta\left(\frac{G}{2}C_3 + \frac{1}{C}\left(2C_1\|\nabla R(\bar{\boldsymbol{w}})\|^2 + C_2\mu^2 + C_3\right)\right) \right].$$

We now take the expectation over $\mathrm{w}_{t-1}$ of the above inequality (i.e. we take $\mathbb{E}_{\mathrm{w}_{t-1}}[\cdot]$): using the law of total expectation ($\mathbb{E}[\cdot] = \mathbb{E}_{\mathrm{w}_{t-1}}[\mathbb{E}[\cdot|\boldsymbol{w}_{t-1}]]$) we obtain:

$$\mathbb{E}R(\boldsymbol{w}_t) - R(\bar{\boldsymbol{w}}) \leq \frac{1}{2\eta}\left[ \left(1 - \frac{1}{\alpha'\kappa_s}\right)\mathbb{E}\|\bar{\boldsymbol{w}} - \boldsymbol{w}_{t-1}\|^2 - (1 - \sqrt{\beta})\mathbb{E}\|\boldsymbol{w}_t - \bar{\boldsymbol{w}}\|^2 \right. \tag{40}$$

$$\left. + 2\eta\left(\frac{G}{2}C_3 + \frac{1}{C}\left(2C_1\|\nabla R(\bar{\boldsymbol{w}})\|^2 + C_2\mu^2 + C_3\right)\right) \right] \tag{41}$$

Let us call $A := 2\eta\left(\frac{G}{2}C_3 + \frac{1}{C}\left(2C_1\|\nabla R(\bar{\boldsymbol{w}})\|^2 + C_2\mu^2 + C_3\right)\right)$ for simplicity. Similarly as in (Liu & Foygel Barber, 2020), we now take a weighted sum over $t = 1, ..., T$, to obtain:

$$\sum_{t=1}^{T} 2\eta \left(\frac{1 - \frac{1}{\alpha'\kappa_s}}{1 - \sqrt{\beta}}\right)^{T-t} \mathbb{E}[R(\boldsymbol{w}_t) - R(\bar{\boldsymbol{w}})]$$

$$\leq \sum_{t=1}^{T} \left(\frac{1 - \frac{1}{\alpha'\kappa_s}}{1 - \sqrt{\beta}}\right)^{T-t} \left[ \left(1 - \frac{1}{\alpha'\kappa_s}\right)\mathbb{E}\|\bar{\boldsymbol{w}} - \boldsymbol{w}_{t-1}\|^2 - (1 - \sqrt{\beta})\mathbb{E}\|\boldsymbol{w}_t - \bar{\boldsymbol{w}}\|^2 + A \right]$$

$$= \sum_{t=1}^{T} \left(\frac{1 - \frac{1}{\alpha'\kappa_s}}{1 - \sqrt{\beta}}\right)^{T-t} \left[ \left(1 - \frac{1}{\alpha'\kappa_s}\right)\mathbb{E}\|\bar{\boldsymbol{w}} - \boldsymbol{w}_{t-1}\|^2 - (1 - \sqrt{\beta})\mathbb{E}\|\boldsymbol{w}_t - \bar{\boldsymbol{w}}\|^2 \right]$$

$$+ \sum_{t=1}^{T} \left(\frac{1 - \frac{1}{\alpha'\kappa_s}}{1 - \sqrt{\beta}}\right)^{T-t} A$$

$$= (1 - \sqrt{\beta})\sum_{t=1}^{T} \left[ \left(\frac{1 - \frac{1}{\alpha'\kappa_s}}{1 - \sqrt{\beta}}\right)^{T-t+1} \mathbb{E}\|\bar{\boldsymbol{w}} - \boldsymbol{w}_{t-1}\|^2 - \left(\frac{1 - \frac{1}{\alpha'\kappa_s}}{1 - \sqrt{\beta}}\right)^{T-t} \mathbb{E}\|\boldsymbol{w}_t - \bar{\boldsymbol{w}}\|^2 \right]$$

$$+ \sum_{t=1}^{T} \left(\frac{1 - \frac{1}{\alpha'\kappa_s}}{1 - \sqrt{\beta}}\right)^{T-t} A$$

$$\overset{(a)}{=}(1-\sqrt{\beta})\left[\left(\frac{1-\frac{1}{\alpha'\kappa_s}}{1-\sqrt{\beta}}\right)^{T}\|\bar{\boldsymbol{w}}-\boldsymbol{w}_0\|^2 - \mathbb{E}\|\boldsymbol{w}_T-\bar{\boldsymbol{w}}\|^2\right]+\sum_{t=1}^{T}\left(\frac{1-\frac{1}{\alpha'\kappa_s}}{1-\sqrt{\beta}}\right)^{T-t}A$$

$$\leq(1-\sqrt{\beta})\left(\frac{1-\frac{1}{\alpha'\kappa_s}}{1-\sqrt{\beta}}\right)^{T}\|\bar{\boldsymbol{w}}-\boldsymbol{w}_0\|^2 + \sum_{t=1}^{T}\left(\frac{1-\frac{1}{\alpha'\kappa_s}}{1-\sqrt{\beta}}\right)^{T-t}A$$

$$\leq\left(\frac{1-\frac{1}{\alpha'\kappa_s}}{1-\sqrt{\beta}}\right)^{T}\|\bar{\boldsymbol{w}}-\boldsymbol{w}_0\|^2 + \sum_{t=1}^{T}\left(\frac{1-\frac{1}{\alpha'\kappa_s}}{1-\sqrt{\beta}}\right)^{T-t}A$$

$$=\left(\frac{1-\frac{1}{\alpha'\kappa_s}}{1-\sqrt{\beta}}\right)^{T}\|\bar{\boldsymbol{w}}-\boldsymbol{w}_0\|^2 + \sum_{t=1}^{T}\left(\frac{1-\frac{1}{\alpha'\kappa_s}}{1-\sqrt{\beta}}\right)^{T-t}2\eta\left(\frac{G}{2}C_3+\frac{1}{C}\left(2C_1\|\nabla R(\bar{\boldsymbol{w}})\|^2+C_2\mu^2+C_3\right)\right)$$

$$=\left(\frac{1-\frac{1}{\alpha'\kappa_s}}{1-\sqrt{\beta}}\right)^{T}\|\bar{\boldsymbol{w}}-\boldsymbol{w}_0\|^2 + \sum_{t=1}^{T}\left(\frac{1-\frac{1}{\alpha'\kappa_s}}{1-\sqrt{\beta}}\right)^{T-t}2\eta\left(\frac{G}{2}C_3+\frac{1}{C}\left(2\frac{\varepsilon_F}{q_t}\|\nabla R(\bar{\boldsymbol{w}})\|^2+\frac{\varepsilon_{abs}\mu^2}{q_t}+C_3\right)\right)$$

$$=\left(\frac{1-\frac{1}{\alpha'\kappa_s}}{1-\sqrt{\beta}}\right)^{T}\|\bar{\boldsymbol{w}}-\boldsymbol{w}_0\|^2 + \sum_{t=1}^{T}\left(\frac{1-\frac{1}{\alpha'\kappa_s}}{1-\sqrt{\beta}}\right)^{T-t}\frac{2\eta}{q_t}\left(\frac{2\varepsilon_F\|\nabla R(\bar{\boldsymbol{w}})\|^2+\varepsilon_{abs}\mu^2}{C}\right)$$

$$+\sum_{t=1}^{T}\left(\frac{1-\frac{1}{\alpha'\kappa_s}}{1-\sqrt{\beta}}\right)^{T-t}2\eta C_3\left(\frac{G}{2}+\frac{1}{C}\right)$$

$$=\left(\frac{1-\frac{1}{\alpha'\kappa_s}}{1-\sqrt{\beta}}\right)^{T}\|\bar{\boldsymbol{w}}-\boldsymbol{w}_0\|^2 + \sum_{t=1}^{T}\left(\frac{1-\frac{1}{\alpha'\kappa_s}}{1-\sqrt{\beta}}\right)^{T-t}\frac{2\eta}{q_t}\left(\frac{2\varepsilon_F\|\nabla R(\bar{\boldsymbol{w}})\|^2}{C}\right)$$

$$+\sum_{t=1}^{T}\left(\frac{1-\frac{1}{\alpha'\kappa_s}}{1-\sqrt{\beta}}\right)^{T-t}2\eta\mu^2\left(\varepsilon_\mu\left(\frac{G}{2}+\frac{1}{C}\right)+\frac{\varepsilon_{abs}}{Cq_t}\right)$$

$$\leq\left(\frac{1-\frac{1}{\alpha'\kappa_s}}{1-\sqrt{\beta}}\right)^{T}\|\bar{\boldsymbol{w}}-\boldsymbol{w}_0\|^2 + \sum_{t=1}^{T}\left(\frac{1-\frac{1}{\alpha'\kappa_s}}{1-\sqrt{\beta}}\right)^{T-t}\frac{2\eta}{q_t}\left(\frac{2\varepsilon_F\|\nabla R(\bar{\boldsymbol{w}})\|^2}{C}\right)$$

$$+\sum_{t=1}^{T}\left(\frac{1-\frac{1}{\alpha'\kappa_s}}{1-\sqrt{\beta}}\right)^{T-t}2\eta\mu^2\left(\varepsilon_\mu\left(\frac{G}{2}+\frac{1}{C}\right)+\frac{\varepsilon_{abs}}{C}\right), \tag{42}$$

where (a) follows from simplifying the telescopic sum. Let us denote for simplicity $\zeta := \frac{2\eta(2\varepsilon_F\|\nabla R(\bar{\boldsymbol{w}})\|^2)}{C} = \frac{4\eta\varepsilon_F\|\nabla R(\bar{\boldsymbol{w}})\|^2}{C}$ and $Z := \varepsilon_\mu\left(\frac{G}{2}+\frac{1}{C}\right)+\frac{\varepsilon_{abs}}{C}$.

We now choose $k$ and $q_t$ as follows: we choose $k \geq 4\alpha'^2\kappa_s^2\bar{k}$, which implies that:

$$\sqrt{\beta}\leq\frac{1}{2\alpha'\kappa_s}$$

$$\implies \sqrt{\beta}\leq\frac{1}{2\alpha'\kappa_s-1}$$

$$\implies 1-\sqrt{\beta}\geq1-\frac{1}{2\alpha'\kappa_s-1}=\frac{2\alpha'\kappa_s-2}{2\alpha'\kappa_s-1}=\frac{1-\frac{1}{\alpha'\kappa_s}}{1-\frac{1}{2\alpha'\kappa_s}}$$

$$\implies \left(\frac{1-\frac{1}{\alpha'\kappa_s}}{1-\sqrt{\beta}}\right)\leq1-\frac{1}{2\alpha'\kappa_s}. \tag{43}$$

We recall that we previously defined $q_t = \lceil\frac{\tau}{\omega^t}\rceil$, with $\tau := \frac{16\kappa_s\varepsilon_F}{(\alpha-1)}$. We now set the value of $\omega$, to $\omega := 1-\frac{1}{4\alpha'\kappa_s}$.

Let us call $\nu := 1-\frac{1}{2\alpha'\kappa_s}$. Note that we have:

$$\nu\leq\omega. \tag{44}$$

And that we have the inequality below:

$$\frac{\nu}{\omega} = \frac{1 - \frac{1}{2\alpha'\kappa_s}}{1 - \frac{1}{4\alpha'\kappa_s}} = \frac{4\alpha'\kappa_s - 2}{4\alpha'\kappa_s - 1} = 1 - \frac{1}{4\alpha'\kappa_s - 1} \leq 1 - \frac{1}{4\alpha'\kappa_s} = \omega. \tag{45}$$

This allows us to simplify equation 42 into:

$$\mathbb{E}\left[\sum_{t=1}^{T} 2\eta \left(\frac{1 - \frac{1}{\alpha'\kappa_s}}{1 - \sqrt{\beta}}\right)^{T-t} [R(\boldsymbol{w}_t) - R(\bar{\boldsymbol{w}})]\right]$$

$$\leq \nu^T\|\bar{\boldsymbol{w}} - \boldsymbol{w}_0\|^2 + \frac{\zeta}{\tau}\sum_{t=1}^{T}\nu^{T-t}\omega^{t-1} + \sum_{t=1}^{T}\left(\frac{1 - \frac{1}{\alpha'\kappa_s}}{1 - \sqrt{\beta}}\right)^{T-t} 2\eta Z\mu^2$$

$$= \nu^T\|\bar{\boldsymbol{w}} - \boldsymbol{w}_0\|^2 + \frac{\zeta}{\tau}\frac{\omega^T}{\omega}\sum_{t=1}^{T}\left(\frac{\nu}{\omega}\right)^{T-t} + \sum_{t=1}^{T}\left(\frac{1 - \frac{1}{\alpha'\kappa_s}}{1 - \sqrt{\beta}}\right)^{T-t} 2\eta Z\mu^2$$

$$= \nu^T\|\bar{\boldsymbol{w}} - \boldsymbol{w}_0\|^2 + \frac{\zeta}{\tau}\frac{\omega^T}{\omega}\frac{1 - \left(\frac{\nu}{\omega}\right)^T}{1 - \left(\frac{\nu}{\omega}\right)} + \sum_{t=1}^{T}\left(\frac{1 - \frac{1}{\alpha'\kappa_s}}{1 - \sqrt{\beta}}\right)^{T-t} 2\eta Z\mu^2$$

$$\leq \nu^T\|\bar{\boldsymbol{w}} - \boldsymbol{w}_0\|^2 + \frac{\zeta}{\tau}\frac{\omega^T}{\omega}\frac{1}{1 - \left(\frac{\nu}{\omega}\right)} + \sum_{t=1}^{T}\left(\frac{1 - \frac{1}{\alpha'\kappa_s}}{1 - \sqrt{\beta}}\right)^{T-t} 2\eta Z\mu^2$$

$$\overset{(a)}{\leq} \nu^T\|\bar{\boldsymbol{w}} - \boldsymbol{w}_0\|^2 + \frac{\zeta}{\tau}\frac{\omega^T}{\omega}\frac{1}{1 - \omega} + \sum_{t=1}^{T}\left(\frac{1 - \frac{1}{\alpha'\kappa_s}}{1 - \sqrt{\beta}}\right)^{T-t} 2\eta Z\mu^2$$

$$\overset{(b)}{\leq} \nu^T\|\bar{\boldsymbol{w}} - \boldsymbol{w}_0\|^2 + \frac{\zeta}{\tau}\frac{4}{3}\omega^T\frac{1}{1 - \omega} + \sum_{t=1}^{T}\left(\frac{1 - \frac{1}{\alpha'\kappa_s}}{1 - \sqrt{\beta}}\right)^{T-t} 2\eta Z\mu^2$$

$$\overset{(c)}{\leq} \omega^T\|\bar{\boldsymbol{w}} - \boldsymbol{w}_0\|^2 + \frac{\zeta}{\tau}\frac{4}{3}\omega^T\frac{1}{1 - \omega} + \sum_{t=1}^{T}\left(\frac{1 - \frac{1}{\alpha'\kappa_s}}{1 - \sqrt{\beta}}\right)^{T-t} 2\eta Z\mu^2$$

$$\overset{(d)}{\leq} \frac{\omega^T}{1 - \omega}\|\bar{\boldsymbol{w}} - \boldsymbol{w}_0\|^2 + \frac{\zeta}{\tau}\frac{4}{3}\omega^T\frac{1}{1 - \omega} + \sum_{t=1}^{T}\left(\frac{1 - \frac{1}{\alpha'\kappa_s}}{1 - \sqrt{\beta}}\right)^{T-t} 2\eta Z\mu^2$$

$$= \frac{\omega^T}{1 - \omega}\left(\|\bar{\boldsymbol{w}} - \boldsymbol{w}_0\|^2 + \frac{\zeta}{\tau}\frac{4}{3}\right) + \sum_{t=1}^{T}\left(\frac{1 - \frac{1}{\alpha'\kappa_s}}{1 - \sqrt{\beta}}\right)^{T-t} 2\eta Z\mu^2$$

$$= 4\alpha'\kappa_s\omega^T\left(\|\bar{\boldsymbol{w}} - \boldsymbol{w}_0\|^2 + \frac{\zeta}{\tau}\frac{4}{3}\right) + \sum_{t=1}^{T}\left(\frac{1 - \frac{1}{\alpha'\kappa_s}}{1 - \sqrt{\beta}}\right)^{T-t} 2\eta Z\mu^2,$$

where in the left hand side we have used the linearity of expectation, and where (a) uses equation 45, (b) uses the fact that $\frac{1}{\omega} = \frac{1}{1 - \frac{1}{4\alpha'\kappa_s}} \leq \frac{1}{1 - \frac{1}{4}} = \frac{4}{3}$ (since $\kappa_s \geq 1$ and $\alpha' \geq 1$ (indeed, we have $\alpha' = 2\alpha = 2(\frac{C}{L_{s'}} + 1)$ with $C > 0$)), (c) uses equation 44, and (d) uses the fact that $\omega < 1$ so $1 < \frac{1}{1-\omega}$.

Let us now normalize the above inequality:

$$\mathbb{E}\frac{\sum_{t=1}^{T} 2\eta \left(\frac{1 - \frac{1}{\alpha'\kappa_s}}{1 - \sqrt{\beta}}\right)^{T-t} R(\boldsymbol{w}_t)}{\sum_{t=1}^{T} 2\eta \left(\frac{1 - \frac{1}{\alpha'\kappa_s}}{1 - \sqrt{\beta}}\right)^{T-t}} \leq R(\bar{\boldsymbol{w}}) + \frac{4\alpha'\kappa_s\omega^T\left(\|\bar{\boldsymbol{w}} - \boldsymbol{w}_0\|^2 + \frac{4}{3}\frac{\zeta}{\tau}\right)}{\sum_{t=1}^{T} 2\eta \left(\frac{1 - \frac{1}{\alpha'\kappa_s}}{1 - \sqrt{\beta}}\right)^{T-t}} + Z\mu^2.$$

The left hand side above is a weighted sum, which is an upper bound on the smallest term of the sum.

Regarding the right hand side, we can simplify it using the fact that $0 < \left( \frac{1 - \frac{1}{\alpha' \kappa_s}}{1 - \sqrt{\beta}} \right)$ , and therefore:

$$\sum_{t=1}^{T} \left( \frac{1 - \frac{1}{\alpha' \kappa_s}}{1 - \sqrt{\beta}} \right)^{T-t} \geq 1.$$

Therefore, we obtain:

$$\mathbb{E} \min_{t \in \{1,..,T\}} R(\boldsymbol{w}_t) - R(\bar{\boldsymbol{w}}) \leq \frac{4\alpha' \kappa_s \omega^T \left( \|\bar{\boldsymbol{w}} - \boldsymbol{w}_0\|^2 + \frac{4}{3} \frac{\varsigma}{\tau} \right)}{2\eta} + Z\mu^2$$

$$= 4\alpha^2 L_{s'} \kappa_s \omega^T \left( \|\bar{\boldsymbol{w}} - \boldsymbol{w}_0\|^2 + \frac{4}{3} \frac{\varsigma}{\tau} \right) + Z\mu^2.$$

Which can be simplified into the expression below, using the definition of $\hat{\boldsymbol{w}}_T$:

$$\mathbb{E} R(\hat{\boldsymbol{w}}_T) - R(\bar{\boldsymbol{w}}) \leq 4\alpha^2 L_{s'} \kappa_s \omega^T \left( \|\bar{\boldsymbol{w}} - \boldsymbol{w}_0\|^2 + \frac{4}{3} \frac{\varsigma}{\tau} \right) + Z\mu^2. \tag{46}$$

To simplify the above result, we recall the assumptions made earlier on: we have chosen $\tau = \frac{16\kappa_s \varepsilon_F}{(\alpha - 1)}$, and $G = \frac{4}{\nu_s}$ .

Therefore, to sum up, we have:

$$Z = \varepsilon_\mu \left( \frac{G}{2} + \frac{1}{C} \right) + \frac{\varepsilon_{abs}}{C} = \varepsilon_\mu \left( \frac{2}{\nu_s} + \frac{1}{C} \right) + \frac{\varepsilon_{abs}}{C}.$$

$$\omega = 1 - \frac{1}{4\alpha' \kappa_s} = 1 - \frac{1}{8\alpha \kappa_s}$$

$$\varsigma = \frac{4\eta \varepsilon_F \|\nabla R(\bar{\boldsymbol{w}})\|^2}{C}$$

The last inequality implies: $\frac{\varsigma}{\tau} = \frac{\frac{4\eta \varepsilon_F \|\nabla R(\bar{\boldsymbol{w}})\|^2}{C}}{16\kappa_s L_{s'} \frac{\varepsilon_F}{C}} = \frac{\eta \|\nabla R(\bar{\boldsymbol{w}})\|^2}{4\kappa_s L_{s'}}.$ $\qquad \square$

**Corollary F.8.** *Additionally, the number of calls to the function $R$ (#IZO), and the number of hard thresholding operations (#HT) such that the upper bound in Theorem 4.3 above is smaller than $\varepsilon + Z\mu$, with $\varepsilon > 0$ are respectively: #HT $= \mathcal{O}(\kappa_s \log(\frac{1}{\varepsilon}))$ and #IZO $= \mathcal{O}\left( \frac{\varepsilon_F \kappa_s^3 L_s}{\varepsilon} \right)$. Note that if $s_2 = d$, we have $\varepsilon_F = \mathcal{O}(s) = \mathcal{O}(k)$, and therefore we obtain a query complexity that is dimension independent.*

*Proof of Corollary F.8.* Let $\varepsilon \in \mathbb{R}_+^*$. Let us find $T$ to ensure that $\mathbb{E} R(\hat{\boldsymbol{w}}_T) - R(\bar{\boldsymbol{w}}) \leq \varepsilon + Z\mu^2$ This will be enforced if:

$$4\alpha^2 L_{s'} \kappa_s \omega^T \left( \|\bar{\boldsymbol{w}} - \boldsymbol{w}_0\|^2 + \frac{4}{3} \frac{\eta \|\nabla R(\bar{\boldsymbol{w}})\|^2}{4\kappa_s L_{s'}} \right) \leq \varepsilon$$

$$\iff T \log(\omega) \leq \log \left( \frac{\varepsilon}{4\alpha^2 L_{s'} \kappa_s \left( \|\bar{\boldsymbol{w}} - \boldsymbol{w}_0\|^2 + \frac{4}{3} \frac{\eta \|\nabla R(\bar{\boldsymbol{w}})\|^2}{4\kappa_s L_{s'}} \right)} \right)$$

$$\iff T \geq \frac{1}{\log(\frac{1}{\omega})} \log \left( \frac{4\alpha^2 L_{s'} \kappa_s \left( \|\bar{\boldsymbol{w}} - \boldsymbol{w}_0\|^2 + \frac{4}{3} \frac{\eta \|\nabla R(\bar{\boldsymbol{w}})\|^2}{4\kappa_s L_{s'}} \right)}{\varepsilon} \right).$$

Therefore, let us take:

$$T := \left\lceil \frac{1}{\log(\frac{1}{\omega})} \log \left( \frac{4\alpha^2 L_{s'} \kappa_s \left( \|\bar{\boldsymbol{w}} - \boldsymbol{w}_0\|^2 + \frac{4}{3} \frac{\eta \|\nabla R(\bar{\boldsymbol{w}})\|^2}{4\kappa_s L_{s'}} \right)}{\varepsilon} \right) \right\rceil. \tag{47}$$

We can now derive the #IZO and #HT. First, we have one hard-thresholding operation at each iteration, therefore #HT= $T$. Using the fact that $\frac{1}{\log(\frac{1}{\omega})} = \frac{1}{-\log(\omega)} = \frac{1}{-\log(1-\frac{1}{8\alpha\kappa_s})} \leq \frac{1}{\frac{1}{8\alpha\kappa_s}} = 8\alpha\kappa_s$ (since by property of the logarithm, for all $x \in (-\infty, -1): \log(1-x) \leq -x$), and the fact that $\alpha = \frac{C}{L_{s'}}$ is independent of $\kappa_s$, we obtain that #HT $= \mathcal{O}(\kappa_s \log(\frac{1}{\varepsilon}))$.

We now turn to computing the #IZO. At each iteration $t$ we have $q_t$ function evaluations, therefore:

$$\text{\#IFO} = \sum_{t=0}^{T-1} q_t$$

$$\leq \sum_{t=0}^{T-1} \left( \frac{\tau}{\omega^t} + 1 \right)$$

$$= T + \tau \frac{\left(\frac{1}{\omega}\right)^T - 1}{\frac{1}{\omega} - 1}$$

$$\leq T + \frac{\tau}{\frac{1}{\omega} - 1} \left( \frac{1}{\omega} \right)^T$$

$$= T + \frac{\tau}{\frac{1}{\omega} - 1} \exp\left( T \log\left( \frac{1}{\omega} \right) \right)$$

$$\overset{(a)}{\leq} 1 + \frac{1}{\log(\frac{1}{\omega})} \log \left( \frac{4\alpha^2 L_{s'} \kappa_s \left( \|\bar{\boldsymbol{w}} - \boldsymbol{w}_0\|^2 + \frac{4}{3} \frac{\eta \|\nabla R(\bar{\boldsymbol{w}})\|^2}{4\kappa_s L_{s'}} \right)}{\varepsilon} \right)$$

$$+ \frac{\tau}{\frac{1}{\omega} - 1} \exp\left( \log\left( \frac{1}{\omega} \right) \left[ \frac{1}{\log(\frac{1}{\omega})} \log \left( \frac{4\alpha^2 L_{s'} \kappa_s \left( \|\bar{\boldsymbol{w}} - \boldsymbol{w}_0\|^2 + \frac{4}{3} \frac{\eta \|\nabla R(\bar{\boldsymbol{w}})\|^2}{4\kappa_s L_{s'}} \right)}{\varepsilon} \right) + 1 \right] \right)$$

$$= 1 + \frac{1}{\log(\frac{1}{\omega})} \log \left( \frac{4\alpha^2 L_{s'} \kappa_s \left( \|\bar{\boldsymbol{w}} - \boldsymbol{w}_0\|^2 + \frac{4}{3} \frac{\eta \|\nabla R(\bar{\boldsymbol{w}})\|^2}{4\kappa_s L_{s'}} \right)}{\varepsilon} \right) + \frac{\frac{\tau}{\omega}}{\frac{1}{\omega} - 1} \frac{4\alpha^2 L_{s'} \kappa_s \left( \|\bar{\boldsymbol{w}} - \boldsymbol{w}_0\|^2 + \frac{4}{3} \frac{\eta \|\nabla R(\bar{\boldsymbol{w}})\|^2}{4\kappa_s L_{s'}} \right)}{\varepsilon}$$

$$= 1 + \frac{1}{\log(\frac{1}{\omega})} \log \left( \frac{4\alpha^2 L_{s'} \kappa_s \left( \|\bar{\boldsymbol{w}} - \boldsymbol{w}_0\|^2 + \frac{4}{3} \frac{\eta \|\nabla R(\bar{\boldsymbol{w}})\|^2}{4\kappa_s L_{s'}} \right)}{\varepsilon} \right) + \frac{\tau}{1 - \omega} \frac{4\alpha^2 L_{s'} \kappa_s \left( \|\bar{\boldsymbol{w}} - \boldsymbol{w}_0\|^2 + \frac{4}{3} \frac{\eta \|\nabla R(\bar{\boldsymbol{w}})\|^2}{4\kappa_s L_{s'}} \right)}{\varepsilon}$$

$$= 1 + \frac{1}{\log(\frac{1}{\omega})} \log \left( \frac{4\alpha^2 L_{s'} \kappa_s \left( \|\bar{\boldsymbol{w}} - \boldsymbol{w}_0\|^2 + \frac{4}{3} \frac{\eta \|\nabla R(\bar{\boldsymbol{w}})\|^2}{4\kappa_s L_{s'}} \right)}{\varepsilon} \right) + \tau \frac{32\alpha^3 L_{s'} \kappa_s^2 \left( \|\bar{\boldsymbol{w}} - \boldsymbol{w}_0\|^2 + \frac{4}{3} \frac{\eta \|\nabla R(\bar{\boldsymbol{w}})\|^2}{4\kappa_s L_{s'}} \right)}{\varepsilon},$$

where (a) follows from equation 47.

And we recall that $\tau := \frac{16\kappa_s \varepsilon_F}{(\alpha-1)}$, which implies that:

$$\tau \frac{32\alpha^3 L_{s'} \kappa_s^2 \left( \|\bar{\boldsymbol{w}} - \boldsymbol{w}_0\|^2 + \frac{4}{3} \frac{\eta \|\nabla R(\bar{\boldsymbol{w}})\|^2}{2\gamma\kappa_s L_{s'}} \right)}{\varepsilon} = \mathcal{O}\left( \frac{\varepsilon_F}{\varepsilon} \left( \kappa_s^3 L_{s'} + \frac{\kappa_s}{\nu_s} \right) \right).$$

Therefore, overall, the # IZO complexity is in $\mathcal{O}\left( \frac{\varepsilon_F}{\varepsilon} \kappa_s^3 L_{s'} \right)$.

$\square$

### F.4.3. PROOF OF THEOREM 4.8

Using the results above, we can now proceed to the proof of Theorem 4.8.

*Proof of Theorem 4.8.* Let us denote for simplicity: $C_1 := \frac{\varepsilon_F}{q_t}$, $C_2 := \frac{\varepsilon_{abs}}{q_t}$, and $C_3 := \varepsilon_\mu \mu^2$. Moreover, let us denote $F := \operatorname{supp}(\boldsymbol{w}_t) \cup \operatorname{supp}(\boldsymbol{w}_{t-1}) \cup \operatorname{supp}(\bar{\boldsymbol{w}})$, where supp denotes the support of a vector, i.e. the set of coordinates of its non-zero components. Note that therefore we have $|F| \leq 2k + \bar{k} \leq 3k$. In addition $[\boldsymbol{u}]_F$ denotes the thresholding of $\boldsymbol{u}$ to the support $F$, that is, the vector $\boldsymbol{u}$ with its components that are not in $F$ set to 0. Since $R$ is $L_{s'}$-RSS', with $s' = \max(s_2, s)$, $R$ is also $s$-RSS' and $s_2$-RSS', with Lipschitz constant $L_{s'}$.

Denote $\boldsymbol{v}_t = \mathcal{H}_k(\boldsymbol{w}_{t-1} - \eta \nabla R(\boldsymbol{w}_{t-1}))$ for any $t \in \mathbb{N}$. The fact that $R$ is $(L_{s'}, s')$-RSS', therefore also $(L_{s'}, s)$-RSS', implies from the remark in Assumption 4.5 that it is also $(L_{s'}, s)$-RSS, therefore:

$$R(\boldsymbol{w}_t)$$

$$\leq R(\boldsymbol{w}_{t-1}) + \langle \nabla R(\boldsymbol{w}_{t-1}), \boldsymbol{w}_t - \boldsymbol{w}_{t-1} \rangle + \frac{L_s}{2}\|\boldsymbol{w}_t - \boldsymbol{w}_{t-1}\|^2$$

$$= R(\boldsymbol{w}_{t-1}) + \langle \boldsymbol{g}_{t-1}, \boldsymbol{w}_t - \boldsymbol{w}_{t-1} \rangle + \frac{L_s}{2}\|\boldsymbol{w}_t - \boldsymbol{w}_{t-1}\|^2 + \langle \nabla R(\boldsymbol{w}_{t-1}) - \boldsymbol{g}_{t-1}, \boldsymbol{w}_t - \boldsymbol{w}_{t-1} \rangle$$

$$= R(\boldsymbol{w}_{t-1}) + \frac{1}{2\eta}\left[\|\boldsymbol{w}_t - (\boldsymbol{w}_{t-1} - \eta \boldsymbol{g}_{t-1})\|^2 - \eta^2\|\boldsymbol{g}_{t-1}\|^2 - \|\boldsymbol{w}_t - \boldsymbol{w}_{t-1}\|^2\right] + \frac{L_s}{2}\|\boldsymbol{w}_t - \boldsymbol{w}_{t-1}\|^2$$

$$\quad + \langle \nabla R(\boldsymbol{w}_{t-1}) - \boldsymbol{g}_{t-1}, \boldsymbol{w}_t - \boldsymbol{w}_{t-1} \rangle$$

$$= R(\boldsymbol{w}_{t-1}) + \frac{1}{2\eta}\|\boldsymbol{w}_t - (\boldsymbol{w}_{t-1} - \eta \boldsymbol{g}_{t-1})\|^2 - \frac{\eta}{2}\|\boldsymbol{g}_{t-1}\|^2 + \left[\frac{L_s - \frac{1}{\eta}}{2}\right]\|\boldsymbol{w}_t - \boldsymbol{w}_{t-1}\|^2$$

$$\quad + \langle [\nabla R(\boldsymbol{w}_{t-1}) - \boldsymbol{g}_{t-1}]_F, \boldsymbol{w}_t - \boldsymbol{w}_{t-1} \rangle$$

$$\overset{(a)}{\leq} R(\boldsymbol{w}_{t-1}) + \frac{1}{2\eta}\left[\|\bar{\boldsymbol{w}} - (\boldsymbol{w}_{t-1} - \eta \boldsymbol{g}_{t-1})\|^2 - \|\boldsymbol{w}_t - \bar{\boldsymbol{w}}\|^2 + \sqrt{\beta}\|\boldsymbol{v}_t - \bar{\boldsymbol{w}}\|^2\right] - \frac{\eta}{2}\|\boldsymbol{g}_{t-1}\|^2$$

$$\quad + \left[\frac{L_s - \frac{1}{\eta}}{2}\right]\|\boldsymbol{w}_t - \boldsymbol{w}_{t-1}\|^2 + \langle [\nabla R(\boldsymbol{w}_{t-1}) - \boldsymbol{g}_{t-1}]_F, \boldsymbol{w}_t - \boldsymbol{w}_{t-1} \rangle$$

$$= R(\boldsymbol{w}_{t-1}) + \frac{1}{2\eta}\left[\|\bar{\boldsymbol{w}} - \boldsymbol{w}_{t-1}\|^2 + \eta^2\|\boldsymbol{g}_{t-1}\|^2 - 2\langle \eta \boldsymbol{g}_{t-1}, \boldsymbol{w}_{t-1} - \bar{\boldsymbol{w}} \rangle\right] - \frac{1}{2\eta}\|\boldsymbol{w}_t - \bar{\boldsymbol{w}}\|^2 + \frac{\sqrt{\beta}}{2\eta}\|\boldsymbol{v}_t - \bar{\boldsymbol{w}}\|^2$$

$$\quad - \frac{\eta}{2}\|\boldsymbol{g}_{t-1}\|^2 + \left[\frac{L_s - \frac{1}{\eta}}{2}\right]\|\boldsymbol{w}_t - \boldsymbol{w}_{t-1}\|^2 + \langle [\nabla R(\boldsymbol{w}_{t-1}) - \boldsymbol{g}_{t-1}]_F, \boldsymbol{w}_t - \boldsymbol{w}_{t-1} \rangle$$

$$= R(\boldsymbol{w}_{t-1}) + \frac{1}{2\eta}\left[\|\bar{\boldsymbol{w}} - \boldsymbol{w}_{t-1}\|^2 - 2\langle \eta \boldsymbol{g}_{t-1}, \boldsymbol{w}_{t-1} - \bar{\boldsymbol{w}} \rangle\right] - \frac{1}{2\eta}\|\boldsymbol{w}_t - \bar{\boldsymbol{w}}\|^2 + \frac{\sqrt{\beta}}{2\eta}\|\boldsymbol{v}_t - \bar{\boldsymbol{w}}\|^2$$

$$\quad + \left[\frac{L_s - \frac{1}{\eta}}{2}\right]\|\boldsymbol{w}_t - \boldsymbol{w}_{t-1}\|^2 + \langle [\nabla R(\boldsymbol{w}_{t-1}) - \boldsymbol{g}_{t-1}]_F, \boldsymbol{w}_t - \boldsymbol{w}_{t-1} \rangle$$

$$\overset{(b)}{=} R(\boldsymbol{w}_{t-1}) + \frac{1}{2\eta}\|\bar{\boldsymbol{w}} - \boldsymbol{w}_{t-1}\|^2 - \langle \boldsymbol{g}_{t-1}, \boldsymbol{w}_{t-1} - \bar{\boldsymbol{w}} \rangle - \frac{1}{2\eta}\|\boldsymbol{w}_t - \bar{\boldsymbol{w}}\|^2 + \frac{\sqrt{\beta}}{2\eta}\|\boldsymbol{v}_t - \bar{\boldsymbol{w}}\|^2$$

$$\quad + \left[\frac{L_s - \frac{1}{\eta} + C}{2}\right]\|\boldsymbol{w}_t - \boldsymbol{w}_{t-1}\|^2 + \frac{1}{2C}\|[\nabla R(\boldsymbol{w}_{t-1}) - \boldsymbol{g}_{t-1}]_F\|^2$$

$$= R(\boldsymbol{w}_{t-1}) + \frac{1}{2\eta}\|\bar{\boldsymbol{w}} - \boldsymbol{w}_{t-1}\|^2 - \langle \nabla R(\boldsymbol{w}_{t-1}), \boldsymbol{w}_{t-1} - \bar{\boldsymbol{w}} \rangle + \langle \nabla R(\boldsymbol{w}_{t-1}) - \boldsymbol{g}_{t-1}, \boldsymbol{w}_{t-1} - \bar{\boldsymbol{w}} \rangle$$

$$\quad - -\frac{1}{2\eta}\|\boldsymbol{w}_t - \bar{\boldsymbol{w}}\|^2 + \frac{\sqrt{\beta}}{2\eta}\|\boldsymbol{v}_t - \bar{\boldsymbol{w}}\|^2 + \left[\frac{L_{s'} - \frac{1}{\eta} + C}{2}\right]\|\boldsymbol{w}_t - \boldsymbol{w}_{t-1}\|^2 + \frac{1}{2C}\|[\nabla R(\boldsymbol{w}_{t-1}) - \boldsymbol{g}_{t-1}]_F\|^2,$$

where (a) follows from Lemma 3.2 and (b) follows from the inequality $\langle a, b \rangle \leq \frac{C}{2}a^2 + \frac{1}{2C}b^2$, for any $(a, b) \in (\mathbb{R}^d)^2$ with $C > 0$ an arbitrary strictly positive constant.

Let us now assume that $\eta := \frac{1}{L_{s'}+C}$: therefore the term $\left[\frac{L_{s'}-\frac{1}{\eta}+C}{2}\right] \|\boldsymbol{w}_t - \boldsymbol{w}_{t-1}\|^2$ above is 0. We now take the conditional expectation (conditioned on $\mathsf{w}_{t-1}$, which is the random variable which realizations are $\boldsymbol{w}_{t-1}$), on both sides, and from Lemma F.3 we obtain the inequality below (we slightly abuse notations and denote $\mathbb{E}[\cdot|\mathsf{w}_{t-1} = \boldsymbol{w}_{t-1}]$ by $\mathbb{E}[\cdot|\boldsymbol{w}_{t-1}]$):

$$\mathbb{E}[R(\boldsymbol{w}_t)|\boldsymbol{w}_{t-1}]$$

$$\leq R(\boldsymbol{w}_{t-1}) + \frac{1}{2\eta}\|\bar{\boldsymbol{w}} - \boldsymbol{w}_{t-1}\|^2 - \langle \nabla R(\boldsymbol{w}_{t-1}), \boldsymbol{w}_{t-1} - \bar{\boldsymbol{w}} \rangle$$

$$- \frac{1}{2\eta}\mathbb{E}\left[\|\boldsymbol{w}_t - \bar{\boldsymbol{w}}\|^2|\boldsymbol{w}_{t-1}\right] + \frac{\sqrt{\beta}}{2\eta}\mathbb{E}\left[\|\boldsymbol{v}_t - \bar{\boldsymbol{w}}\|^2|\boldsymbol{w}_{t-1}\right] + \langle [\nabla R(\boldsymbol{w}_{t-1}) - \mathbb{E}[\boldsymbol{g}_{t-1}|\boldsymbol{w}_{t-1}]]_F, \boldsymbol{w}_{t-1} - \bar{\boldsymbol{w}} \rangle$$

$$+ \mathbb{E}\left[\frac{1}{2C}\|[\nabla R(\boldsymbol{w}_{t-1}) - \boldsymbol{g}_{t-1}]_F\|^2|\boldsymbol{w}_{t-1}\right]$$

$$\overset{(a)}{\leq} R(\boldsymbol{w}_{t-1}) + \frac{1}{2\eta}\|\bar{\boldsymbol{w}} - \boldsymbol{w}_{t-1}\|^2 - \langle \nabla R(\boldsymbol{w}_{t-1}), \boldsymbol{w}_{t-1} - \bar{\boldsymbol{w}} \rangle$$

$$- \frac{1}{2\eta}\mathbb{E}\left[\|\boldsymbol{w}_t - \bar{\boldsymbol{w}}\|^2|\boldsymbol{w}_{t-1}\right] + \frac{\sqrt{\beta}}{2\eta}\mathbb{E}\left[\|\boldsymbol{v}_t - \bar{\boldsymbol{w}}\|^2|\boldsymbol{w}_{t-1}\right] +$$

$$\frac{G}{2}\|[\nabla R(\boldsymbol{w}_{t-1}) - \mathbb{E}[\boldsymbol{g}_{t-1}|\boldsymbol{w}_{t-1}]]_F\|^2 + \frac{1}{2G}\|\boldsymbol{w}_{t-1} - \bar{\boldsymbol{w}}\|^2 + \frac{1}{2C}\mathbb{E}\left[\|\nabla R(\boldsymbol{w}_{t-1}) - \boldsymbol{g}_{t-1}\|^2|\boldsymbol{w}_{t-1}\right]$$

$$= R(\boldsymbol{w}_{t-1}) + \left[\frac{1}{2\eta} + \frac{1}{2G}\right]\|\bar{\boldsymbol{w}} - \boldsymbol{w}_{t-1}\|^2 - \langle \nabla R(\boldsymbol{w}_{t-1}), \boldsymbol{w}_{t-1} - \bar{\boldsymbol{w}} \rangle$$

$$- \frac{1}{2\eta}\mathbb{E}\left[\|\boldsymbol{w}_t - \bar{\boldsymbol{w}}\|^2|\boldsymbol{w}_{t-1}\right] + \frac{\sqrt{\beta}}{2\eta}\mathbb{E}\left[\|\boldsymbol{v}_t - \bar{\boldsymbol{w}}\|^2|\boldsymbol{w}_{t-1}\right] + \frac{G}{2}\|[\nabla R(\boldsymbol{w}_{t-1}) - \mathbb{E}[\boldsymbol{g}_{t-1}|\boldsymbol{w}_{t-1}]]_F\|^2$$

$$+ \frac{1}{2C}\mathbb{E}\left[\|[\nabla R(\boldsymbol{w}_{t-1}) - \boldsymbol{g}_{t-1}]_F\|^2|\boldsymbol{w}_{t-1}\right]$$

$$\overset{(b)}{\leq} R(\boldsymbol{w}_{t-1}) + \left[\frac{1}{2\eta} + \frac{1}{2G}\right]\|\bar{\boldsymbol{w}} - \boldsymbol{w}_{t-1}\|^2 - \langle \nabla R(\boldsymbol{w}_{t-1}), \boldsymbol{w}_{t-1} - \bar{\boldsymbol{w}} \rangle$$

$$- \frac{1}{2\eta}\mathbb{E}\left[\|\boldsymbol{w}_t - \bar{\boldsymbol{w}}\|^2|\boldsymbol{w}_{t-1}\right] + \frac{\sqrt{\beta}}{2\eta}\mathbb{E}\left[\|\boldsymbol{v}_t - \bar{\boldsymbol{w}}\|^2|\boldsymbol{w}_{t-1}\right] + \frac{G}{2}\|[\nabla R(\boldsymbol{w}_{t-1}) - \mathbb{E}[\boldsymbol{g}_{t-1}|\boldsymbol{w}_{t-1}]]_F\|^2$$

$$+ \frac{1}{2C}\left(2\|[\nabla R(\boldsymbol{w}_{t-1}) - \mathbb{E}[\boldsymbol{g}_{t-1}|\boldsymbol{w}_{t-1}]]_F\|^2 + 2\|[\boldsymbol{g}_{t-1} - \mathbb{E}[\boldsymbol{g}_{t-1}|\boldsymbol{w}_{t-1}]]_F\|^2\right)$$

$$\overset{(34)+(36)}{\leq} R(\boldsymbol{w}_{t-1}) + \left[\frac{1}{2\eta} + \frac{1}{2G}\right]\|\bar{\boldsymbol{w}} - \boldsymbol{w}_{t-1}\|^2 - \langle \nabla R(\boldsymbol{w}_{t-1}), \boldsymbol{w}_{t-1} - \bar{\boldsymbol{w}} \rangle$$

$$- \frac{1}{2\eta}\mathbb{E}\left[\|\boldsymbol{w}_t - \bar{\boldsymbol{w}}\|^2|\boldsymbol{w}_{t-1}\right] + \frac{\sqrt{\beta}}{2\eta}\mathbb{E}\left[\|\boldsymbol{v}_t - \bar{\boldsymbol{w}}\|^2|\boldsymbol{w}_{t-1}\right] + \frac{G}{2}C_3$$

$$+ \frac{1}{2C}\left(2C_3 + 2C_1\|\nabla R(\boldsymbol{w}_{t-1})\|^2 + 2C_2\mu^2\right)$$

$$\overset{(c)}{\leq} R(\boldsymbol{w}_{t-1}) + \left[\frac{1}{2\eta} + \frac{1}{2G}\right]\|\bar{\boldsymbol{w}} - \boldsymbol{w}_{t-1}\|^2 - \langle \nabla R(\boldsymbol{w}_{t-1}), \boldsymbol{w}_{t-1} - \bar{\boldsymbol{w}} \rangle$$

$$- \frac{1}{2\eta}\mathbb{E}\left[\|\boldsymbol{w}_t - \bar{\boldsymbol{w}}\|^2|\boldsymbol{w}_{t-1}\right] + \frac{\sqrt{\beta}}{2\eta}\mathbb{E}\left[\|\boldsymbol{v}_t - \bar{\boldsymbol{w}}\|^2|\boldsymbol{w}_{t-1}\right] + \frac{G}{2}C_3$$

$$+ \frac{1}{2C}\left(2C_1\left(2\|\nabla R(\boldsymbol{w}_{t-1}) - \nabla R(\bar{\boldsymbol{w}})\|^2 + 2\|\nabla R(\bar{\boldsymbol{w}})\|^2\right) + 2C_2\mu^2 + 2C_3\right)$$

$$\overset{(d)}{\leq} R(\boldsymbol{w}_{t-1}) + \left[\frac{1}{2\eta} + \frac{1}{2G}\right]\|\bar{\boldsymbol{w}} - \boldsymbol{w}_{t-1}\|^2 - \langle \nabla R(\boldsymbol{w}_{t-1}), \boldsymbol{w}_{t-1} - \bar{\boldsymbol{w}} \rangle$$

$$- \frac{1}{2\eta}\mathbb{E}\left[\|\boldsymbol{w}_t - \bar{\boldsymbol{w}}\|^2|\boldsymbol{w}_{t-1}\right] + \frac{\sqrt{\beta}}{2\eta}\mathbb{E}\left[\|\boldsymbol{v}_t - \bar{\boldsymbol{w}}\|^2|\boldsymbol{w}_{t-1}\right] + \frac{G}{2}C_3$$

$$+ \frac{1}{2C}\left(2C_1\left(2L_{s'}^2\|\boldsymbol{w}_{t-1} - \bar{\boldsymbol{w}}\|^2 + 2\|\nabla R(\bar{\boldsymbol{w}})\|^2\right) + 2C_2\mu^2 + 2C_3\right)$$

$$= R(\boldsymbol{w}_{t-1}) + \left[\frac{1}{2\eta} + \frac{1}{2G} + \frac{2C_1 L_{s'}^2}{C}\right] \|\bar{\boldsymbol{w}} - \boldsymbol{w}_{t-1}\|^2 - \langle \nabla R(\boldsymbol{w}_{t-1}), \boldsymbol{w}_{t-1} - \bar{\boldsymbol{w}} \rangle$$

$$- \frac{1}{2\eta} \mathbb{E}\left[\|\boldsymbol{w}_t - \bar{\boldsymbol{w}}\|^2 | \boldsymbol{w}_{t-1}\right] + \frac{\sqrt{\beta}}{2\eta} \mathbb{E}\left[\|\boldsymbol{v}_t - \bar{\boldsymbol{w}}\|^2 | \boldsymbol{w}_{t-1}\right] + \frac{G}{2}C_3 + \frac{1}{C}\left(2C_1 \|\nabla R(\bar{\boldsymbol{w}})\|^2 + C_2\mu^2 + C_3\right)$$

$$\overset{(e)}{\leq} R(\boldsymbol{w}_{t-1}) + \left[\frac{1}{2\eta} + \frac{1}{2G} + \frac{2C_1 L_{s'}^2}{C}\right] \|\bar{\boldsymbol{w}} - \boldsymbol{w}_{t-1}\|^2 + \left[R(\bar{\boldsymbol{w}}) - R(\boldsymbol{w}_{t-1}) - \frac{\nu_s}{2}\|\boldsymbol{w}_{t-1} - \bar{\boldsymbol{w}}\|^2\right]$$

$$- \frac{1}{2\eta} \mathbb{E}\left[\|\boldsymbol{w}_t - \bar{\boldsymbol{w}}\|^2 | \boldsymbol{w}_{t-1}\right] + \frac{\sqrt{\beta}}{2\eta} \mathbb{E}\left[\|\boldsymbol{v}_t - \bar{\boldsymbol{w}}\|^2 | \boldsymbol{w}_{t-1}\right] + \frac{G}{2}C_3 + \frac{1}{C}\left(2C_1 \|\nabla R(\bar{\boldsymbol{w}})\|^2 + C_2\mu^2 + C_3\right)$$

$$= R(\bar{\boldsymbol{w}}) + \left[\frac{\frac{1}{\eta} - \nu_s}{2} + \frac{1}{2G} + \frac{2C_1 L_{s'}^2}{C}\right] \|\bar{\boldsymbol{w}} - \boldsymbol{w}_{t-1}\|^2$$

$$- \frac{1}{2\eta} \mathbb{E}\left[\|\boldsymbol{w}_t - \bar{\boldsymbol{w}}\|^2 | \boldsymbol{w}_{t-1}\right] + \frac{\sqrt{\beta}}{2\eta} \mathbb{E}\left[\|\boldsymbol{v}_t - \bar{\boldsymbol{w}}\|^2 | \boldsymbol{w}_{t-1}\right] + \frac{G}{2}C_3 + \frac{1}{C}\left(2C_1 \|\nabla R(\bar{\boldsymbol{w}})\|^2 + C_2\mu^2 + C_3\right)$$

$$\overset{(f)}{\leq} R(\bar{\boldsymbol{w}}) + \left[\frac{\frac{1}{\eta} - \nu_s}{2} + \frac{1}{2G} + \frac{2\varepsilon_F L_{s'}^2}{\tau C}\right] \|\bar{\boldsymbol{w}} - \boldsymbol{w}_{t-1}\|^2$$

$$- \frac{1}{2\eta} \mathbb{E}\left[\|\boldsymbol{w}_t - \bar{\boldsymbol{w}}\|^2 | \boldsymbol{w}_{t-1}\right] + \frac{\sqrt{\beta}}{2\eta} \mathbb{E}\left[\|\boldsymbol{v}_t - \bar{\boldsymbol{w}}\|^2 | \boldsymbol{w}_{t-1}\right] + \frac{G}{2}C_3 + \frac{1}{C}\left(2C_1 \|\nabla R(\bar{\boldsymbol{w}})\|^2 + C_2\mu^2 + C_3\right) \quad (48)$$

Where (a) follows from the inequality $\langle a, b \rangle \leq \frac{G}{2}a^2 + \frac{1}{2G}b^2$, for any $(a, b) \in (\mathbb{R}^d)^2$ with $G > 0$ an arbitrary strictly positive constant, (b) and (c) follow from the inequality $\|a + b\|^2 \leq 2\|a\|^2 + 2\|b\|^2$ for any $(a, b) \in (\mathbb{R}^d)^2$, (d) follows from the fact that $R$ is $(L_{s'}, s')$-RSS' (Assumption 4.5 with sparsity level $s'$), therefore it is also $(L_{s'}, s)$-RSS', (e) follows from the RSC condition, and for (f), we recall that $C_1 = \frac{\varepsilon_F}{q_t}$, and we define $q_t = \left\lceil \frac{\tau}{\omega^t} \right\rceil$, for some $\omega > 1$ and $\tau > 0$ that will be chosen later in the proof.

Recall that we have chosen $\eta := \frac{1}{L_{s'} + C}$. Let us define $\alpha := \frac{C}{L_{s'}} + 1$. Then $C = (\alpha - 1)L_{s'}$, and $\eta = \frac{1}{\alpha L_{s'}}$. Also recall that $\kappa_s = \frac{L_{s'}}{\nu_s}$.

We will now choose the constant $G$ and $C$, in order to simplify the inequality above, such that it matches as much as possible the structure of the previous proofs:

We will seek to rewrite:

$$\left[\frac{\frac{1}{\eta} - \nu_s}{2} + \frac{1}{2G} + \frac{2\frac{\varepsilon_F}{\tau} L_{s'}^2}{C}\right] \left(= \frac{1}{2\eta}\left[1 + \frac{1}{G\alpha L_{s'}} + \frac{4L_{s'}^2 \frac{\varepsilon_F}{\tau}}{(\alpha - 1)\alpha L_{s'}^2} - \frac{1}{\alpha \kappa_s}\right]\right), \text{ into:}$$

$$\frac{1}{2\eta}\left[1 - \frac{1}{\alpha' \kappa_s}\right] \text{ for some } \alpha' > 0 \text{ (we will seek } \alpha' \propto \alpha, \text{ with a dimensionless proportionality constant for simplicity).}$$

Therefore, let us choose $G := \frac{4}{\nu_s}$, which implies:

$$\frac{1}{G\alpha L_{s'}} = \frac{1}{4\alpha \kappa_s}. \tag{49}$$

And let us choose $\tau := \frac{16\kappa_s \varepsilon_F}{(\alpha - 1)}$, which implies:

$$\frac{4L_{s'}^2 \frac{\varepsilon_F}{\tau}}{(\alpha - 1)\alpha L_{s'}^2} = \frac{1}{4\alpha \kappa_s}. \tag{50}$$

Therefore, using equations 49 and 50, we obtain:

$$\left[\frac{\frac{1}{\eta} - \nu_s}{2} + \frac{1}{2G} + \frac{2\frac{\varepsilon_F}{\tau} L_{s'}^2}{C}\right] = \frac{1}{2\eta}\left[1 + \frac{1}{G\alpha L_{s'}} + \frac{4L_{s'}^2 \frac{\varepsilon_F}{\tau}}{(\alpha - 1)\alpha L_{s'}^2} - \frac{1}{\alpha \kappa_s}\right]$$

$$= \frac{1}{2\eta}\left[1 + \frac{1}{4\alpha \kappa_s} + \frac{1}{4\alpha \kappa_s} - \frac{1}{\alpha \kappa_s}\right]$$

$$= \frac{1}{2\eta}\left[1 - \frac{1}{2\alpha\kappa_s}\right] = \frac{1}{2\eta}\left[1 - \frac{1}{\alpha'\kappa_s}\right],$$

where for simplicity we denote $\alpha' = 2\alpha$.

We can therefore simplify (48) into:

$$\mathbb{E}[R(\boldsymbol{w}_t)|\boldsymbol{w}_{t-1}] - R(\bar{\boldsymbol{w}}) \leq \frac{1}{2\eta}\left[\left(1 - \frac{1}{\alpha'\kappa_s}\right)\|\bar{\boldsymbol{w}} - \boldsymbol{w}_{t-1}\|^2 - \frac{1}{2\eta}\mathbb{E}\left[\|\boldsymbol{w}_t - \bar{\boldsymbol{w}}\|^2|\boldsymbol{w}_{t-1}\right]\right.$$
$$+ \frac{\sqrt{\beta}}{2\eta}\mathbb{E}\left[\|\boldsymbol{v}_t - \bar{\boldsymbol{w}}\|^2|\boldsymbol{w}_{t-1}\right]$$
$$\left.+ 2\eta\left(\frac{G}{2}C_3 + \frac{1}{C}\left(2C_1\|\nabla R(\bar{\boldsymbol{w}})\|^2 + C_2\mu^2 + C_3\right)\right)\right].$$

We now take the expectation over $\mathrm{w}_{t-1}$ of the above inequality (i.e. we take $\mathbb{E}_{\mathrm{w}_{t-1}}[\cdot]$): using the law of total expectation ($\mathbb{E}[\cdot] = \mathbb{E}_{\mathrm{w}_{t-1}}[\mathbb{E}[\cdot|\boldsymbol{w}_{t-1}]]$) we obtain:

$$\mathbb{E}R(\boldsymbol{w}_t) - R(\bar{\boldsymbol{w}}) \leq \frac{1}{2\eta}\left[\left(1 - \frac{1}{\alpha'\kappa_s}\right)\mathbb{E}\|\bar{\boldsymbol{w}} - \boldsymbol{w}_{t-1}\|^2 - \frac{1}{2\eta}\mathbb{E}\left[\|\boldsymbol{w}_t - \bar{\boldsymbol{w}}\|^2\right]\right. \tag{51}$$
$$+ \frac{\sqrt{\beta}}{2\eta}\mathbb{E}\left[\|\boldsymbol{v}_t - \bar{\boldsymbol{w}}\|^2\right] \tag{52}$$
$$\left.+ 2\eta\left(\frac{G}{2}C_3 + \frac{1}{C}\left(2C_1\|\nabla R(\bar{\boldsymbol{w}})\|^2 + C_2\mu^2 + C_3\right)\right)\right]. \tag{53}$$

Let us call $A := 2\eta\left(\frac{G}{2}C_3 + \frac{1}{C}\left(2C_1\|\nabla R(\bar{\boldsymbol{w}})\|^2 + C_2\mu^2 + C_3\right)\right)$ for simplicity.

This gives:

$$\mathbb{E}R(\boldsymbol{w}_t) - R(\bar{\boldsymbol{w}}) \leq \frac{1}{2\eta}\left[\left(1 - \frac{1}{\alpha'\kappa_s}\right)\mathbb{E}\|\bar{\boldsymbol{w}} - \boldsymbol{w}_{t-1}\|^2 - \frac{1}{2\eta}\mathbb{E}\|\boldsymbol{w}_t - \bar{\boldsymbol{w}}\|^2 + \frac{\sqrt{\beta}}{2\eta}\mathbb{E}\|\boldsymbol{v}_t - \bar{\boldsymbol{w}}\|^2 + A\right]. \tag{54}$$

Additionally, in view of equation 40 applied at $\boldsymbol{v}_t$ instead of $\boldsymbol{w}_t$, (since $\boldsymbol{v}_t$ here corresponds to the $\boldsymbol{w}_t$ from Section F.2.2, i.e. $\boldsymbol{v}_t$ is the hard-thresholding of an iterate after a gradient step), we know that:

$$\mathbb{E}R(\boldsymbol{v}_t) - R(\bar{\boldsymbol{w}}) \leq \frac{1}{2\eta}\left[\left(1 - \frac{1}{\alpha'\kappa_s}\right)\mathbb{E}\|\bar{\boldsymbol{w}} - \boldsymbol{w}_{t-1}\|^2 - (1 - \sqrt{\beta})\mathbb{E}\|\boldsymbol{w}_t - \bar{\boldsymbol{w}}\|^2 + A\right].$$

We now take a convex combination similarly as in the case without additional constraint (section E.2), for some $\rho \in (0, 1)$.

$$\mathbb{E}(1-\rho)R(\boldsymbol{w}_t) + \rho R(\boldsymbol{v}_t)$$
$$\leq R(\bar{\boldsymbol{w}}) + \frac{1}{2\eta}\left[\left(1 - \frac{1}{\alpha'\kappa_s}\right)\mathbb{E}\|\bar{\boldsymbol{w}} - \boldsymbol{w}_{t-1}\|^2 - (1-\rho)\mathbb{E}\|\boldsymbol{w}_t - \bar{\boldsymbol{w}}\|^2\right.$$
$$\left.+ \left((1-\rho)\sqrt{\beta} - (1-\sqrt{\beta})\rho\right)\mathbb{E}\|\boldsymbol{v}_t - \bar{\boldsymbol{w}}\|^2 + A\right]$$
$$= R(\bar{\boldsymbol{w}}) + \frac{1}{2\eta}\left[\left(1 - \frac{1}{\alpha'\kappa_s}\right)\mathbb{E}\|\bar{\boldsymbol{w}} - \boldsymbol{w}_{t-1}\|^2 - (1-\rho)\mathbb{E}\|\boldsymbol{w}_t - \bar{\boldsymbol{w}}\|^2\right.$$
$$\left.- \left(\rho - \sqrt{\beta}\right)\mathbb{E}\|\boldsymbol{v}_t - \bar{\boldsymbol{w}}\|^2 + A\right]$$
$$\overset{(b)}{\leq} R(\bar{\boldsymbol{w}}) + \frac{1}{2\eta}\left[\left(1 - \frac{1}{\alpha'\kappa_s}\right)\mathbb{E}\|\bar{\boldsymbol{w}} - \boldsymbol{w}_{t-1}\|^2 - (1-\rho)\mathbb{E}\|\boldsymbol{w}_t - \bar{\boldsymbol{w}}\|^2\right.$$
$$\left.- \left(\rho - \sqrt{\beta}\right)\mathbb{E}\|\boldsymbol{w}_t - \bar{\boldsymbol{w}}\|^2 + A\right]$$
$$= R(\bar{\boldsymbol{w}}) + \frac{1}{2\eta}\left[\left(1 - \frac{1}{\alpha'\kappa_s}\right)\mathbb{E}\|\bar{\boldsymbol{w}} - \boldsymbol{w}_{t-1}\|^2 - (1-\sqrt{\beta})\mathbb{E}\|\boldsymbol{w}_t - \bar{\boldsymbol{w}}\|^2 + A\right].$$

where in (b), we have assumed that $\sqrt{\beta} \leq \rho$ (later we will verify that our choice of $k$ ensures such a condition), and have used the fact that projection onto a convex set is non-expansive (which implies that $\|v_t - \bar{w}\|^2 \geq \|w_t - \bar{w}\|^2$).

Similarly as in (Liu & Foygel Barber, 2020), we now take a weighted sum over $t = 1, ..., T$, to obtain:

$$\sum_{t=1}^{T} 2\eta \left( \frac{1 - \frac{1}{\alpha'\kappa_s}}{1 - \sqrt{\beta}} \right)^{T-t} \mathbb{E}[(1-\rho)R(w_t) + \rho R(v_t) - R(\bar{w})]$$

$$\leq \sum_{t=1}^{T} \left( \frac{1 - \frac{1}{\alpha'\kappa_s}}{1 - \sqrt{\beta}} \right)^{T-t} \left[ \left(1 - \frac{1}{\alpha'\kappa_s}\right) \mathbb{E}\|\bar{w} - w_{t-1}\|^2 - (1-\sqrt{\beta})\mathbb{E}\|w_t - \bar{w}\|^2 + A \right]$$

$$= \sum_{t=1}^{T} \left( \frac{1 - \frac{1}{\alpha'\kappa_s}}{1 - \sqrt{\beta}} \right)^{T-t} \left[ \left(1 - \frac{1}{\alpha'\kappa_s}\right) \mathbb{E}\|\bar{w} - w_{t-1}\|^2 - (1-\sqrt{\beta})\mathbb{E}\|w_t - \bar{w}\|^2 \right]$$

$$+ \sum_{t=1}^{T} \left( \frac{1 - \frac{1}{\alpha'\kappa_s}}{1 - \sqrt{\beta}} \right)^{T-t} A$$

$$= (1-\sqrt{\beta}) \sum_{t=1}^{T} \left[ \left( \frac{1 - \frac{1}{\alpha'\kappa_s}}{1 - \sqrt{\beta}} \right)^{T-t+1} \mathbb{E}\|\bar{w} - w_{t-1}\|^2 - \left( \frac{1 - \frac{1}{\alpha'\kappa_s}}{1 - \sqrt{\beta}} \right)^{T-t} \mathbb{E}\|w_t - \bar{w}\|^2 \right]$$

$$+ \sum_{t=1}^{T} \left( \frac{1 - \frac{1}{\alpha'\kappa_s}}{1 - \sqrt{\beta}} \right)^{T-t} A$$

$$\overset{(a)}{=} (1-\sqrt{\beta}) \left[ \left( \frac{1 - \frac{1}{\alpha'\kappa_s}}{1 - \sqrt{\beta}} \right)^{T} \|\bar{w} - w_0\|^2 - \mathbb{E}\|w_T - \bar{w}\|^2 \right] + \sum_{t=1}^{T} \left( \frac{1 - \frac{1}{\alpha'\kappa_s}}{1 - \sqrt{\beta}} \right)^{T-t} A$$

$$\leq (1-\sqrt{\beta}) \left( \frac{1 - \frac{1}{\alpha'\kappa_s}}{1 - \sqrt{\beta}} \right)^{T} \|\bar{w} - w_0\|^2 + \sum_{t=1}^{T} \left( \frac{1 - \frac{1}{\alpha'\kappa_s}}{1 - \sqrt{\beta}} \right)^{T-t} A$$

$$\leq \left( \frac{1 - \frac{1}{\alpha'\kappa_s}}{1 - \sqrt{\beta}} \right)^{T} \|\bar{w} - w_0\|^2 + \sum_{t=1}^{T} \left( \frac{1 - \frac{1}{\alpha'\kappa_s}}{1 - \sqrt{\beta}} \right)^{T-t} A$$

$$= \left( \frac{1 - \frac{1}{\alpha'\kappa_s}}{1 - \sqrt{\beta}} \right)^{T} \|\bar{w} - w_0\|^2 + \sum_{t=1}^{T} \left( \frac{1 - \frac{1}{\alpha'\kappa_s}}{1 - \sqrt{\beta}} \right)^{T-t} 2\eta \left( \frac{G}{2}C_3 + \frac{1}{C}\left(2C_1\|\nabla R(\bar{w})\|^2 \right. \right.$$
$$\left. \left. + C_2\mu^2 + C_3 \right) \right)$$

$$= \left( \frac{1 - \frac{1}{\alpha'\kappa_s}}{1 - \sqrt{\beta}} \right)^{T} \|\bar{w} - w_0\|^2 + \sum_{t=1}^{T} \left( \frac{1 - \frac{1}{\alpha'\kappa_s}}{1 - \sqrt{\beta}} \right)^{T-t} 2\eta \left( \frac{G}{2}C_3 + \frac{1}{C}\left(2\frac{\varepsilon_F}{q_t}\|\nabla R(\bar{w})\|^2 \right. \right.$$
$$\left. \left. + \frac{\varepsilon_{abs}\mu^2}{q_t} + C_3 \right) \right)$$

$$= \left( \frac{1 - \frac{1}{\alpha'\kappa_s}}{1 - \sqrt{\beta}} \right)^{T} \|\bar{w} - w_0\|^2 + \sum_{t=1}^{T} \left( \frac{1 - \frac{1}{\alpha'\kappa_s}}{1 - \sqrt{\beta}} \right)^{T-t} \frac{2\eta}{q_t} \left( \frac{2\varepsilon_F\|\nabla R(\bar{w})\|^2 + \varepsilon_{abs}\mu^2}{C} \right)$$

$$+ \sum_{t=1}^{T} \left( \frac{1 - \frac{1}{\alpha'\kappa_s}}{1 - \sqrt{\beta}} \right)^{T-t} 2\eta C_3 \left( \frac{G}{2} + \frac{1}{C} \right)$$

$$= \left( \frac{1 - \frac{1}{\alpha'\kappa_s}}{1 - \sqrt{\beta}} \right)^{T} \|\bar{w} - w_0\|^2 + \sum_{t=1}^{T} \left( \frac{1 - \frac{1}{\alpha'\kappa_s}}{1 - \sqrt{\beta}} \right)^{T-t} \frac{2\eta}{q_t} \left( \frac{2\varepsilon_F\|\nabla R(\bar{w})\|^2}{C} \right)$$

$$+ \sum_{t=1}^{T} \left( \frac{1 - \frac{1}{\alpha'\kappa_s}}{1 - \sqrt{\beta}} \right)^{T-t} 2\eta\mu^2 \left( \varepsilon_\mu \left( \frac{G}{2} + \frac{1}{C} \right) + \frac{\varepsilon_{abs}}{Cq_t} \right)$$

$$\leq \left(\frac{1 - \frac{1}{\alpha' \kappa_s}}{1 - \sqrt{\beta}}\right)^T \|\bar{\boldsymbol{w}} - \boldsymbol{w}_0\|^2 + \sum_{t=1}^{T} \left(\frac{1 - \frac{1}{\alpha' \kappa_s}}{1 - \sqrt{\beta}}\right)^{T-t} \frac{2\eta}{q_t} \left(\frac{2\varepsilon_F \|\nabla R(\bar{\boldsymbol{w}})\|^2}{C}\right)$$

$$+ \sum_{t=1}^{T} \left(\frac{1 - \frac{1}{\alpha' \kappa_s}}{1 - \sqrt{\beta}}\right)^{T-t} 2\eta\mu^2 \left(\varepsilon_\mu \left(\frac{G}{2} + \frac{1}{C}\right) + \frac{\varepsilon_{abs}}{C}\right), \tag{55}$$

where (a) follows from simplifying the telescopic sum. Let us denote for simplicity $\zeta := \frac{2\eta(2\varepsilon_F \|\nabla R(\bar{\boldsymbol{w}})\|^2)}{C} = \frac{4\eta\varepsilon_F \|\nabla R(\bar{\boldsymbol{w}})\|^2}{C}$ and $Z := \varepsilon_\mu \left(\frac{G}{2} + \frac{1}{C}\right) + \frac{\varepsilon_{abs}}{C}$.

We now choose $k$ and $s_t$ as follows: we choose $k \geq 4\frac{\alpha'^2}{\rho}\kappa_s^2 \bar{k}$, which implies that:

$$\sqrt{\beta} \leq \frac{1}{2\frac{\alpha'}{\rho}\kappa_s}$$

$$\implies \sqrt{\beta} \leq \frac{1}{2\frac{\alpha'}{\rho}\kappa_s - 1}$$

$$\implies 1 - \sqrt{\beta} \geq 1 - \frac{1}{2\frac{\alpha'}{\rho}\kappa_s - 1} = \frac{2\frac{\alpha'}{\rho}\kappa_s - 2}{2\frac{\alpha'}{\rho}\kappa_s - 1} = \frac{1 - \frac{1}{\frac{\alpha'}{\rho}\kappa_s}}{1 - \frac{1}{2\frac{\alpha'}{\rho}\kappa_s}}$$

$$\implies \left(\frac{1 - \frac{1}{\frac{\alpha'}{\rho}\kappa_s}}{1 - \sqrt{\beta}}\right) \leq 1 - \frac{1}{2\frac{\alpha'}{\rho}\kappa_s}. \tag{56}$$

We recall that we previously defined $q_t = \left\lceil \frac{\tau}{\omega^t} \right\rceil$, with $\tau = 16\kappa_s \frac{\varepsilon_F}{(\alpha-1)}$. We now set the value of $\omega$, to $\omega := 1 - \frac{1}{\frac{\alpha'}{\rho}\kappa_s}$.

Let us call $\nu := 1 - \frac{1}{2\frac{\alpha'}{\rho}\kappa_s}$. Note that we have:

$$\nu \leq \omega. \tag{57}$$

And that we have the inequality below:

$$\frac{\nu}{\omega} = \frac{1 - \frac{1}{2\frac{\alpha'}{\rho}\kappa_s}}{1 - \frac{1}{4\frac{\alpha'}{\rho}\kappa_s}} = \frac{4\frac{\alpha'}{\rho}\kappa_s - 2}{4\frac{\alpha'}{\rho}\kappa_s - 1} = 1 - \frac{1}{4\frac{\alpha'}{\rho}\kappa_s - 1} \leq 1 - \frac{1}{4\frac{\alpha'}{\rho}\kappa_s} = \omega. \tag{58}$$

This allows us to simplify equation 55 into:

$$\mathbb{E} \sum_{t=1}^{T} 2\eta \left(\frac{1 - \frac{1}{\alpha' \kappa_s}}{1 - \sqrt{\beta}}\right)^{T-t} [(1-\rho)R(\boldsymbol{w}_t) + \rho R(\boldsymbol{v}_t) - R(\bar{\boldsymbol{w}})]$$

$$\leq \nu^T \|\bar{\boldsymbol{w}} - \boldsymbol{w}_0\|^2 + \sum_{t=1}^{T} \nu^{T-t} \omega^{t-1} + \frac{\zeta}{\tau} \sum_{t=1}^{T} \left(\frac{1 - \frac{1}{\alpha' \kappa_s}}{1 - \sqrt{\beta}}\right)^{T-t} 2\eta Z\mu^2$$

$$= \nu^T \|\bar{\boldsymbol{w}} - \boldsymbol{w}_0\|^2 + \frac{\omega^T}{\omega} \sum_{t=1}^{T} \left(\frac{\nu}{\omega}\right)^{T-t} + \frac{\zeta}{\tau} \sum_{t=1}^{T} \left(\frac{1 - \frac{1}{\alpha' \kappa_s}}{1 - \sqrt{\beta}}\right)^{T-t} 2\eta Z\mu^2$$

$$= \nu^T \|\bar{\boldsymbol{w}} - \boldsymbol{w}_0\|^2 + \frac{\omega^T}{\omega} \frac{1 - \left(\frac{\nu}{\omega}\right)^T}{1 - \left(\frac{\nu}{\omega}\right)} + \frac{\zeta}{\tau} \sum_{t=1}^{T} \left(\frac{1 - \frac{1}{\alpha' \kappa_s}}{1 - \sqrt{\beta}}\right)^{T-t} 2\eta Z\mu^2$$

$$\leq \nu^T \|\bar{\boldsymbol{w}} - \boldsymbol{w}_0\|^2 + \frac{\omega^T}{\omega} \frac{1}{1 - \left(\frac{\nu}{\omega}\right)} + \frac{\zeta}{\tau} \sum_{t=1}^{T} \left(\frac{1 - \frac{1}{\alpha' \kappa_s}}{1 - \sqrt{\beta}}\right)^{T-t} 2\eta Z\mu^2$$

$$\stackrel{(a)}{\leq} \nu^T \|\bar{\boldsymbol{w}} - \boldsymbol{w}_0\|^2 + \frac{\omega^T}{\omega} \frac{1}{1-\omega} + \frac{\zeta}{\tau} \sum_{t=1}^T \left(\frac{1 - \frac{1}{\alpha' \kappa_s}}{1 - \sqrt{\beta}}\right)^{T-t} 2\eta Z \mu^2$$

$$\stackrel{(b)}{\leq} \nu^T \|\bar{\boldsymbol{w}} - \boldsymbol{w}_0\|^2 + \frac{4}{3}\omega^T \frac{1}{1-\omega} + \frac{\zeta}{\tau} \sum_{t=1}^T \left(\frac{1 - \frac{1}{\alpha' \kappa_s}}{1 - \sqrt{\beta}}\right)^{T-t} 2\eta Z \mu^2$$

$$\stackrel{(c)}{\leq} \omega^T \|\bar{\boldsymbol{w}} - \boldsymbol{w}_0\|^2 + \frac{4}{3}\omega^T \frac{1}{1-\omega} + \frac{\zeta}{\tau} \sum_{t=1}^T \left(\frac{1 - \frac{1}{\alpha' \kappa_s}}{1 - \sqrt{\beta}}\right)^{T-t} 2\eta Z \mu^2$$

$$\stackrel{(d)}{\leq} \frac{\omega^T}{1-\omega} \|\bar{\boldsymbol{w}} - \boldsymbol{w}_0\|^2 + \frac{4}{3}\omega^T \frac{1}{1-\omega} + \frac{\zeta}{\tau} \sum_{t=1}^T \left(\frac{1 - \frac{1}{\alpha' \kappa_s}}{1 - \sqrt{\beta}}\right)^{T-t} 2\eta Z \mu^2$$

$$= \frac{\omega^T}{1-\omega} \left(\|\bar{\boldsymbol{w}} - \boldsymbol{w}_0\|^2 + \frac{4}{3}\right) + \frac{\zeta}{\tau} \sum_{t=1}^T \left(\frac{1 - \frac{1}{\alpha' \kappa_s}}{1 - \sqrt{\beta}}\right)^{T-t} 2\eta Z \mu^2$$

$$= 4\frac{\alpha'}{\rho} \kappa_s \omega^T \left(\|\bar{\boldsymbol{w}} - \boldsymbol{w}_0\|^2 + \frac{4}{3}\right) + \frac{\zeta}{\tau} \sum_{t=1}^T \left(\frac{1 - \frac{1}{\alpha' \kappa_s}}{1 - \sqrt{\beta}}\right)^{T-t} 2\eta Z \mu^2,$$

where in the left hand side we have used the linearity of expectation, and where (a) uses equation 58, (b) uses the fact that $\frac{1}{\omega} = \frac{1}{1 - \frac{1}{4\frac{\alpha'}{\rho}\kappa_s}} \leq \frac{1}{1 - \frac{1}{4}} = \frac{4}{3}$ (since $\kappa_s \geq 1$ and $\alpha' \geq 1$ (indeed, we have $\alpha' = 2\alpha = 2(\frac{C}{L_{s'}} + 1)$ with $C > 0$), so consequently $\frac{\alpha'}{\rho} \geq 1$), (c) uses equation 57, and (d) uses the fact that $\omega < 1$ so $1 < \frac{1}{1-\omega}$.

Let us now normalize the above inequality:

$$\mathbb{E}\frac{\sum_{t=1}^T 2\eta \left(\frac{1 - \frac{1}{\alpha' \kappa_s}}{1 - \sqrt{\beta}}\right)^{T-t} [(1-\rho)R(\boldsymbol{w}_t) + \rho R(\boldsymbol{v}_t)]}{\sum_{t=1}^T 2\eta \left(\frac{1 - \frac{1}{\alpha' \kappa_s}}{1 - \sqrt{\beta}}\right)^{T-t}} \leq R(\bar{\boldsymbol{w}}) + \frac{4\frac{\alpha'}{\rho}\kappa_s \omega^T \left(\|\bar{\boldsymbol{w}} - \boldsymbol{w}_0\|^2 + \frac{4}{3}\frac{\zeta}{\tau}\right)}{\sum_{t=1}^T 2\eta \left(\frac{1 - \frac{1}{\alpha' \kappa_s}}{1 - \sqrt{\beta}}\right)^{T-t}} + Z\mu^2.$$

The left hand side above is a weighted sum, which is an upper bound on the smallest term of the sum.

Regarding the right hand side, we can simplify it using the fact that $0 < \left(\frac{1 - \frac{1}{\alpha' \kappa_s}}{1 - \sqrt{\beta}}\right)$, and therefore:

$$\sum_{t=1}^T \left(\frac{1 - \frac{1}{\alpha' \kappa_s}}{1 - \sqrt{\beta}}\right)^{T-t} \geq 1.$$

Therefore, we obtain:

$$\mathbb{E} \min_{t \in \{1,..,T\}} [(1-\rho)R(\boldsymbol{w}_t) + \rho R(\boldsymbol{v}_t) - R(\bar{\boldsymbol{w}})] \leq \frac{4\frac{\alpha'}{\rho}\kappa_s \omega^T \left(\|\bar{\boldsymbol{w}} - \boldsymbol{w}_0\|^2 + \frac{4}{3}\frac{\zeta}{\tau}\right)}{2\eta} + Z\mu^2$$

$$= 4\frac{\alpha^2}{\rho} L_{s'} \kappa_s \omega^T \left(\|\bar{\boldsymbol{w}} - \boldsymbol{w}_0\|^2 + \frac{4}{3}\frac{\zeta}{\tau}\right) + Z\mu^2,$$

which can be simplified into the expression below, using the definition of $\hat{\boldsymbol{w}}_T$:

$$\mathbb{E}[\min_{t \in [T]} (1-\rho)R(\boldsymbol{w}_t) + \rho R(\boldsymbol{v}_t) - R(\bar{\boldsymbol{w}})] \leq 4\frac{\alpha^2}{\rho} L_{s'} \kappa_s \omega^T \left(\|\bar{\boldsymbol{w}} - \boldsymbol{w}_0\|^2 + \frac{4}{3}\frac{\zeta}{\tau}\right) + Z\mu^2. \tag{59}$$

To simplify the above result, we recall the assumptions made earlier on: we have chosen

$\tau = \frac{16\kappa_s \varepsilon_F}{(\alpha - 1)}$, and $G = \frac{4}{\nu_s}$ .

Therefore, to sum up, we have:

$$Z = \varepsilon_\mu \left( \frac{G}{2} + \frac{1}{C} \right) + \frac{\varepsilon_{abs}}{C} = \varepsilon_\mu \left( \frac{2}{\nu_s} + \frac{1}{C} \right) + \frac{\varepsilon_{abs}}{C}$$

$$\omega = 1 - \frac{1}{4\frac{\alpha'}{\rho}\kappa_s} = 1 - \frac{1}{8\frac{\alpha}{\rho}\kappa_s}$$

$$\zeta = \frac{4\eta\varepsilon_F \|\nabla R(\bar{w})\|^2}{C}$$

The last inequality implies: $\frac{\zeta}{\tau} = \frac{\frac{4\eta\varepsilon_F \|\nabla R(\bar{w})\|^2}{C}}{16\kappa_s L_{s'}\frac{\varepsilon_F}{C}} = \frac{\eta\|\nabla R(\bar{w})\|^2}{4\kappa_s L_{s'}}$.

Let us denote by $\varepsilon_T$ the right-hand side term from equation 59:

$$\varepsilon_T = 4\frac{\alpha^2}{\rho}L_{s'}\kappa_s\omega^T \left( \|\bar{w} - w_0\|^2 + \frac{4}{3}\frac{\eta\|\nabla R(\bar{w})\|^2}{4\kappa_s L_{s'}} \right) + Z\mu^2.$$

We now proceed similarly as in the proof of Theorem 4.3 above. Recall that we have assumed in the Assumptions of Theorem 4.8, without loss of generality, that $R$ is non-negative, which implies that $R(v_t) \geq 0$. Plugging this in equation 59 implies that:

$$\mathbb{E}\min_{t\in[T]} R(w_t) \leq \frac{1}{1-\rho}R(\bar{w}) + \frac{\varepsilon_T}{1-\rho} + \frac{Z}{(1-\rho)}\mu^2 \leq (1+2\rho)R(\bar{w}) + \frac{\varepsilon_T}{1-\rho} + \frac{Z}{1-\rho}\mu^2. \tag{60}$$

Plugging the change of variable $\varepsilon'_T = \frac{\varepsilon_T}{1-\rho}$ into equation 60 above, and redefining $Z$ into $Z := \frac{1}{1-\rho}\left(\varepsilon_\mu\left(\frac{2}{\nu_s} + \frac{1}{C}\right) + \frac{\varepsilon_{abs}}{C}\right)$, we obtain that:

$$\mathbb{E}\min_{t\in[T]} R(w_t) \leq (1+2\rho)R(\bar{w}) + \varepsilon'_T + Z\mu^2.$$

Further, consider an ideal case where $\bar{w}$ is a global minimizer of $R$ over $\mathcal{B}_0(k) := \{w : \|w\|_0 \leq k\}$. Then $R(v_t) \geq R(\bar{w})$ is always true for all $t \geq 1$. It follows that the bound in equation 59 yields:

$$\mathbb{E}\min_{t\in[T]}\{(1-\rho)R(w_t) + \rho R(\bar{w})\} \leq \mathbb{E}\min_{t\in[T]}\{(1-\rho)R(w_t) + \rho R(v_t)\} \leq R(\bar{w}) + \varepsilon_T,$$

which implies: $\mathbb{E}\min_{t\in[T]} R(w_t) \leq R(\bar{w}) + \frac{\varepsilon_T}{1-\rho}$. In this case, we can simply set $\rho = 0.5$, and define $\varepsilon'_T = \frac{\varepsilon_T}{1-\rho} = 2\varepsilon_T$ similarly as above. The proof is completed.

$\square$

### F.5. Proof of Corollary 4.9

*Proof of Corollary 4.9.* Let $\varepsilon \in \mathbb{R}_+^*$. Let us find $T$ to ensure that $\mathbb{E}\min_{t\in\{1,..,T\}}(1-\rho)R(w_t) + \rho R(v_t) - R(\bar{w}) \leq \varepsilon + Z\mu^2$

This will be enforced if:

$$4\alpha^2\frac{1}{\rho}L_{s'}\kappa_s\omega^T \left( \|\bar{w} - w_0\|^2 + \frac{4}{3}\frac{\eta\|\nabla R(\bar{w})\|^2}{4\kappa_s L_{s'}} \right) \leq \varepsilon$$

$$\iff T\log(\omega) \leq \log\left( \frac{\varepsilon}{4\alpha^2\frac{1}{\rho}L_{s'}\kappa_s\left(\|\bar{w} - w_0\|^2 + \frac{4}{3}\frac{\eta\|\nabla R(\bar{w})\|^2}{4\kappa_s L_{s'}}\right)} \right)$$

$$\iff T \geq \frac{1}{\log(\frac{1}{\omega})}\log\left( \frac{4\alpha^2\frac{1}{\rho}L_{s'}\kappa_s\left(\|\bar{w} - w_0\|^2 + \frac{4}{3}\frac{\eta\|\nabla R(\bar{w})\|^2}{4\kappa_s L_{s'}}\right)}{\varepsilon} \right).$$

Therefore, let us take:

$$T := \left\lceil \frac{1}{\log(\frac{1}{\omega})} \log\left( \frac{4\alpha^2 \frac{1}{\rho} L_{s'} \kappa_s \left( \|\bar{\boldsymbol{w}} - \boldsymbol{w}_0\|^2 + \frac{4}{3} \frac{\eta \|\nabla R(\bar{\boldsymbol{w}})\|^2}{4\kappa_s L_{s'}} \right)}{\varepsilon} \right) \right\rceil . \tag{61}$$

We can now derive the #IZO and #HT. First, we have one hard-thresholding operation at each iteration, therefore #HT$= T$. Using the fact that $\frac{1}{\log(\frac{1}{\omega})} = \frac{1}{-\log(\omega)} = \frac{1}{-\log(1-\frac{1}{8\alpha\frac{1}{\rho}\kappa_s})} \leq \frac{1}{\frac{1}{8\alpha\frac{1}{\rho}\kappa_s}} = 8\alpha\frac{1}{\rho}\kappa_s$ (since by property of the logarithm, for all $x \in (-\infty, -1) : \log(1-x) \leq -x$ ), and the fact that $\alpha = \frac{C}{L_{s'}}$ is independent of $\kappa_s$, we obtain that #HT $= \mathcal{O}(\kappa_s \log\left(\frac{1}{\varepsilon}\right))$.

We now turn to computing the #IZO. At each iteration $t$ we have $q_t$ function evaluations, therefore:

$$\begin{aligned}
\text{\#IZO} &= \sum_{t=0}^{T-1} q_t \\
&\leq \sum_{t=0}^{T-1} \left( \frac{\tau}{\omega^t} + 1 \right) \\
&= T + \tau \frac{\left(\frac{1}{\omega}\right)^T - 1}{\frac{1}{\omega} - 1} \\
&\leq T + \frac{\tau}{\frac{1}{\omega} - 1} \left(\frac{1}{\omega}\right)^T \\
&= T + \frac{\tau}{\frac{1}{\omega} - 1} \exp\left( T \log\left(\frac{1}{\omega}\right) \right) \\
&\overset{(a)}{\leq} 1 + \frac{1}{\log(\frac{1}{\omega})} \log\left( \frac{4\alpha^2 \frac{1}{\rho} L_{s'} \kappa_s \left( \|\bar{\boldsymbol{w}} - \boldsymbol{w}_0\|^2 + \frac{4}{3} \frac{\eta \|\nabla R(\bar{\boldsymbol{w}})\|^2}{4\kappa_s L_{s'}} \right)}{\varepsilon} \right) \\
&\quad + \frac{\tau}{\frac{1}{\omega} - 1} \exp\left( \log\left(\frac{1}{\omega}\right) \left[ \frac{1}{\log(\frac{1}{\omega})} \log\left( \frac{4\alpha^2 \frac{1}{\rho} L_{s'} \kappa_s \left( \|\bar{\boldsymbol{w}} - \boldsymbol{w}_0\|^2 + \frac{4}{3} \frac{\eta \|\nabla R(\bar{\boldsymbol{w}})\|^2}{4\kappa_s L_{s'}} \right)}{\varepsilon} \right) \right. \right. \\
&\quad \left. \left. +1 \right] \right) \\
&= 1 + \frac{1}{\log(\frac{1}{\omega})} \log\left( \frac{4\alpha^2 \frac{1}{\rho} L_{s'} \kappa_s \left( \|\bar{\boldsymbol{w}} - \boldsymbol{w}_0\|^2 + \frac{4}{3} \frac{\eta \|\nabla R(\bar{\boldsymbol{w}})\|^2}{4\kappa_s L_{s'}} \right)}{\varepsilon} \right) \\
&\quad + \frac{\frac{\tau}{\omega}}{\frac{1}{\omega} - 1} \frac{4\alpha^2 \frac{1}{\rho} L_{s'} \kappa_s \left( \|\bar{\boldsymbol{w}} - \boldsymbol{w}_0\|^2 + \frac{4}{3} \frac{\eta \|\nabla R(\bar{\boldsymbol{w}})\|^2}{4\kappa_s L_{s'}} \right)}{\varepsilon} \\
&= 1 + \frac{1}{\log(\frac{1}{\omega})} \log\left( \frac{4\alpha^2 \frac{1}{\rho} L_{s'} \kappa_s \left( \|\bar{\boldsymbol{w}} - \boldsymbol{w}_0\|^2 + \frac{4}{3} \frac{\eta \|\nabla R(\bar{\boldsymbol{w}})\|^2}{4\kappa_s L_{s'}} \right)}{\varepsilon} \right) \\
&\quad + \frac{\tau}{1 - \omega} \frac{4\alpha^2 \frac{1}{\rho} L_{s'} \kappa_s \left( \|\bar{\boldsymbol{w}} - \boldsymbol{w}_0\|^2 + \frac{4}{3} \frac{\eta \|\nabla R(\bar{\boldsymbol{w}})\|^2}{4\kappa_s L_{s'}} \right)}{\varepsilon} \\
&= 1 + \frac{1}{\log(\frac{1}{\omega})} \log\left( \frac{4\alpha^2 \frac{1}{\rho} L_{s'} \kappa_s \left( \|\bar{\boldsymbol{w}} - \boldsymbol{w}_0\|^2 + \frac{4}{3} \frac{\eta \|\nabla R(\bar{\boldsymbol{w}})\|^2}{4\kappa_s L_{s'}} \right)}{\varepsilon} \right) \\
&\quad + \tau \frac{32\alpha^3 \frac{1}{\rho^2} L_{s'} \kappa_s^2 \left( \|\bar{\boldsymbol{w}} - \boldsymbol{w}_0\|^2 + \frac{4}{3} \frac{\eta \|\nabla R(\bar{\boldsymbol{w}})\|^2}{4\kappa_s L_{s'}} \right)}{\varepsilon} ,
\end{aligned}$$

where (a) follows from equation 61.

And we recall that $\tau = 16\kappa_s \frac{\varepsilon_F}{(\alpha-1)}$, which implies that:

$$\tau \frac{32\alpha^3 \frac{1}{\rho^2} L_{s'} \kappa_s^2 \left( \|\bar{\boldsymbol{w}} - \boldsymbol{w}_0\|^2 + \frac{4}{3} \frac{\eta \|\nabla R(\bar{\boldsymbol{w}})\|^2}{2\gamma \kappa_s L_{s'}} \right)}{\varepsilon} = \mathcal{O} \left( \frac{\varepsilon_F}{\varepsilon} \left( \kappa_s^3 L_{s'} + \frac{\kappa_s}{\nu_s} \right) \right).$$

Therefore, overall, the IZO (query complexity) is in $\mathcal{O}\left(\frac{\varepsilon_F}{\varepsilon} \kappa_s^3 L_{s'}\right)$. The proof is completed. $\qquad\square$

## G. Differences Between Two-Step Projection and Euclidean Projection

In this section, we describe the differences between the two-step projection and the Euclidean projection onto the mixed constraints $\Gamma \cap \mathcal{B}_0(k)$. One can encounter several possible cases:

- **Case (i):** the two-step projection (2SP) and the Euclidean projection onto $\Gamma \cap \mathcal{B}_0(k)$ are identical (see e.g. Remark 3.1): in that case, the contribution of our paper are on the theoretical side: Theorems 3.7, 4.3, and 4.8 give global convergence guarantee which therefore in this case apply to the usual (non-convex) projected gradient descent algorithm with Euclidean projection.

- **Case (ii):** the 2SP and the Euclidean projection onto the mixed constraints are different: this case can be declined into several sub-cases as described below:

    - Case (a): the Euclidean projection onto the mixed constraint $\Gamma \cap \mathcal{B}_0(k)$ is unknown: in that case, the 2SP can allow to fill such gap, since the 2SP only requires the knowledge of the projection onto $\Gamma$, which is often known and easy to do.
    - Case (b): the Euclidean projection onto the mixed constraint $\Gamma \cap \mathcal{B}_0(k)$ is known, but computationally expensive: in that case, the 2SP can provide a simpler and faster alternative to the Euclidean projection, while still enjoying some convergence guarantees as shown in this paper.
    - Case (c): the Euclidean projection onto the mixed constraint $\Gamma \cap \mathcal{B}_0(k)$ is known and is efficient enough (e.g. when $\Gamma$ belongs to the set of positive symmetric sets such as in (Lu, 2015)). In such cases, it is unclear whether the 2SP can improve upon Euclidean projection since, at the iteration level, using the Euclidean projection is optimal (indeed, a (Euclidean) projected gradient descent step minimizes a quadratic upper bound on the objective value under constraints (derived from the smoothness of $R$)), and the 2SP is therefore suboptimal in that sense (at the iteration level).

## H. Experiments

In this section, we provide some experiments to validate experimentally our theoretical results. Our experiments take 2h30 to run in total on a MacBook Pro with a 2.6GHz 6-core Intel Core i7 and 16GB of memory. Before describing our experiments, we provide a short discussion about the settings and algorithms that we will illustrate. For constraints $\Gamma$ for which the Euclidean projection onto $\mathcal{B}_0(k) \cap \Gamma$ has a closed form equal to the 2SP, our algorithm is identical to a vanilla non-convex projected gradient descent baseline (see Remark 3.1). In such case, our contribution in this paper is on the theoretical side, by providing some global guarantees on the optimization, instead of the local guarantees from existing work (cf. Table 1). Additionally, there are case in which there exists a closed form for projection onto $\Gamma \cap \mathcal{B}_0(k)$, different from the 2SP (e.g. when $\Gamma = \mathbb{R}_+^d$, cf. (Lu, 2015)). Although our framework allows us to get approximate global convergence results when using the 2SP, still, at the iteration level, a gradient step followed by Euclidean projection (not 2SP) is optimal, since it minimizes a constrained quadratic upper bound on $R$. Therefore, we may not expect much improvement of the 2SP over the Euclidean projection in such case, except on the computational side. With this in mind, we provide below the outline of our experiments:

- In Section H.1, we illustrate on a synthetic example the trade-off between sparsity and optimality that is introduced by the extra constraint $\Gamma$, and that is balanced by the parameter $\rho$.

- In Section H.2, we consider a portfolio index tracking problem where the goal is to illustrate a real-life application of our methods.

- In section H.3, we consider a multi-class logistic regression on a real life dataset, to illustrate in more details in particular the stochastic and the zeroth-order versions of our method.

For reproducibility, our code is available at `https://github.com/wdevazelhes/2SP_icml2025`.

## H.1. Synthetic Experiments: Illustrating the Sparsity/Optimality Trade-Off

In the section below, we provide a synthetic experiment to illustrate our Theorem 3.7, i.e. the trade-off between sparsity and optimality that is introduced by the extra constraint $\Gamma$, and that is balanced by $\rho \in (0, 0.5]$. We consider the synthetic linear regression example from (Axiotis & Sviridenko, 2022) (Section E), with the risk below:

$$R(\boldsymbol{w}) := \frac{1}{n} \|\boldsymbol{X}\boldsymbol{w} - \boldsymbol{y}\|_2^2,$$

and where $\boldsymbol{X}$ is diagonal with:

$$\boldsymbol{X}_{ii} = \begin{cases} 1 & \text{if } i \in I_1 \\ \sqrt{\kappa} & \text{if } i \in I_2 \\ 1 & \text{if } i \in I_3, \end{cases}$$

where $I_1 = [s], I_2 = [s+1, s(\kappa+1)], I_3 = [s(\kappa+1)+1, s(\kappa^2+\kappa+1)]$ for some $s \geq 1$ and $\kappa \geq 1$ (we choose $s = 50$ and $\kappa = 2$, which results in having $d = 350$), $n$ denotes the number of rows of $\boldsymbol{X}$, and $\boldsymbol{y}$ is defined as

$$y_i = \begin{cases} \kappa\sqrt{1-4\delta} & \text{if } i \in I_1 \\ \sqrt{\kappa}\sqrt{1-2\delta} & \text{if } i \in I_2 \\ 1 & \text{if } i \in I_3 \end{cases}$$

for some small $\delta > 0$ used for tie-breaking (we set it to $1e-4$). We chose such an example as it is used by (Axiotis & Sviridenko, 2022) to prove a lower bound on the fundamental trade-off between sparsity and optimality proper to IHT: they use it to show that the relaxation of the sparsity $k$, of the order $k = \Omega(\kappa^2 \bar{k})$ (see also Table 1) is in fact unavoidable for IHT-type algorithms.

**Case without Extra Constraints.** First, we illustrate our Theorem 3.4 which considers vanilla IHT, without extra constraints. In Figure 2, on the one hand, we plot in blue, for every $k \in [d]$, the value of $R(\hat{\boldsymbol{w}}_k)$ where $\hat{\boldsymbol{w}}_k$ is the result of running vanilla IHT with sparsity $k$ up to convergence. Then, on the other hand, we go through every value of $\bar{k} \in [d]$, and for each of them, we plot a point $(K(\bar{k}), R(\bar{\boldsymbol{w}}_{\bar{k}}))$, where $K(\bar{k})$ denotes the value of $k$ required in our Theorem 3.4, i.e.: $K(\bar{k}) := 4\kappa^2\bar{k}$, and $\bar{\boldsymbol{w}}_{\bar{k}} := \min_{\boldsymbol{w} \in \mathbb{R}^d : \|\boldsymbol{w}\|_0 \leq \bar{k}} R(\boldsymbol{w})$. Therefore, each of such point $R(\bar{\boldsymbol{w}}_{\bar{k}})$ constitutes an upper bound on the value of $R(\hat{\boldsymbol{w}}_{K(\bar{k})})$, as we can indeed observe on Figure 2.

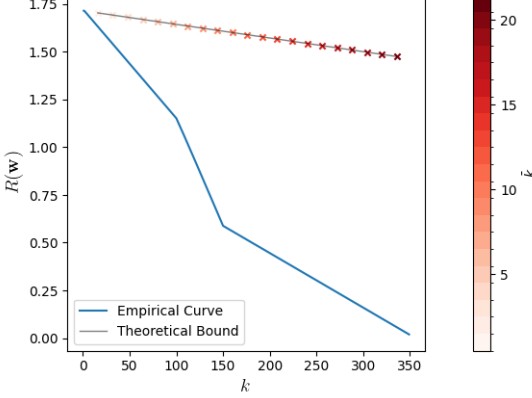

*Figure 2.* Illustration of Theorem 3.4 (i.e. $\Gamma = \mathbb{R}^d$).

**Case with Extra Constraints.** We now illustrate the influence of the extra constraint $\Gamma$ on the problem. We consider for $\Gamma$ an $\ell_\infty$ norm constraint of radius $\lambda > 0$, that is: $\Gamma = \{w \in \mathbb{R}^d : \forall i \in [d], |w_i| \leq \lambda\}$. In this new setting, we also go through every value of $\bar{k} \in [d]$, but this time, each of those values actually defines a curve parameterized by $\rho$, according to our Theorem 3.7: for each $\bar{k}$ we plot the parametric curve $(K(\bar{k}, \rho), (1 + 2\rho)R(\bar{w}_{\bar{k}}))$, where, similarly as above, $K(\bar{k}, \rho)$ denotes the required value of $k$ according to Theorem 3.7 (i.e., $K(\bar{k}, \rho) = \frac{4(1-\rho)^2 \bar{k}\kappa^2}{\rho^2}$), and $\bar{w}_{\bar{k}} := \min_{w \in \mathbb{R}^d : \|w\|_0 \leq \bar{k}} R(w)$, and where $\rho$ ranges in $(0, 0.5]$. We present the results for several values of $\lambda$ in Figure 7 below. Note *that a priori*, the curves are allowed to cross, i.e. for a given $k$ on the x-axis, one could have a point from a curve of small $\bar{k}$ (i.e. lighter shade of red) which could potentially also belong to a curve of larger $\bar{k}$ (let us denote it $\bar{k}'$) (darker shade of red), which would necessarily have a larger $\rho$ (let us denote it $\rho'$), but for which the overall $(1 + 2\rho')R(\bar{w}_{\bar{k}'})$ could be equal to $(1 + 2\rho)R(\bar{w}_{\bar{k}})$ (since the problem will be less constrained with $\bar{k}'$ than with $k$). However, interestingly, this is not the case here due to the simplicity of the structure of the example. We can also observe that similarly as in the case where $\Gamma = \mathbb{R}^d$, the bound is a bit tighter in the small $k$ regime (i.e. when $k \in [50, 100]$).

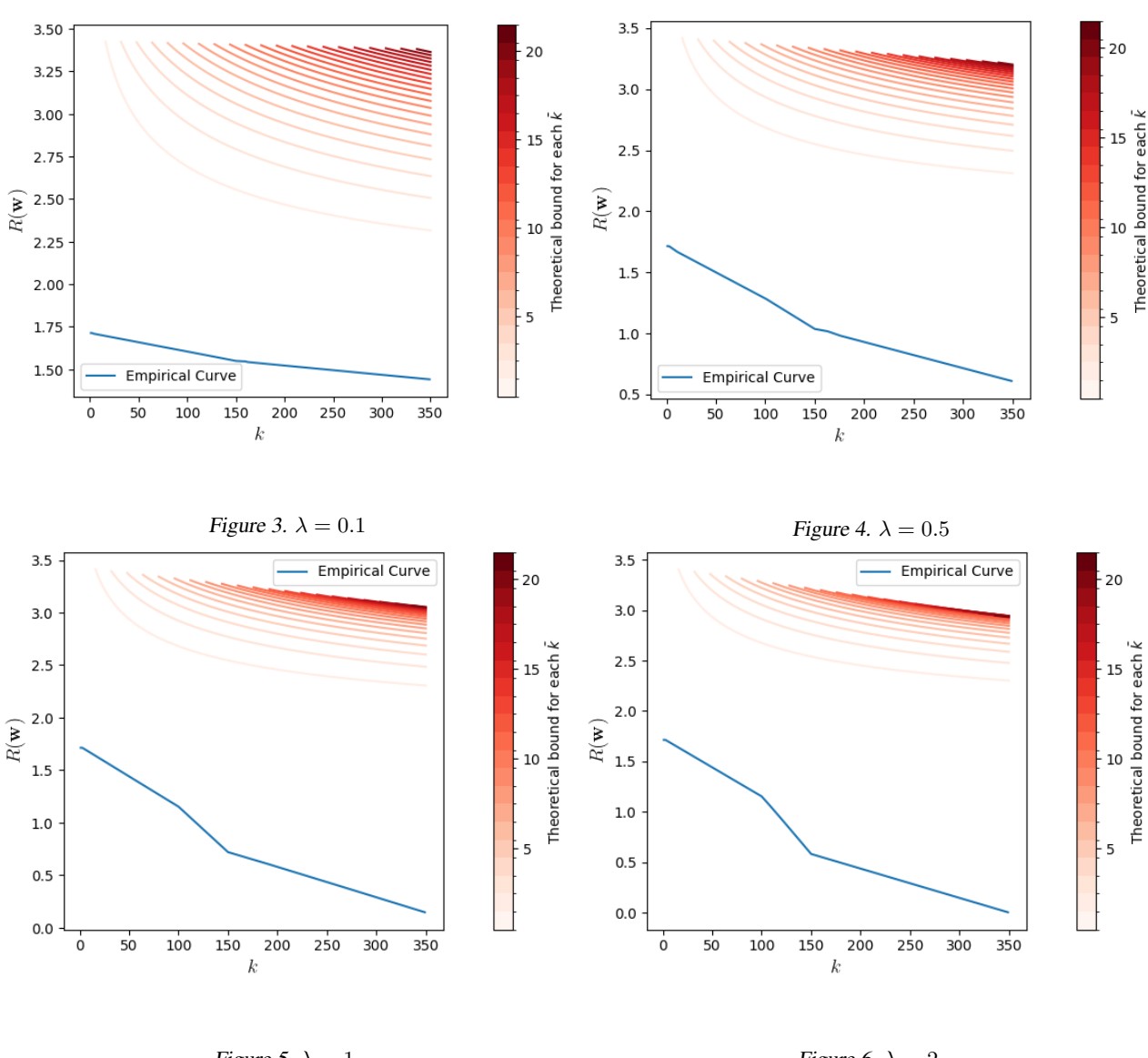

*Figure 3.* $\lambda = 0.1$

*Figure 4.* $\lambda = 0.5$

*Figure 5.* $\lambda = 1$

*Figure 6.* $\lambda = 2$

*Figure 7.* Illustration of Theorem 3.4 (with $\Gamma$ an $\ell_\infty$ ball of radius $\lambda$).

## H.2. Real Data Experiment: Portfolio Index Tracking

We now consider the following index tracking problem, originally presented in (Takeda et al., 2013), and used as well in (Lu, 2015; Beck & Hallak, 2016). It is also similar to the portfolio optimization problem presented in (Kyrillidis et al., 2013). We seek to reproduce the performance of an index fund (such as S&P500), by investing only in a few key $k$ assets, in order to limit transaction costs. The general problem can be formulated as a linear regression problem:

$$\min_{\boldsymbol{w} \in \mathcal{B}_0(k) \cap \Gamma} \|\boldsymbol{A}\boldsymbol{w} - \boldsymbol{y}\|^2 \tag{62}$$

where $\boldsymbol{w}$ represents the amount invested in each asset. For each $i \in [n]$ denoting a timestep , the $i$-th row of $\boldsymbol{A}$ denotes the returns of the $d$ stocks at timestep $i$, and $y_i$ the return of the index fund. In our scenario, we seek to limit to a value $D > 0$ the amount of transactions in each of $c$ activity sector (group) of the portfolio (e.g. Industrials, Healthcare, etc.), denoted as $G_i$ for $i \in [c]$. We ensure such constraint through an $\ell_1$ norm constraint on each group: $\Gamma = \{\boldsymbol{w} \in \mathbb{R}^d : \forall i \in [c], \|\boldsymbol{w}_{G_i}\|_1 \leq D\}$, where $\boldsymbol{w}_{G_i}$ is the restriction of $\boldsymbol{w}$ to group $G_i$ (i.e. for $j \in [d]$, $w_{G_{ij}} = w_j$ if $j \in G_i$ and 0 otherwise). In our case, $\boldsymbol{y}$ denotes the daily returns of a given portfolio index (e.g. S&P500) for a given time period (e.g. a given year), and $\boldsymbol{A}$ the returns of the corresponding $d$ assets (over $c$ sectors) of the index during such period.

**Baselines.** Up to our knowledge, there is no efficient closed form for the Euclidean projection onto $\mathcal{B}_0(k) \cap \Gamma$ (although such projection could be done by enumerating all sparse supports sets and computing the restricted projection onto those sets, such method would have exponential complexity), but the two-step projection can easily be done by projecting onto the $\ell_1$ ball for each sector independently. We compare our algorithm (FG-HT-2SP) to two naive baselines: (a) the first one. called "PGD($\Gamma$) + final$\Pi_{\mathcal{B}_0}$", consists in only ensuring the constraints in $\Gamma$, followed at the end of training by a simple hard-thresholding step to keep the $k$ largest components of $\boldsymbol{w}$ in absolute value, and (b) the second one, called "PGD($\mathcal{B}_0$) + final$\Pi_\Gamma$", consists in running vanilla IHT, followed at the end of training by a simple projection onto $\Gamma$ to keep $\boldsymbol{w}$ in $\Gamma \cap \mathcal{B}_0$. We learn the weights of the portfolio on 80% of the considered period, and evaluate the out of sample (test set) performance on the remaining 20% (shaded area in the figure).

**Datasets.** We compare our algorithms on three portfolio indices datasets:

- **S&P500**: We take $k = 15$ and $D = 50$. $\boldsymbol{y}$ denotes the daily returns from January 1, 2021, to December 31, 2022, and $\boldsymbol{A}$ denotes the returns of the corresponding $d = 497$ assets (over $c = 11$ sectors). We plot our results in Figure 8.

- **HSI**: We take $k = 15$ and $D = 1000$. $\boldsymbol{y}$ denotes the daily returns from January 1, 2021, to December 31, 2022, and $\boldsymbol{A}$ denotes the returns of the corresponding $d = 72$ assets (over $c = 4$ sectors). We plot our results in Figure 9.

- **CSI300**: We take $k = 15$ and $D = 100$. $\boldsymbol{y}$ denotes the daily returns from March 1, 2021 (due to missing values in early 2021), to December 31, 2022, and $\boldsymbol{A}$ denotes the returns of the corresponding $d = 291$ assets (over $c = 10$ sectors). We plot our results in Figure 10.

The data for those three indices is scrapped from the web using the `beautifulsoup`[1] library to gather information about the index, and the `yfinance`[2] library to scrap the returns of such stocks during the considered time period. We provide in Table 2 below the respective dimensions of the train-sets used for the experiments (which constitutes, as we recall, 80% of the total dataset).

| INDEX | $n$ | $d$ |
|---|---|---|
| S&P500 | 402 | 497 |
| CSI300 | 353 | 291 |
| HSI | 394 | 72 |

*Table 2.* Number of samples ($n$) and dimension ($d$) of the training sets for the index tracking experiment

---

[1]https://pypi.org/project/beautifulsoup4/
[2]https://github.com/ranaroussi/yfinance

**Results.**  As we can observe on Figure 11, overall, the true index (blue curve) is more successfully tracked by our method (FG-HT-2SP, green curve), on the train-set of S&P500 and CSI300 and on the test-set of HSI and CSI300. Additionally, we have observed that for S&P500, our algorithm solution non-zero weights spans 9 of the 11 sectors for the S&P500 index, 7 sectors out of 10 for the CSI300 index, and 3 of the 4 sectors the one for the HSI index. Therefore, such portfolios are well diversified, as successfully enforced by our constraint.

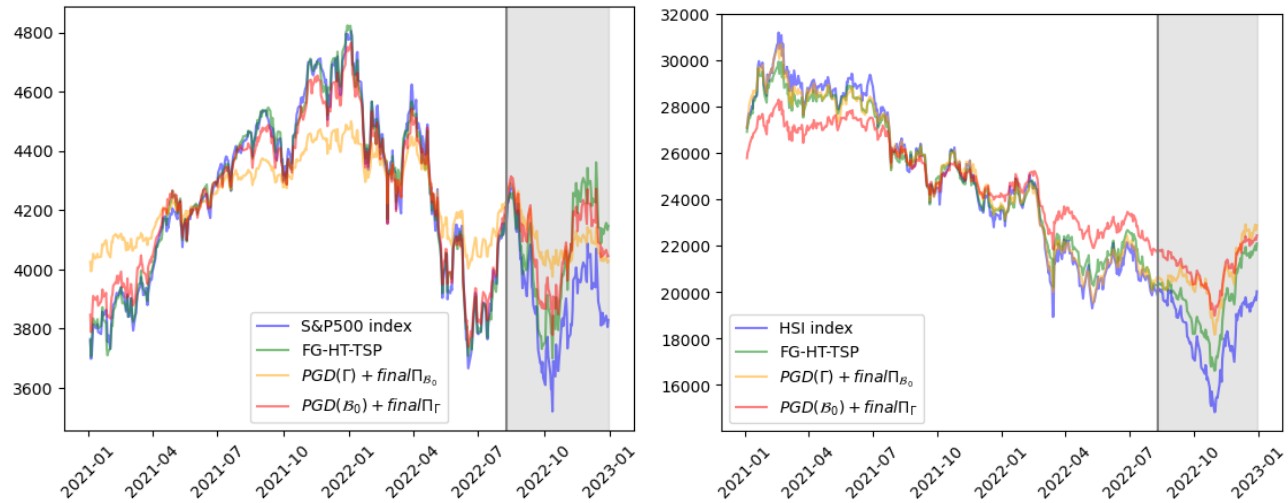

*Figure 8.* S&P500                                          *Figure 9.* HSI

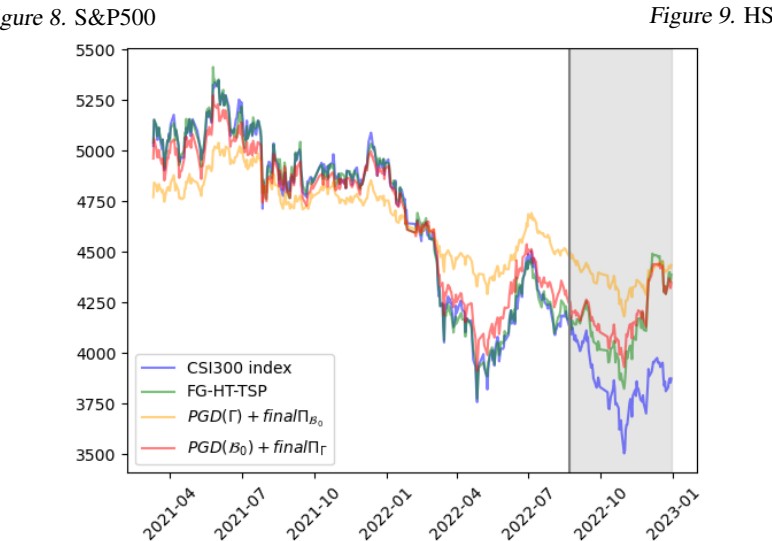

*Figure 10.* CSI300

*Figure 11.* Index tracking with sector constraints for various indices

**On the Verification of Assumptions 2.1 to 2.3:**  Note that such index tracking experiments verify Assumptions 2.1, 2.2 and 2.3:

- **Assumption 2.1** is verified since the cost function is quadratic, with a design matrix of size $n > d$ (except in the case of S&P500). As can be expected with such matrices in general, the Hessian $\boldsymbol{H} = 2\boldsymbol{A}^\top \boldsymbol{A}$ is positive-definite (we have indeed verified in our code that it is). Therefore the RSC constant is bounded below by $\lambda_{\min}$ where $\lambda_{\min}$ is the smallest

eigenvalue of $2\boldsymbol{A}^\top \boldsymbol{A}$. Note that for S&P500, strong convexity is not verified since $d > n$: however, since we take $k = 15$, with high probability (i.e. unless we can find $s = 2k = 30$ columns of $\boldsymbol{A}$ that are exactly linearly dependent), RSC should be verified.

- **Assumption 2.2 and Assumption 4.5** are both verified since the cost function is quadratic, therefore the (strong) RSS constant is bounded above by $2\|\boldsymbol{A}\|_s^2$, where $\|\cdot\|_s$ denotes the spectral norm.

- **Definition 2.3** is verified since projection onto $\Gamma$ can be done group-wise, and for each group the projection is onto an $\ell_1$ ball, which is a convex symmetric set (which is support-preserving from Remark 2.4), therefore, overall, $\Gamma$ is support-preserving).

## H.3. Real Data Experiment: Multiclass Logistic Regression

We now consider the multiclass logistic regression problem with class group-wise $\ell_2$ norm constraint as follows. We have $R_i(\boldsymbol{w}) = \sum_{j=1}^c \left[ \frac{\lambda}{c} \|\boldsymbol{w}_j\|_2^2 - \mathbf{1}\left\{y_i = j\right\} \log \frac{\exp(\boldsymbol{x}_i^\top \boldsymbol{w}_j)}{\sum_{l=1}^c \exp(\boldsymbol{x}_i^\top \boldsymbol{w}_l)} \right]$, where $y_i$ is the target output of $\boldsymbol{x}_i$, $c$ is the number of classes, and $\boldsymbol{w}_j$ is the weight vector specific to class $j$. In addition to the sparsity constraint $\mathcal{B}_0(k)$, we enforce the following additional constraint $\Gamma = \{\boldsymbol{w} \in \mathbb{R}^d : \forall j \in [c] : \|\boldsymbol{w}_j\|_2 \le D\}$, for some constant $D \in \mathbb{R}_+$, where $d = p \times c$, with $p$ the number of features of the samples $\boldsymbol{x}_i$. More precisely, in such multiclass logistic regression, we seek to ensure an extra regularization not only on the whole global weight vector $\boldsymbol{w}$ (with the used squared $\ell_2$ penalty), but also on each weight vector related to each class (through $\Gamma$), in order to prevent a potential class-wise overfitting.

Up to our knowledge, there is no known closed form for the Euclidean projection onto such $\Gamma \cap \mathcal{B}_0(k)$. However, the two-step projection (2SP) can be done easily: once the first projection is done (projection onto $\mathcal{B}_0(k)$, i.e. hard-thresholding) and the sparse support $S$ is identified as per Section 3.1, the projection onto $\Gamma$ restricted to $S$ can be easily done since $\Gamma$ is class-wise decomposable, and therefore it suffices to project, for each $j \in [c]$, each $\boldsymbol{w}_j$ onto the $\ell_2$ ball of radius $D$.

We have the smoothness constant $L$ as below (see (Böhning, 1992) for a derivation):

$$L = \sigma_{\max}\left( \frac{1}{2n}\left(\boldsymbol{I}_{c\times c} - \frac{1}{c}\mathbf{1}_c \mathbf{1}_c^\top\right) \otimes \boldsymbol{X}^\top \boldsymbol{X} + 2\lambda \boldsymbol{I}_{d\times d} \right) \tag{63}$$

Where $\otimes$ denotes the Kronecker product, $\sigma_{\max}$ the largest singular value of a matrix, $\boldsymbol{I}_{m\times m}$ the identity matrix of size $m \times m$ for some $m$, and $\mathbf{1}_c$ the vector $[1, 1, .., 1]^\top \in \mathbb{R}^c$.

We consider the `dna` dataset from the LibSVM dataset repository (Chang & Lin, 2011), and we choose $D = 0.5$, $\lambda = 10$. For the stochastic case we take $B = 1e^5$, and for the stochastic and ZO case we take $\alpha = 2$. Note that in the stochastic case, if the growing batch-size required by Theorem 4.3 becomes larger than $n$, we keep it fixed to $n$ (i.e. in such case we take the whole dataset at each step). In the zeroth-order case, we take $\mu = 1e - 6$. We set set all other hyperparameters as per Theorems 3.7, 4.3 and 4.8. In Figures 15, 19, 23 and 27, we plot the number of calls to a gradient $\nabla R_i$ (IFO: iterative first order oracle), and number of hard-thresholding operations (NHT), for various values of $k$ and $D$ (for the zeroth-order case, we plot the IZO (number of calls to the function $R$) instead of the IFO). We can observe that HSG-HT-2SP allows a smaller IFO than FG-HT-2SP in early iterations, since it does not need to compute a full gradient at each iteration.

In addition, to illustrate the theoretical improvement of our results on zeroth-order, even in the case where there is no additional constraint, we compare in Figures 30, 33 and 36 our algorithm HZO-HT with ZOHT (de Vazelhes et al., 2022), choosing for both algorithm an initial number of random direction as prescribed by our Theorem 4.8, and choosing, for the learning rate, in our case the one prescribed by Theorem 4.8, and for ZOHT, the one prescribed by Theorem 1 from (de Vazelhes et al., 2022) (and in both cases we fix $s = 3k$ as per Theorem 4.8): we can see that, in addition to being able to obtain a convergence in risk without system error, contrary to ZOHT (cf. Table 1), our Theorem 4.8 also prescribes a better (larger) learning rate (i.e. less conservative), leading to faster convergence.

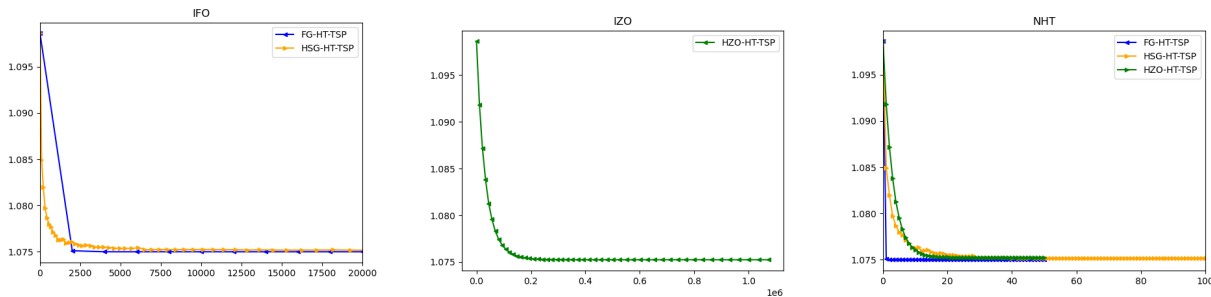

*Figure 12.* #IFO

*Figure 13.* #IZO

*Figure 14.* #NHT

*Figure 15.* Multiclass Logistic Regression with 2SP, $k = 50$, $D = 0.5$

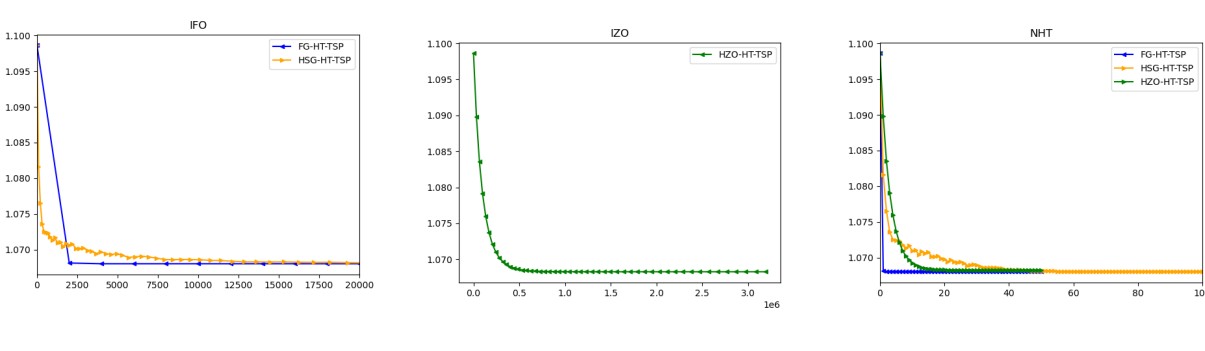

*Figure 16.* #IFO

*Figure 17.* #IZO

*Figure 18.* #NHT

*Figure 19.* Multiclass Logistic Regression with 2SP, $k = 150$, $D = 0.5$

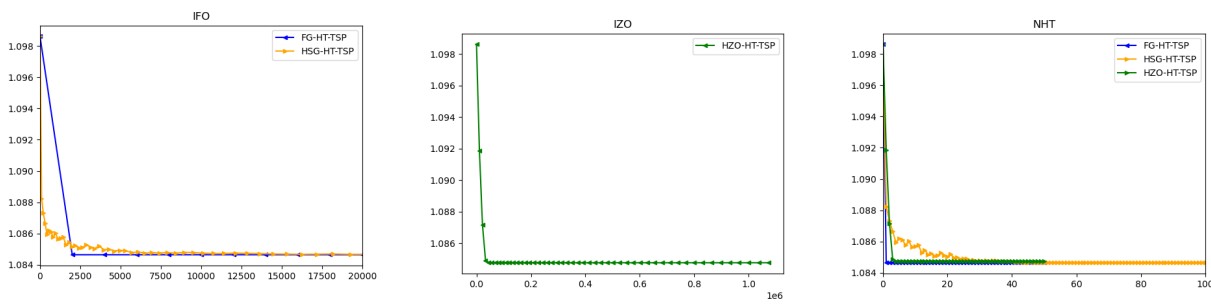

*Figure 20.* #IFO

*Figure 21.* #IZO

*Figure 22.* #NHT

*Figure 23.* Multiclass Logistic Regression with 2SP, $k = 50$, $D = 0.01$

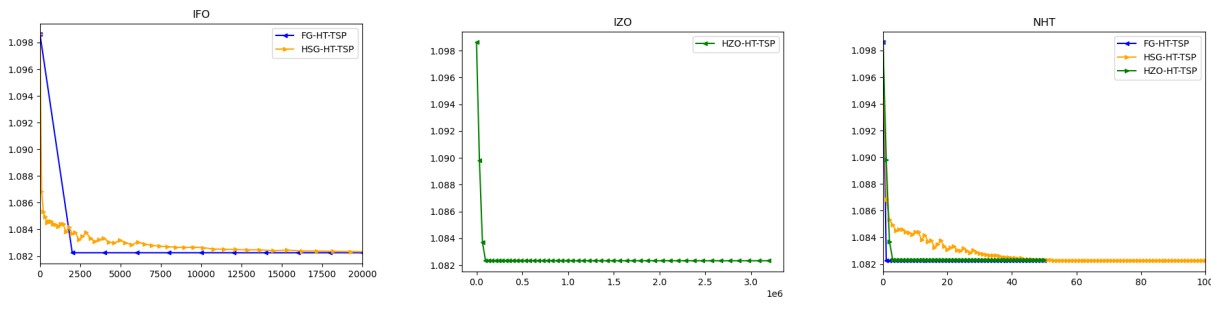

Figure 24. #IFO

Figure 25. #IZO

Figure 26. #NHT

Figure 27. Multiclass Logistic Regression with 2SP, $k = 150$, $D = 0.01$

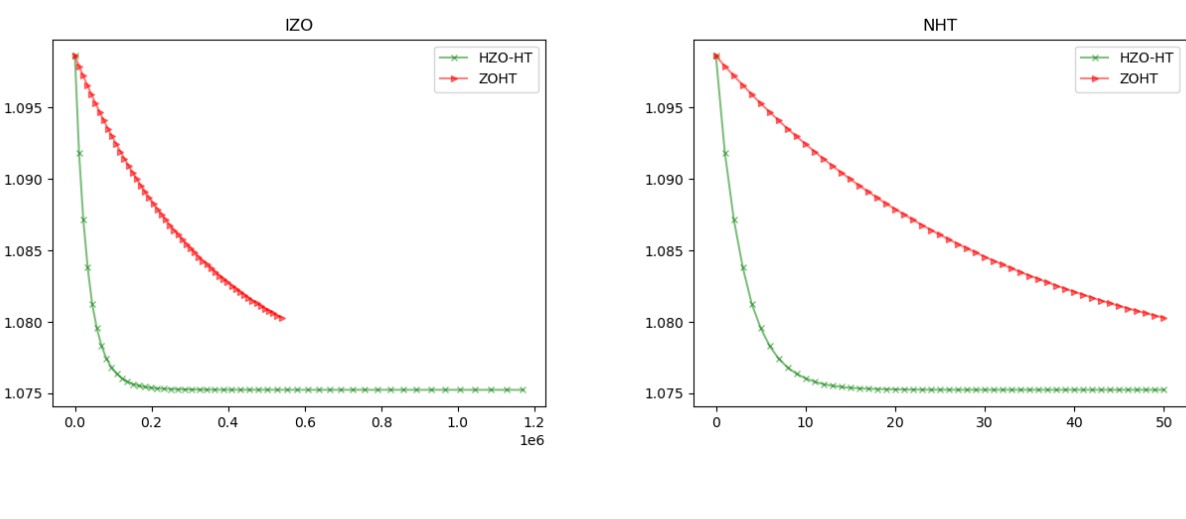

Figure 28. #IZO

Figure 29. #NHT

Figure 30. Multiclass Logistic Regression: HZO-HT vs. ZOHT, $k = 50$

**On the Verification of Assumptions 2.1 to 2.3:** Note that such logistic regression experiments verify Assumptions 2.1, 2.2, 4.5 and 2.3:

- **Assumption 2.1** is verified thanks to the added squared $\ell_2$ regularization, which makes the problem strongly convex and hence also restricted strongly convex.

- **Assumption 2.2 and Assumption 4.5** are both verified since the problem is smooth with a constant $L$ as described above in equation 63, and therefore such constant is also a valid (strong) restricted-smoothness constant.

- **Definition 2.3** is verified since, since, similarly as in the index tracking experiments from Section H.2, projection onto $\Gamma$ can be done group-wise, and for each group the projection is onto an $\ell_1$ ball, which is a convex sign-free set (which is support-preserving from Remark 2.4), therefore, overall, $\Gamma$ is support-preserving.

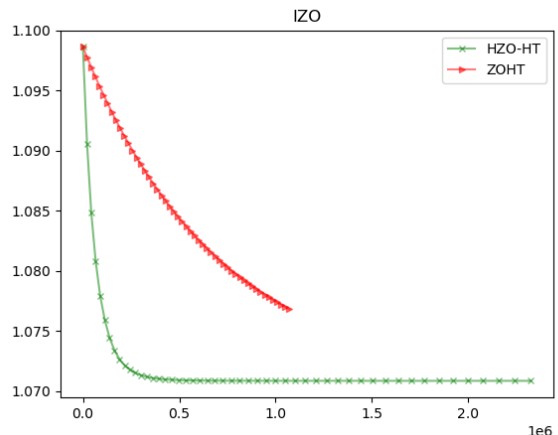

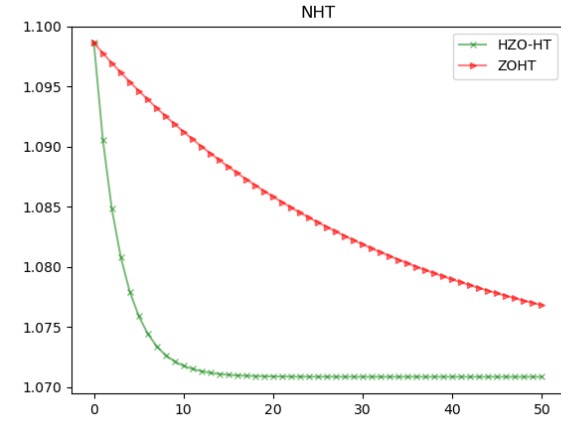

*Figure 31.* #IZO

*Figure 32.* #NHT

*Figure 33.* Multiclass Logistic Regression: HZO-HT

vs. ZOHT, $k = 100$

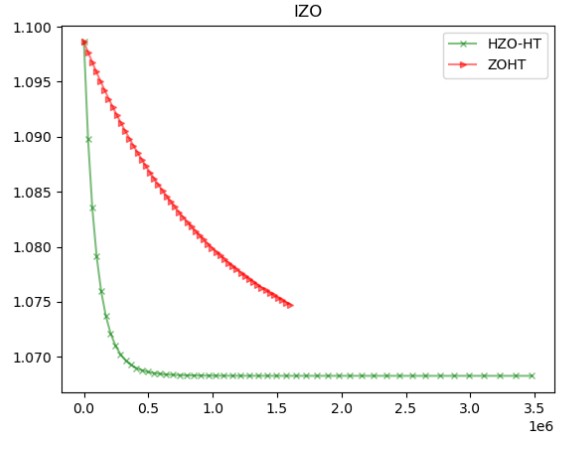

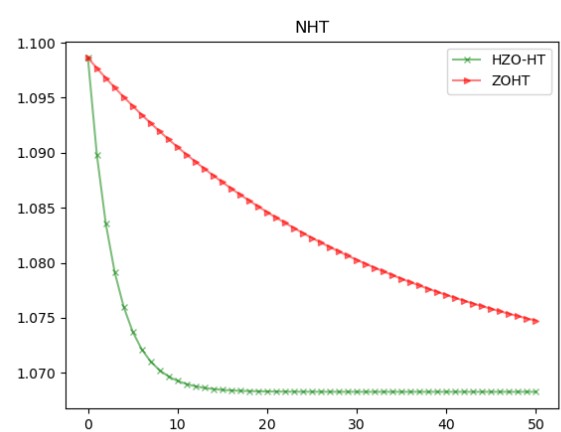

*Figure 34.* #IZO

*Figure 35.* #NHT

*Figure 36.* Multiclass Logistic Regression: HZO-HT

vs. ZOHT, $k = 150$

