# OpenReview forum: "Optimization over Sparse Support-Preserving Sets: Two-Step Projection with Global Optimality Guarantees"
_ICML.cc/2025/Conference — ICML 2025 poster_

### Official Review · Reviewer_2cef · 2025-03-10

**Overall Recommendation:** 2

**Summary:**

Refer to the abstract.

## update after rebuttal:
Following the discussion I updated my recommendation to weak-reject (from rejection).
I am still not convinced that the contribution is sufficiently solid.

**Claims And Evidence:**

Proofs are provided.
I think that the assumptions are sufficiently restrictive to require some concrete examples of problems that satisfy them.

**Essential References Not Discussed:**

no

**Experimental Designs Or Analyses:**

There are some issues listed in other subsections.

**Methods And Evaluation Criteria:**

Yes.

**Other Comments Or Suggestions:**

In my opinion, the abstract is too long, and too much space is given to results that are less interesting in the context of the main contribution -- a sparse projection method that bypasses the difficulty in projecting onto sparse sets (unconstrained is not interesting since the projection is trivial).

**Other Strengths And Weaknesses:**

Strengths:
- The two projection approach is very interesting with potential
- The paper is overall well-written
- If all issues are resolved/justified, then the theoretical contribution on the constrained optimization case is worth publication

Weaknesses:
- Some results are on unconstrained optimization, which is less interesting
- RSC and RSS are potentially very restrictive nullifying the sparsity part from the sparse optimization problem
- Issues with the RSC and RSS parameters
- The paper is too long to be properly assessed in the framework of this venue

I recommend rejection because I do not believe that the issues above can be corrected in the scope of a rebuttal.
Nonetheless, I like the paper and think that if all the issues are resolved/justified it can be submitted to a journal that accepts papers of this magnitude, or be shorten and resubmitted.

**Questions For Authors:**

See comments in previous sections.

**Relation To Broader Scientific Literature:**

good

**Theoretical Claims:**

I have several issues with the theoretical claims.

The first issue, which I also mention in another subsection, concerns the scope of the paper and its results, requiring a 60-page submission (some in double-column) to an ML conference. It is unrealistic to expect a reviewer to thoroughly assess the correctness of such an extensive submission within the reviewing process of the ICML. I cannot recommend acceptance for a paper whose validity I am unable to verify to a reasonable extent.

The second issue is the assumptions. If I understand correctly (please see my proof sketch below), the RSC and RSS assumptions essentially imply that there is a unique solution to the underlying problem (1) and that any fixed-point of the projected gradient (IHT here) is this optimal solution.
This implies that the problem effectively becomes tractable, nullifying the sparsity part from the sparse optimization problem.
I speculate that by building on this fact one can establish many results from non-sparse optimization.


**Claim.** Let $y\in P_{\Gamma\cap B_0(k)}(y-L_s^{-1}\nabla R(y))$ and suppose that the RSC and RSS assumptions hold true with $s\geq 2k$. If $\nu_s > L_s$, then $y$ is the unique optimal solution of Problem (1). Consequently, it is enough to achieve stationarity wrt the sparse gradient projection operator to globally solve the optimization problem (1).

**Proof.** Any optimal solution of (1) must also be a fixed point of the hard thresholding operator, so it is enough to prove that it is the optimal solution.

By the definition of $y$ we have that

$$y\in \argmin_z \{ \langle\nabla R(y), z- y \rangle+\frac{L_s}{2}\|z-y\|^2 : z\in \Gamma \cap B_0 (k) \}$$

Therefore, in particular for the optimal solution $x$ of (1) we have that

$$0 \leq \langle\nabla R(y), x- y \rangle+\frac{L_s}{2}\|x-y\|^2$$

On the other hand, by the RSC,

$$R(x) \geq R(y)+\langle\nabla R(y), x-y \rangle+\frac{\nu_s}{2}\|x-y\|^2$$

Combining the former and latter, and using the optimality of $x$, we obtain that

$$0\geq R(x)-R(y)\geq \langle\nabla R(y), x-y \rangle+\frac{\nu_s}{2}\|x-y\|^2 \geq \frac{\nu_s-L_s}{2}\|x-y\|^2$$

Thus, if $\nu_s > L_s$ we obtain that $y=x$.


The third issue, which I am not sure if it is a typo or a more problematic mistake, is the contradiction in the results with respect to $\nu_s$ and $L_s$.
Essentially, it is required that $k \geq 4 \kappa_s^2 \bar{k}$ which means that $\nu_s \geq 2 L_s \sqrt{\bar{k} k^{-1}}$.
Considering the plausible scenario where the optimal solution has sparsity $k$, we have that $\nu_s \geq 2 L_s > L_s$; this is aligned with the claim above leading to a unique solution.
On the other hand, Thm 3.4, Thm 3.7,   take a log on $(L_s - \nu_s)<0$ which is undefined.

---

> ### Author Rebuttal · Authors · 2025-03-30
>
> Thanks a lot for your comments, we hope the answers below can address them, and we remain at your disposal for any additional questions.
>
> Regarding examples of problems satisfying the assumptions, note that we provide in Appendix G.2 and G.3 several examples, and also write there why such examples verify our assumptions.
>
> Regarding the assumptions, actually, by definition, we will always have $L_s \geq \nu_s$ (and we can never have $\nu_s < L_s$), as the (restricted) smoothness characterizes an upper bound on the function and the (restricted) strong convexity characterizes a lower bound on the function. Therefore we believe that there is no such degenerate case as you mention and we believe our analysis still remains relevant.
>
> Regarding the end of your comment, we note that $\bar{k}$ is not actually the sparsity of the optimal solution of the original problem (with k-sparse constraints): $\bar{k}$ is actually an arbitrary number (smaller than k) that is found such that the condition on $k$ is valid, typically $\bar{k}$ is taken as large as can be, but not larger. We agree that such results with $k$ and $\bar{k}$ are more complex than usual results in optimization, but such compromise with $k$ and $\bar{k}$ is unavoidable for IHT-like algorithms as proven recently by Axiotis & Sviridenko (2021; 2022), and it is standard in the literature of IHT (see Table 1). Please see also the end of the answer to reviewer EgPU on how to read and interpret such results.
>
> Regarding the weaknesses:
>
> In the case where there is no constraint $\Gamma$ (but note that there is still the sparsity constraint, which makes this case still far from trivial), we still improve upon the state of the art: in the stochastic case our constants are better than Zhou et al., and in the zeroth-order case we are the first to be able to obtain a bound on $R$ without system error, improving upon de Vazelhes et. al. (2022).
> Regarding RSC and RSS, we agree that these are somewhat restrictive, but our experimental section with the details on why we verify such assumptions, and the fact that there is a large literature on RSC and RSS problems, should make our setting still relevant.
>
> Finally, regarding the length of the paper, we agree that it is long especially in the appendix, however this is just for sake of completeness, for the case where one would want to reuse (as we hope so), some of our proofs or tools, without needing to piece many things together as would be the case if our proofs were factorized too much. But actually, the main theoretical ingredients are somewhat succinct, (though, we believe, important enough), and many long parts in the appendix are actually standard derivations from usual optimization proofs.
>
> Regarding the abstract, we will shorten it in the next revision, thanks for your suggestion.

---

> > ### Comment · Reviewer_2cef · 2025-04-03
> >
> > Thank you for your clarifications.
> >
> > I could not fully understand your explanation regarding $\bar{k}$ and $k$.
> > I will try to describe my understanding as simple as possible:
> > 1. We must have that $L_s \geq \nu_s$
> > 2. The theoretical guarantees are given wrt solutions of sparsity $\bar{k}$
> > 3. The sparsity $\bar{k}$ must satisfy that $\frac{\nu_s^2}{4 L_s^2} \geq \frac{\bar{k}}{k}$, which together with $L_s \geq \nu_s$, implies that in the best case $\bar{k} \leq k/4$
> > 4. So overall the guarantees for the solution are wrt sparsity of at most 1/4 of the desired sparsity
> >
> > Please correct any error in the above.
> > What implications do these guarantees (of 1/4) provide for the underlying problem? Why are these results interesting when optimizing with sparsity k?
> >
> > Regarding the feasible set -- yes sparsity provides a sufficiently difficult constraint, however, the motivation of the paper as I read it right from its title is the "two-step projection" which is obviously unneeded in trivial setting such as projection onto the entire space.
> > In the context of stochastic method guarantee, the same issue with the referenced sparsity arise together with the complexity of the assumptions and constants.
> >
> > Overall, I am not convinced in the contribution of the presented results.
> >
> > Nonetheless, I acknowledge that the work is positioned within existing literature and so I update my recommendation to weak reject.

---

> > > ### Author Response · Authors · 2025-04-04
> > >
> > > Dear Reviewer 2cef, thanks a lot for your response to our rebuttal, and for your update on your recommendation. Below we hope to address some of the remaining concerns you mentioned above: feel free to reach out if you still have remaining concerns.
> > >
> > > Regarding 1 to 4, you are right (by the way, sorry in our rebuttal we made a typo, we meant "we can never have $L\_s < \nu\_s$" not "we can never have  $\nu\_s < L\_s$). Indeed, as you mentioned, in the best case, $\bar{k} \leq k/4$. Regarding the implications of that result, we answered a similar concern in the last paragraph of our response to reviewer EgPU, which you may refer to: indeed, in IHT literature bounds (like our paper), while the left part of the bound contains $R(w\_{\hat{T}})$ where $w\_{\hat{T}}$ is the output of the algorithm (of sparsity $k$), the right part of the bound does not contain $R^*_{k} :=R(\arg\min\_{w ~ s.t.
> > >  ||w||\_0 \leq k} R(w))$ as is usually the case in classical optimization results, but rather, it contains instead $R^*_{\bar{k}}: = R(\arg\min_{w ~ s.t.
> > >  ||w||\_0 \leq \bar{k}} R(w))$, where $\bar{k} < k$. Since obviously $R^*_{\bar{k}} \geq R^*_{k}$ (as the $\ell_0$ pseudo-ball of radius $\bar{k}$ is included in the one of radius $k$), such bound still provides **a** guarantee: it says that after $T$ iteration of IHT, we can be sure that $R(w_{\hat{T}})$ is smaller than the bound, so it still quantifies the progress of IHT in some way (though indeed differently than in usual (non-IHT) optimization results). **Importantly though, note that this form of result is standard in the IHT literature (cf. Jain (2014), Zhou (2018), de Vazelhes (2022), Foygel Barber (2020), and it was even proven to be unavoidable (in Axiotis (2022))**, in other words, this type of bound is the best we can do for IHT (which makes sense as IHT tries to solve an NP-hard problem). **As such we believe this form of result is not a shortcoming of our work but just a special characteristic of IHT, which was proven in Axiotis (2022) to be unavoidable**. We believe it is still informative as it still provides a bound for the algorithm with sparsity $k$, even if that bound is different than usual classical optimization ones.
> > >
> > > Regarding the feasible set, you are right, if $\Gamma=\mathbb{R}$, the two step projection becomes just the hard thresholding operator. However, **even in that case, we still improve upon the state of the art**, by improving the constants from Zhou (2018), simplifying their proof, and improving upon de Vazelhes (2022) by providing the first result without system error for zeroth-order IHT. The proof techniques we employ are, even in that case, not trivial: we base our proofs on our new non-convex three point lemma and need to ingeniously deal with the gradient error (and its bias in the zeroth-order case) in the convergence proof so that the related terms vanish, in order to obtain results without system error (see our Table 1), which up to our knowledge, is novel.
> > >
> > > Again thanks a lot for your comments, and let us know if you have any additional concern, we hope this answer above can strengthen further your opinion of our work.

---

### Official Review · Reviewer_ezfv · 2025-03-17

**Overall Recommendation:** 2

**Summary:**

This paper studies a variant iterative hard thresholding (IHT) algorithm for minimizing a smooth objective over support-preserving constraints. The constraint is expressed as the k-l_0 pseudo-ball and a support preserving convex set. In the proposed variant of IHT, the orthogonal projector is approached by the composition of the projector on the sparsity l_0 constraint and then that on the support-preserving one. The paper provides convergence guarantees on the objective value in the deterministic and stochastic (with a finite sum structure) settings, as well as with a zeroth-order oracle, relying on the restricted strong convexity (RSC) and restricted smoothness (RSS) assumptions. The main technical result of the result is a firm quasinonexpansiveness result of the approximate (two-step) "projector". Once this is done, the proof techniques are rather standard.

**Claims And Evidence:**

Overall, the claims and proofs are correct as far as I can tell. Some rewriting of the main statements would deserve some rewriting (see below). However, the terminology "local convergence" for some cited works (i.e. those based on KL inequality) and "global convergence" for the current work are misleading and even unfair. The authors probably mean "convergence to a global minimizer" when mentioning "global convergence".  Actually, not only the other works are much more general, but they also prove GLOBAL convergence of the iterates to a stationary point, which is not a necessarily a global minimizer. However, they do not need the RSC condition required by the authors, while this condition is known to get rid of spurious critical points. On the other hand, the authors speak of their global convergence guarantees. In fact, they are giving only guarantees on the objective value and more precisely on the best iterate, which is not what is termed global convergence of an optimization algorithm. Moreover, they did not state anything about the convergence of the iterates, though I think this would be straightforward from the RSC.
The paper also lacks some motivating examples that justify their generalization beyond sign-symmetric sets, and the discussion in the paper is not compelling in this respect.

**Essential References Not Discussed:**

On of the first (if not the first) papers on IHT is that of Blumensath and Davies in 2008. It is surprising not to cite this seminal work.

**Experimental Designs Or Analyses:**

The paper is only theoretical and no experiments are reported.

**Methods And Evaluation Criteria:**

The paper is theoretical and discussion/comparison to related work is comprehensive enough as in Table 1.

**Other Comments Or Suggestions:**

1) It would be wise to state that problem (1) is assumed to be well-posed from the very beginning (typically R is bounded from below and the set of minimizers is non-empty).
2) Algorithm 2: in the last line "=" should be an inclusion to remove any ambiguity.
3) Foygel Barber is also cited as Barber. The former is the appropriate one.
4) Page 3: "the derivation a variant" -> "the derivation of a variant".

**Other Strengths And Weaknesses:**

Strengths:
Mostly well written with solid results.

Weaknesses:
Novelty and lack of motivation.
Overstatements.

**Questions For Authors:**

1) The "three-point lemma" (in fact Lemma 3.6, whose proof is a few lines) is claimed as the main technical finding and the crux of the other proofs. But once this is done, the rest of the proof follows the same pattern as what has been done in the literature. Could the authors argue what is really challenging then ?
2) How to estimate L_s and thus implement your IHT algorithm as the stepsize \eta depends on it ? Generally, one can do this with random sampling/design operators using random matrix theory results.
3) In Theorem 3.7 and others: What is the deep reason behind going from a uniform bound on the iterates to the best iterate here compared to IHT, i.e. the case where \Gamma is the whole space ?
4) Assumption 4.1: This is not always realistic, e.g. MSE loss wich is very standard in sparse recovery.
5) Theorem 4.2: why putting a proof if it is already in Zhou et al., 2018. Moreover, the condition on the batch size is really weird. It means the the batch size has to increase exponentially with iteration, while the objective is in a finite sum. This means that the finite sum structure has no particular interest or role in the analysis. Same for Theorem 4.3. Could the authors comment on this ?

**Relation To Broader Scientific Literature:**

The discussion of the literature is good enough though one key paper is missing (see hereafter). The relation to theory of (firm) quasinonexpansive operator theory is also missing.

**Theoretical Claims:**

Overall, the claims and proofs are correct as far as I can tell.
1) One can have statements on the iterates themselves using RSC, though probably with a system error term but this is not done here. In this respect, the discussion to the work of Vazelhes et al., 2022 is not really fair.
2) The authors should discuss sample complexity bounds under which RSC and RSS hold true (though these are known results).
3) The statements of most theorems (e.g. Theorem 3.4, Theorem 4.2, 4.3 and others) should be rephrased. Indeed, assumptions are stated before the quantities they use are defined, which is done later, and the assumptions are not stated in the correct order, etc.
4) Remark 2.4: the first example CANNOT be true in general as the projection can become dense if the hypercube does not contain the origin.
5) In the beginning of Section 3.2 and throughout the manuscript concerning the "three-point lemma": this is known as firm quasiexpansiveness in operator and fixed point theory. The authors state that this is their main theoretical finding (in fact Lemma 3.6), and once this is done, the rest of the proof follows the same pattern as what has been done in the literature. In this respect, I was not convinced by the originality of this paper (which is still 60 page long) especially for a top venue such as ICML.

---

> ### Author Rebuttal · Authors · 2025-03-30
>
> Thanks a lot for your comments, we hope the answers below can address them, and we remain at your disposal for any additional questions.
>
> Regarding terminology and prior work, by “global convergence” we don’t mean convergence to a global minimizer (which is intractable due to NP-hardness), but rather a global optimality guarantee—a bound on $ R(w)$ for iterates of sparsity $k$, in terms of $R(\bar{w})$, where $\bar{w}$ is a global optimum under sparsity $\bar{k}$ (with $k > \bar{k} $, as standard and necessary per Axiotis and Sviridenko, 2022). We agree the term “global convergence guarantees'' is misleading and will replace it by "global optimality guarantees"
>
> Regarding the theoretical claims:
>
> 1. You're right that one can derive results on the iterates with system error from guarantees on $R$ without system error, but the reverse is not true. So in IHT literature, results on $R$ without system error are considered stronger — e.g., Zhou et al. use complex support analysis to get such bounds. A main contribution of our work is to (a) simplify their proof and improve constants, and (b) extend it to the zeroth-order setting, which is harder due to gradient bias, using a new three-point lemma framework (also adaptable to extra constraints).
> 2. We agree it's helpful to add sample complexity bounds (in terms of $k$), e.g. for Gaussian designs. We’ll add this in the appendix using known results like those after Theorem 3 in Jain et al. (2014). Since our conditions on $k$, $\bar{k}$, and $\kappa_s$ are as good or better than prior work (see Table 1), our sample bounds will follow.
> 3 and 4: Thanks — we will revise the theorems (as noted also by reviewer Rh51), and add the missing sign constraint: $l_i \leq 0$, $u_i \geq 0$.
> 5. Regarding the three-point lemma: in the convex case, multiple versions exist (e.g., with Bregman divergences). But in the non-convex setting (e.g., sparsity projections), the literature is more limited. We cite Foygel-Barber et al., who discuss expansiveness (related to your mention of firm quasi-expansiveness), but their results are in deterministic settings with simple sparsity constraints. We extend this to extra constraints $\Gamma$ and to stochastic and ZO cases, with new tweaks (e.g., handling variance and bias in ZO gradients, combining inequalities to control the tradeoff via $\rho$). If there is relevant operator theory literature handling non-convex expansive operators, we’d be very interested and happy to revise our claims accordingly.
> We’ll briefly mention the operator theory connection when introducing the lemma, but note that expansiveness is a key challenge in our setting.
> Thanks for suggesting the Blumensath citation — we’ll add it in the revision.
>
>
> Comments and suggestions:
> Thank you — we will incorporate them in the next revision.
>
> Answer to Questions:
>
> 1. Yes, Lemma 3.6 is key, but other novelties lie in how we combine inequalities to derive our bounds (see also reply to 5) above).
> 2. While we don't detail $L_s$ estimation, one can use the (unrestricted) smoothness constant (when it exists), or indeed, known bounds from random matrix / compressed sensing literature.
> 3. In vanilla IHT, $R$ is non-increasing (as gradient step + euclidean projection minimizes the upper bound induced by RSS), so the last iterate is bounded. But with extra constraints (in which in general TSP $\neq$ EP) or stochastic gradients (which are not the true gradient), we lose this property. Hence, Theorems 3.7+ bound the best iterate. Bounding the last one may be possible with extra work and slightly worse constants, which we leave to future work.
> 4. You're right that the assumption (though common in many SGD papers) doesn’t always hold, though it does in common cases like logistic regression or ball-like $\Gamma$. Relaxing it is a good direction for future work, potentially using recent advances in stochastic optimization.
> 5. Although Zhou et al. prove a similar result, we include ours because (a) our constants are better, (b) our proof is simpler and fits our general framework. The batch-size schedule is classical in stochastic IHT, where the learning rate cannot be decreased due to projection expansiveness. Better use of the finite-sum (e.g., via variance reduction) is interesting, but non-trivial in IHT — we leave that for future work.

---

### Official Review · Reviewer_PBWr · 2025-03-18

**Overall Recommendation:** 4

**Summary:**

- Introduce TSP (projected version of IHT) with iteration rule:
$$\omega_{t+1} =  \Pi_\Gamma \circ \Pi_{B_0 (k)} (\omega_{t} - \eta \nabla R(\omega_{t}))$$
- Introduce three-point lemma for hard thresholding $(\Pi_{B_0 (k)})$:
If $\bar \omega \in B_0 (\bar k)$ then
$$\|w-\bar w\|^2 \ge \|\Pi_{B_0 (k)} w - w\|^2 + \|\Pi_{B_0 (k)} w - \bar w\|^2 - \sqrt{\beta}\|\Pi_{B_0 (k)} w - \bar w\|^2$$
Where $\beta = \frac{\bar k}{k}$ and $\bar k \le k$
- Provide global convergence guarantees for different versions of TPS (deterministic, stochastic, zeroth-order) in special settings: restricted strong convexity and restricted strong smoothness on objective (standard), and support-preserving set on constraints (new, says that projection onto $\Gamma$ can't make a zero-coordinate suddenly non-zero)
$$supp(\Pi_\Gamma w)\subseteq supp(w)  \quad  \forall w\in B_0(k)$$

**Claims And Evidence:**

I checked Section D of the Appendix and found no problems there.

**Essential References Not Discussed:**

I am not aware of such literature.

**Experimental Designs Or Analyses:**

I checked Section G of the Appendix and found no problems there.

**Methods And Evaluation Criteria:**

The paper is mainly theoretical, but the empirical validation in Seciton G2 and G3 made sense.

**Other Comments Or Suggestions:**

Authors relaxed the assumptions on the constraint set $\Gamma$ in their guarantees. For me, as a reader, it would be interesting to learn which of the practical problems satisfy the new condition (support-preserving set) but did not satisfy the conditions considered previously (\ell_\infty ball, symmetric convex sets). It would also be interesting to understand which of the practical problems do NOT satisfy the condition of support-preserving sets.

**Other Strengths And Weaknesses:**

The work is original, presents an elegant algorithm that is simple to understand and implement. It provides with a readable explanation of complex results.

**Questions For Authors:**

No questions

**Relation To Broader Scientific Literature:**

This work is a generalization of the previous results on sparsity-constrained optimization, which is by itself a subfield of non-convex optimization.
It provides with a valuable addition to the family of hard thresholding algorithms. The alternative methods involve regularization-based approaches and approaches through mixed-integer programming. The paper does not make direct comparisons to the methods outside of its realm.

**Theoretical Claims:**

I checked Section D of the Appendix and found no problems there.

---

> ### Author Rebuttal · Authors · 2025-03-30
>
> Thanks a lot for your review and appreciation of our work. Regarding practical problems which satisfy the new condition but did not satisfy conditions considered previously, actually our example on portfolio optimization (in appendix G.2) with sector-wise constraints, is such an example: it is not a symmetric convex set, as one cannot ensure that a point from the set remains in the set if we swap its coordinates. As we describe in such an example, such a set does not admit a closed form for Euclidean projection onto it, and our two-step projection allows us to tackle it. As for examples of sets which do NOT satisfy the condition of support-preserving sets, a simple example is a hyperplane which does not contain the origin: in such case, the projection of a k-sparse point onto onto $\Gamma$ can be dense in general, hence the support of the point after projection may not be included in the original support of size k of the initial point.

---

### Official Review · Reviewer_A66t · 2025-03-18

**Overall Recommendation:** 4

**Summary:**

This paper studies with the problem of optimizing a convex function over the intersection of a sparsity ($\ell_0$) constraint and other convex constraints $\Gamma$. The additional constraints are required to be support-preserving, i.e. such that any orthogonal projection onto $\Gamma$ preserves the support. This is important, since otherwise projection can blow up the number of non-zeros in the solution.

The authors present an analysis of an iterative hard thresholding (IHT) variant for the above problem, which applies a two-step projection -- first apply the sparse projection, and then $\Pi_\Gamma$. Typical IHT analyses from previous work guarantee a solution with $k = O((L/\nu)^2 \bar{k}^2)$ non-zeros (i.e. $||w||_0 \leq k$), where $L, \nu$ are the restricted smoothness and strong convexity parameters of the function being optimized and $\bar{k}$ is the number of non-zeros of the target solution. In addition, the objective value can come arbitrary close to the optimum, i.e. $R(w) \leq R(\bar{w}) + \epsilon$ for any $\epsilon > 0$.

In the authors' analysis, adding the support-preserving constraint changes the objective bound to
$R(w) \leq (1+\rho)R(\bar{w}) + \epsilon$ with $k = O((L/\nu)^2 \bar{k}^2 / \rho^2)$. In addition, the authors extend their analysis to the stochastic and zero-order optimization settings. The proof technique builds upon the work of Liu & Foygel Barber 2020 by incorporating the additional convex projection at each step.

The authors present some syntethic and real experiments to validate the benefit of projecting at each step instead of a cruder post-training projection.

Overall, I beileve the direction of incorporating projections into the results from sparse optimization literature is a theoretically and practically relevant question, as I do not believe it is well understood how well sparse projections interact with other projections. On the other hand, the factor $\rho$ is undesirable and weakens the result. I also like the stochastic extension although I did not read it carefully.

**Claims And Evidence:**

In general the claims look correct to me, although I did not check the stochastic and zero-order sections.

One observation is about the necessity of $\rho$. I don't see a fundamental reason why the $\rho$ dependency on the sparsity should be there. Perhaps this is confirmed by the experiments in the appendix that show that the actual bounds are far from the theoretical predictions. The dependency is undesirable, since to get down to additive $\epsilon$ error, the sparsity blows up by $(R(\bar{w}) / \epsilon)^2$, so practically speaking it's not possible to get to arbitrarily small error. If the authors disagree and have a concrete reason why this dependency is necessary, I would be interested to hear that.

By the way, another way to rephrase the main result (maybe add as Corollary), is by adding an $\ell_2$ regularizer to the objective: $(\rho R(w^*) / ||w^*||^2) ||w||^2$. This will not change the result qualitatively (it will only add additive $\rho R(w^*)$ error), but allows getting rid of the RSC requirement, since the objective is automatically $(\rho R(w^*) / ||w^*||^2)$ - RSC.

**Essential References Not Discussed:**

N/A

**Experimental Designs Or Analyses:**

I believe that the baselines used in the SP500 experiments could be improved. For the intersection of $\ell_0$ and the cartesian product of $\ell_1$ balls, I believe the following $\ell_1$-based projection might be worth comparing against: Prune entries iteratively by in each iteration pruning 1) the smallest entry that is in a violated $\ell_1$ constraint and 2) the smallest entry overall, if there are no violated constraints. I expect that a slight tweak of this method will also be the optimal $\ell_1$ projection.

**Methods And Evaluation Criteria:**

This is mainly a theory paper. The synthetic and real tasks (e.g. SP500 sparse prediction) make sense to me.

**Other Comments Or Suggestions:**

- Are the results extensible to non-support preserving projections under some assumptions? Is it possible to define a soft version of non-support preserving projection and have the analysis still go through?

- The authors could mention the main result of Axiotis et al 2022 somewhere in their intro. While it is not analyzing vanilla IHT, it is relevant enough that it should be added to the context IMO. In the future it could be also interesting to see if the authors' results are compatible with the results from that paper.

- I would optionally suggest naming it 2-SP or 2SP instead of TSP, since the latter points to the traveling salesman problem.

**Other Strengths And Weaknesses:**

The paper is very well written and easy to follow.

**Questions For Authors:**

Added above.

**Relation To Broader Scientific Literature:**

The authors build on top of a series of works analyzing the IHT algorithm by Jain et al, Liu et al, etc. Their contribution is technical, and has to do with simplifying and modifying the core ideas from these analyses, as well as applying it to stochastic and zero-order applications.

**Theoretical Claims:**

I checked the soundness of the claims and skimmed the proofs in Section 4. They look correct to me.

---

> ### Author Rebuttal · Authors · 2025-03-30
>
> Though we leave a more insightful and in-depth investigation of the tightness of our bound in terms of $\rho$ to future work, the main reason why there is a dependence in $\rho$ is that the original three point lemma is not valid anymore, and an extended version needs to be done, which contains an extra term, see our Lemma 3.6 compared to Lemma 3.2. This extension that we come up with, and the corresponding proof technique that we use in Theorem 3.7, comes from the difficulty to directly characterize the expansiveness coefficient of our complex mixed constraints. We don’t know if such a general characterization would be possible though, but we leave that for future work, since we believe this would be beyond the reach of this paper: such characterization would mostly likely be very complex, as already the vanilla $\ell_0$ three points lemma itself relies heavily on very specific properties of projection onto a sparse constraint. For more details, please check our proof which relies mostly on Lemma 2 from Liu and Foygel Barber, itself making heavy use of the sparsity structure of the set and relying on advanced tools such as the Borsuk-Ulam theorem (cf. proof of Lemma 2 in Liu and Foygel Barber et. al.).
>
> Regarding the SP500 experiment, this baseline is indeed interesting and we will add it in the next revision.
>
> **Other comments and suggestions:**
>
> Non-support preserving projections: this is an interesting question, we believe if one is able to quantify the amount of expansiveness induced by such sets (i.e. the modification of the constants in our three point lemmas), then one will be able to reuse our proofs for the most part.
>
> We agree that it would be best to mention Axiotis et al in the Intro, we will do so in the next revision. Indeed, it would be interesting to modify the algorithm of Axiotis et al to work with the constraints in our paper, though we leave this for future work.
>
> You are right, 2-SP is a better name, we will adapt the next revision accordingly.

---

### Official Review · Reviewer_EgPU · 2025-03-22

**Overall Recommendation:** 2

**Summary:**

1. This paper considers a variant of IHT that addresses sparse optimization problems while incorporating additional convex constraints.
2. The authors propose a two-stage projection gradient method.
3. They evaluate the effectiveness of these approaches under both stochastic and non-stochastic settngs.

**Claims And Evidence:**

yes.

**Essential References Not Discussed:**

No

**Experimental Designs Or Analyses:**

A comparison with state-of-the-art approaches is lacking.
I suggest the authors compare their method with the coordinate-wise optimization techniques proposed by Amir Beck et al.

**Methods And Evaluation Criteria:**

yes

**Other Comments Or Suggestions:**

No

**Other Strengths And Weaknesses:**

**Strengths**

1. This paper is generally well-written and provides a clear presentation of the current state of art.

2. The authors propose a new IHT-style algorithm which is based on two-step projection methods.

3. Their theoretical results establish global convergence of the objective value and provide a new bound on the solution’s sparsity of order $\mathcal{O}(\kappa^2 \bar{k})$, which improves upon existing methods.



**Weaknesses**

1. This work, building on Jain et al. (2014), focuses on deriving tight bounds for the sparsity level $s$ without introducing much error in the objective value. However, there is a noticeable gap between theory and practice. In practical applications, we typically aim to obtain a $k$-sparse solution, where $k$ is a fixed integer specified by the user. As a result, additional objective error is inevitably introduced. This discrepancy makes the theoretical guarantees provided in Theorems 3.7, 4.3, 4.7, and 4.8 less compelling or meaningful in real-world settings.

2. The authors consider a $k$-sparsity constrained problem, but there is no theoretical guarantee on the objective value bounds for a **fixed** sparsity $k$, even when $k$ is allowed to be an arbitrarily large constant.

3. The authors should benchmark their approach against Amir Beck’s coordinate-wise optimization method, a strong baseline, especially for the portfolio index tracking problem.

**Questions For Authors:**

NO

**Relation To Broader Scientific Literature:**

This work builds on seminal contributions to sparse optimization, particularly iterative hard-thresholding (IHT) as presented in (Jain et al., 2014; de Vazelhes et al., 2022), which demonstrated strong global convergence guarantees.

**Theoretical Claims:**

I have reviewed certain sections of the proofs.

---

> ### Author Rebuttal · Authors · 2025-03-30
>
> Thanks a lot for your comments, we hope the answers below can address them, and we remain at your disposal for any additional question.
>
> Regarding the comparison with the coordinate-wise optimization method by Beck et al., we cite that paper in our paper but do not compare our convergence rates in detail with it, as Beck et al. only give local convergence guarantees: they analyze properties of stationary points. Regarding an experimental comparison and whether the algorithm of Beck would be better than our algorithm empirically, we can consider 3 cases:
> 1) (See also the related Appendix F, Case (ii, a) and (ii, b)) The sets considered cannot be tackled by the algorithm from Beck. In such a case, it can happen that our algorithm can tackle them: this is the case for instance in our experiments in Appendix G.2. There, our algorithm provides an advantage.
> 2) (See Appendix F, Case (ii, c)) The sets considered can be tackled by Beck et al. (such as for sign-free symmetric sets): in such a case, the algorithm from Beck is expected to perform better than our algorithm as it is itself an improvement on projected gradient descent (PGD), and PGD is itself likely to be better than our algorithm at least at the iteration level, as we describe in Appendix F, Case (ii, c).
> 3) (See Appendix F, Case (i)) The sets considered are such that the two step projection IS equal to the euclidean projection: similarly, in that case, the algorithm of Beck is expected to be better, at least at the iteration level, as it is an improvement over projected gradient descent.
>
> But note that in cases 2 and 3 above, the contribution of our paper is still present on the theoretical side: we do not pretend to provide a better algorithm, but rather, we give global guarantees on the result returned by the algorithm. Then one could run both our algorithm and Beck’s algorithm, and if Beck’s algorithm returns a better function value than our algorithm, our global guarantee will also of course also apply (as it is an upper bound on R) to the output of Beck’s algorithm, which can potentially be useful in various applications where having guarantees on sub-optimality is important.
>
> Regarding the relaxation of the sparsity k, note that this presentation of results is classical in the literature of IHT with global guarantees, as in the works of Jain et al. (2014), Zhou et al. (2018), and de Vazelhes et. al. (2022). It was also recently proven in Axiotis & Sviridenko, 2022) that such relaxation is unavoidable for IHT (which makes sense as a “true” global guarantee is impossible to obtain due to the NP-hard nature of the problem), as we recall in our Remark 3.5. However, one way to read the results which is maybe more informative, is to consider the sparsity k of the iterates fixed, but to accept that the right part of the upper bound on convergence contains a term in $R(\bar{w})$, that is, an “optimum function value” for an iterate $\bar{w}$ of smaller sparsity. Therefore our bounds (and similar bounds in the literature) are actually bounds on the iterates of our algorithm, but it is just that the “reference point” is an optimum of a more constrained problem: as such it still offers some guarantee on the actual iterates of the algorithm.

---

### Official Review · Reviewer_Rh51 · 2025-03-23

**Overall Recommendation:** 1

**Summary:**

This paper considers the problem of  minimizing a function subject to a mixed constraints here two. One of the constraint enforces  sparsity via the pseudo-norm $\ell_0$ which make this constraints nonconvex and hard to deal with. The try to force the solution to belong into a convex set $\Gamma$. The authors consider a class of set which are support preserving sets. They define such sets are convex sets for which the projection of a k-sparse vector onto them, preserved the support. To solve the problem, they consider the approach of Iterative Hard Thresholding modified with a two-step projection operator. They claimed to provide global sub-optimality guarantees without system error for the objective value, for such an algorithm as well as its stochastic and zeroth-order variants, under the restricted strong-convexity and restricted smoothness assumptions.

**Claims And Evidence:**

This paper modified the Iterative Hard Thresholding to solve the problem of minimizing an objective $R$ subject to a mixed constraint
(sparsity and suport preserving sets). The analyzes seems based on an extension of the famous three point identity also know in the Bregman setting to the case  of their projections. They provide global convergence guarantees in objective value without system error for the algorithm above, in the RSC/RSS setting, highlighting a novel trade- off between sparsity of iterates and sub-optimality gap in such mixed constraints setting.

**Essential References Not Discussed:**

Teboulle, Chen and Bolte

**Experimental Designs Or Analyses:**

Not interesting if the theoretical part is not so well proved.

**Methods And Evaluation Criteria:**

The theoetical part of this paper recquire a better analysis espcially for each algorithms of interest. Thus, the numerical simulation, benchmarks datasets set are not very relevant if the  Theoretical part is not so rigourous.

**Other Comments Or Suggestions:**

I strongly suggest to the authors to rewrite the paper and consider to start analysing properly the deterministic case of the problem for one paper.

**Other Strengths And Weaknesses:**

No

**Questions For Authors:**

Have you consider study in detailed just one case ?

**Relation To Broader Scientific Literature:**

Regarding your three point identity you should cite Teboulle, Chen and Bolte.

**Theoretical Claims:**

We checked the correctness of the  proofs of this paper. The main Claim are stated in Theorem 3.4, 3.7, 4.2, 4.3, 4.7 and 4.8.
In line 153, it shoud be written Definition 2.3 instead of Assumption 2.3. The authors of the paper did not provide no direct assumption on the regularity of the objective before starting imposing Lischitz continuity of the gradient or strong convexity.
This result are not so general as you claimed. Not that the support set is a Riemannian manifold and Riemannian optimization is not so easy. Each extension  are not so obvious and each time you  take $\Gamma=\mathbb{R}^d$ and thus doing Riemmanian optimization.
Your two step projection is just Alternating projection and you check authors such as  Bauchke.

---

> ### Author Rebuttal · Authors · 2025-03-30
>
> Thanks a lot for your comments, we hope the answers below can address them, and we remain at your disposal for any additional questions.
>
> Q1: Assumption 2.3
>
> A1: For Assumption 2.3, we will write it as a definition and state in the theorems that our objects verify such definition.
>
> Q2: The regularity of the objective
>
> A2: For the regularity of the objective, we indeed implicitly assume that the gradient is well defined (otherwise our assumption is void), so we will add this explicitly in the new revision.
>
> Q3: The support set is a Riemannian manifold and Riemannian optimization is not so easy.
>
> A3: I need to clarify that the support set defined as the indices of nonzero parameters, is a discrete set, the subspace corresponding to a support set is a Euclidean space, NOT a non-trivial Riemannian manifold. Thus, the optimization defined on the support set which can be optimized easily in the most cases.
>
> Q4: References
>
> A4: Regarding the related references, we are aware that there is an extensive related literature on proximal gradient descent and its generalizations (such as Riemannian optimization), as cited in our introduction (for instance, we already cite Bolte, Sabach, and Teboulle(2014)), and we appreciate the references you recommend, which are indeed relevant and which we will add in the next revision (for instance we can add [1], [2], and other related references by similar authors). Note however that we did NOT elaborate too exhaustively on such literature, as it actually considers local optimization guarantees such as convergence to local minima or stationary points, which are very different from the global optimization guarantees that we derive in our paper. In a way, our paper is much closer (in terms of assumptions, proof techniques, and results) to papers such as Jain (2014), Zhou (2018), or de Vazelhes (2022).
>
> As mentioned above, we appreciate the reviewer can double-check the contributions of the paper and reconsider the evaluation.
>
> [1] A Descent Lemma Beyond Lipschitz Gradient Continuity: First-Order Methods Revisited and Application, Heinz H. Bauschke, Jérôme Bolte, Marc Teboulle
>
>
> [2] On Linear Convergence of Non-Euclidean Gradient Methods without Strong Convexity and Lipschitz Gradient Continuity, Heinz H. Bauschke, Jérôme Bolte, Jiawei Chen, Marc Teboulle, Xianfu Wang

---

### Decision · Program_Chairs · 2025-05-01

**Decision:**

Accept (poster)

**Comment:**

This submission studies an iterative hard thresholding algorithm for minimizing a smooth function over so-called support-preserving constraints. The main development here is how the projection onto the constraint set is handled. The paper provides theoretical results in the form of convergence analysis of the objective function value in the determinstic and stochastic/finite-sum settings. There are also results for a zeroth-order oracle version of the algorithm and eventually some experiments.

The reviewers were split on this paper, with two reviewers giving scores of 4 and four reviewers giving scores of 1 or 2. The primary concerns of the negative reviewers were the incremental nature of the results (the algorithm and its analysis are relatively minor modifications of IHT and prior work) and the potentially strong assumptions used in some of the analysis. In the rebuttal period the authors were successful in convincing a reviewer to change their score from 1 to 2 and they addressed many of the concerns of other reviewers (questions about assumptions, citations to prior work).

For these reasons, I am weakly recommending to accept the paper.